EMBO
Molecular Medicine

# Disease-specific phenotypes in iPSC-derived neural stem cells with *POLG* mutations

Kristina Xiao Liang[1,2,†,*] , Cecilie Katrin Kristiansen[2,†], Sepideh Mostafavi[2], Guro Helén Vatne[2],
Gina Alien Zantingh[3], Atefeh Kianian[2], Charalampos Tzoulis[1,2], Lena Elise Høyland[4], Mathias Ziegler[4] ,
Roberto Megias Perez[4] , Jessica Furriol[2,5], Zhuoyuan Zhang[6,7], Novin Balafkan[2], Yu Hong[1,2],
Richard Siller[8,9], Gareth John Sullivan[8,9,10,11,12] & Laurence A Bindoff[1,2,**]

## Abstract

Mutations in *POLG* disrupt mtDNA replication and cause devastating diseases often with neurological phenotypes. Defining disease mechanisms has been hampered by limited access to human tissues, particularly neurons. Using patient cells carrying *POLG* mutations, we generated iPSCs and then neural stem cells. These neural precursors manifested a phenotype that faithfully replicated the molecular and biochemical changes found in patient post-mortem brain tissue. We confirmed the same loss of mtDNA and complex I in dopaminergic neurons generated from the same stem cells. POLG-driven mitochondrial dysfunction led to neuronal ROS overproduction and increased cellular senescence. Loss of complex I was associated with disturbed NAD$^+$ metabolism with increased UCP2 expression and reduced phosphorylated SirT1. In cells with compound heterozygous *POLG* mutations, we also found activated mitophagy via the BNIP3 pathway. Our studies are the first that show it is possible to recapitulate the neuronal molecular and biochemical defects associated with *POLG* mutation in a human stem cell model. Further, our data provide insight into how mitochondrial dysfunction and mtDNA alterations influence cellular fate determining processes.

**Keywords** mitochondria; mitophagy; neural stem cells; POLG; reactive oxygen species
**Subject Categories** Genetics, Gene Therapy & Genetic Disease; Neuroscience; Stem Cells & Regenerative Medicine

## Introduction

Mitochondria are membrane enclosed, intracellular organelles involved in multiple cellular functions, but best known for generating adenosine triphosphate (ATP). Mitochondria are the only organelles besides the nucleus that possess their own DNA (mitochondrial DNA; mtDNA) and their own machinery for synthesizing RNA and proteins. DNA polymerase gamma, Polγ, is a heterotrimeric protein that catalyzes the replication and repair of the mitochondrial genome. The holoenzyme is a heterotrimer composed of one catalytic subunit (POLG) with the size of 122 kDa, encoded by the *POLG* gene, and a dimer of two accessory subunits (POLG2) of 55 kDa encoded by *POLG2*.

Mutations in *POLG* cause a wide variety of diseases that vary in age of onset and severity. More than 200 disease-causing mutations are known, and these cause diverse phenotypes including devastating early onset encephalopathy syndromes such as Alpers' syndrome (Naviaux & Nguyen, 2004; Ferrari *et al*, 2005) or severe adult onset disorders with progressive spinocerebellar ataxia and epilepsy (Van Goethem *et al*, 2004; Hakonen *et al*, 2005; Winterthun *et al*, 2005). Other phenotypes include progressive external ophthalmoplegia (PEO) (Lamantea *et al*, 2002) and parkinsonism (Luoma *et al*, 2004). Mitochondrial dysfunction is also

1 Neuro-SysMed, Center of Excellence for Clinical Research in Neurological Diseases, Haukeland University Hospital, Bergen, Norway
2 Department of Clinical Medicine, University of Bergen, Bergen, Norway
3 Leiden University Medical Centre, Leiden University, Leiden, The Netherlands
4 Department of Biomedicine, University of Bergen, Bergen, Norway
5 Department of Medicine, Haukeland University Hospital, Bergen, Norway
6 State Key Laboratory of Oral Diseases, West China School of Stomatology, Sichuan University, Chengdu, China
7 Department of Head and Neck Cancer Surgery, West China School of Stomatology, Sichuan University, Chengdu, China
8 Department of Molecular Medicine, Institute of Basic Medical Sciences, University of Oslo, Oslo, Norway
9 Norwegian Center for Stem Cell Research, Oslo University Hospital and University of Oslo, Oslo, Norway
10 Institute of Immunology, Oslo University Hospital, Oslo, Norway
11 Hybrid Technology Hub - Centre of Excellence, Institute of Basic Medical Sciences, University of Oslo, Oslo, Norway
12 Department of Pediatric Research, Oslo University Hospital, Oslo, Norway
*Corresponding author. Tel: +47 55 97 50 96; E-mail: Xiao.Liang@uib.no
**Corresponding author. Tel: +47 55 97 57 04; E-mail: Laurence.Bindoff@uib.no
†These authors contributed equally to this work

implicated in the pathophysiology of common forms of neurodegeneration, such as Parkinson's disease. Studying the effect of *POLG* mutation on mitochondrial function and cellular homeostasis is, therefore, relevant to a wide spectrum of diseases.

Our previous studies using post-mortem human brain revealed that while POLG-related disease caused widespread damage in the brain, dopaminergic neurons of the substantia nigra were particularly affected (Tzoulis *et al*, 2014). In addition to progressive loss, nigral neurons also showed an age-related progressive accumulation of mtDNA deletions and point mutations (Tzoulis *et al*, 2014). While informative, post-mortem studies often represent the end stage of disease and are not tractable. The need for models to study disease mechanisms is, therefore, clear, and since mouse models often fail to recapitulate the human neural phenotype, we chose to examine the potential of induced pluripotent stem cells (iPSCs). IPSCs retain the potential to differentiate into any cell type and, while still at an early developmental stage, carry the disease mutation and the patients' own genetic background, giving us the possibility to study disease during tissue development (Marchetto *et al*, 2011).

Primary neural stem cells (NSCs) provide a continued source of neurons and glial cells in the brain that further serve as a foundation for development, repair, and functional modulations of human adult neurogenesis. It is not surprising, therefore, that dysfunction of neural precursor cells contributes to an assortment of neurological disorders (Li *et al*, 2018). While primary NSCs derived from patients have the ability to circumvent immune rejection, they are hard to acquire and display a limited expansion and engraftment capacity. A solution to this problem is the neural induction of iPSCs to NSCs, either through an intermediate rosette-like stage or directly by application of a cocktail of small molecules (Lorenz *et al*, 2017). Since iPSC-derived NSCs are relatively straightforward to generate, they provide an alternative to primary NSCs for disease-relevant phenotype studies and drug development (Griffin *et al*, 2015; Lorenz *et al*, 2017; Li *et al*, 2018). Others have also suggested that NSC models might provide new insights into mtDNA disorders (Kim *et al*, 2013).

Mitochondria conserve the energy generated by substrate oxidation and use this to generate a membrane potential. The proton electrochemical gradient, termed the mitochondrial membrane potential (MMP), provides the energy that drives ATP synthesis. The MMP also regulates mitochondrial calcium sequestration, import of proteins into the mitochondrion and mitochondrial membrane dynamics. Mitochondria are the major producer of superoxide and other downstream reactive oxygen species (ROS) in the cell (Bae *et al*, 2011), with the main sites of mitochondrially derived superoxide being complexes I and III (Brand, 2016). Mitochondrially generated ROS can mediate redox signaling or, in excess, affect replication and transcription of mtDNA and result in a decline in mitochondrial function, which in turn, can further enhance ROS production (Cui *et al*, 2012).

Accumulating evidence demonstrates that ROS plays a critical role in induction and maintenance of cellular senescence (Davalli *et al*, 2016; Zheng *et al*, 2018). This state of stable cell cycle arrest, in which proliferating cells become resistant to growth-promoting stimuli, typically occurs in response to DNA damage. Cellular senescence also appears to cause mitochondrial dysfunction including loss of membrane potential, decreased respiratory coupling, and

increased ROS production, and these changes were associated with altered mitophagy (Korolchuk *et al*, 2017). Mitophagy is the autophagic pathway involved in mitochondrial quality control that removes damaged mitochondria and regulates mitochondrial number to match metabolic demand. In mammals, more than 20 proteins have been associated with this process, including PTEN-induced kinase 1 (PINK1), parkin, serine/threonine-protein kinase ULK1 (ULK1), BCL2/adenovirus E1B 19 kDa protein-interacting protein 3-like (BNIP3L/NIX), and serine/threonine-protein kinase TBK1 (TBK1) (Kerr *et al*, 2017). Alterations in mitophagy have been linked to neurodegenerative diseases such as Alzheimer' disease (AD) (Fang *et al*, 2019). Mitophagy also plays a vital role in neuronal function and survival by maintaining a healthy mitochondrial pool and inhibiting neuronal death (Fang *et al*, 2019). The role of mitophagy in mitochondrial disease such as those caused by *POLG* mutation remains, however, unclear.

In the present study, we generated an experimental model for POLG-related brain disease using iPSCs reprogrammed from patient fibroblasts that were differentiated to NSCs. NSCs showed defective ATP production and increased oxidative stress reflected by elevated levels of intracellular and mitochondrial ROS. In addition, we found depletion of mtDNA and loss of mitochondrial respiratory chain complex I, findings that precisely recapitulate those from post-mortem tissue studies. Further mechanistic studies showed that these neural cells had disturbed $NAD^+$ metabolism-mediated UCP2/SirT1 and increased cellular senescence and BNIP3-mediated mitophagy, which may contribute to pathological mechanisms involved in this form of mitochondrial neurodegeneration.

## Results

### Generating iPSCs from patient cells carrying *POLG* mutations

We generated iPSCs from parental fibroblasts from two patients carrying *POLG* mutations, one homozygous for c.2243G>C; p.W748S (WS5A) and one compound heterozygous c.1399G>A/c.2243G>C; p.A467T/W748S (CP2A). The clinical symptoms of both patients included ataxia, peripheral neuropathy, stroke-like episodes, and PEO (Tzoulis *et al*, 2006, 2014). Fibroblast lines Detroit 551, CCD-1079Sk and AG05836 were reprogrammed as disease-free controls, and two different human embryonic stem cells, H1 (ESC1) and line 360 (ESC2), were used as internal controls for iPSC generation and characterization.

All fibroblasts were reprogrammed into iPSCs using a retroviral or Sendai virus system. Four retrovirus viral particles including *hOCT4, hSOX2, hKLF4,* and *hcMYC* were transduced at an MOI of 5 according to a previously described report (Siller *et al*, 2016). In order to account for clonal variation arising during iPSC reprogramming, 2–4 clones from each iPSC line were selected and used for further investigation (Appendix Table S1).

The Detroit 551 control and patient fibroblast (WS5A, CP2A) derived iPSCs displayed typical ESC morphology with well-defined sharp edges and contained tightly packed cells (Fig 1A). No obvious different appearance was noticed in patient WS5A and CP2A iPSCs compared to iPSCs generated from Detroit 551 control (Fig 1A). In order to confirm normal karyotypes for all the reprogrammed iPSC

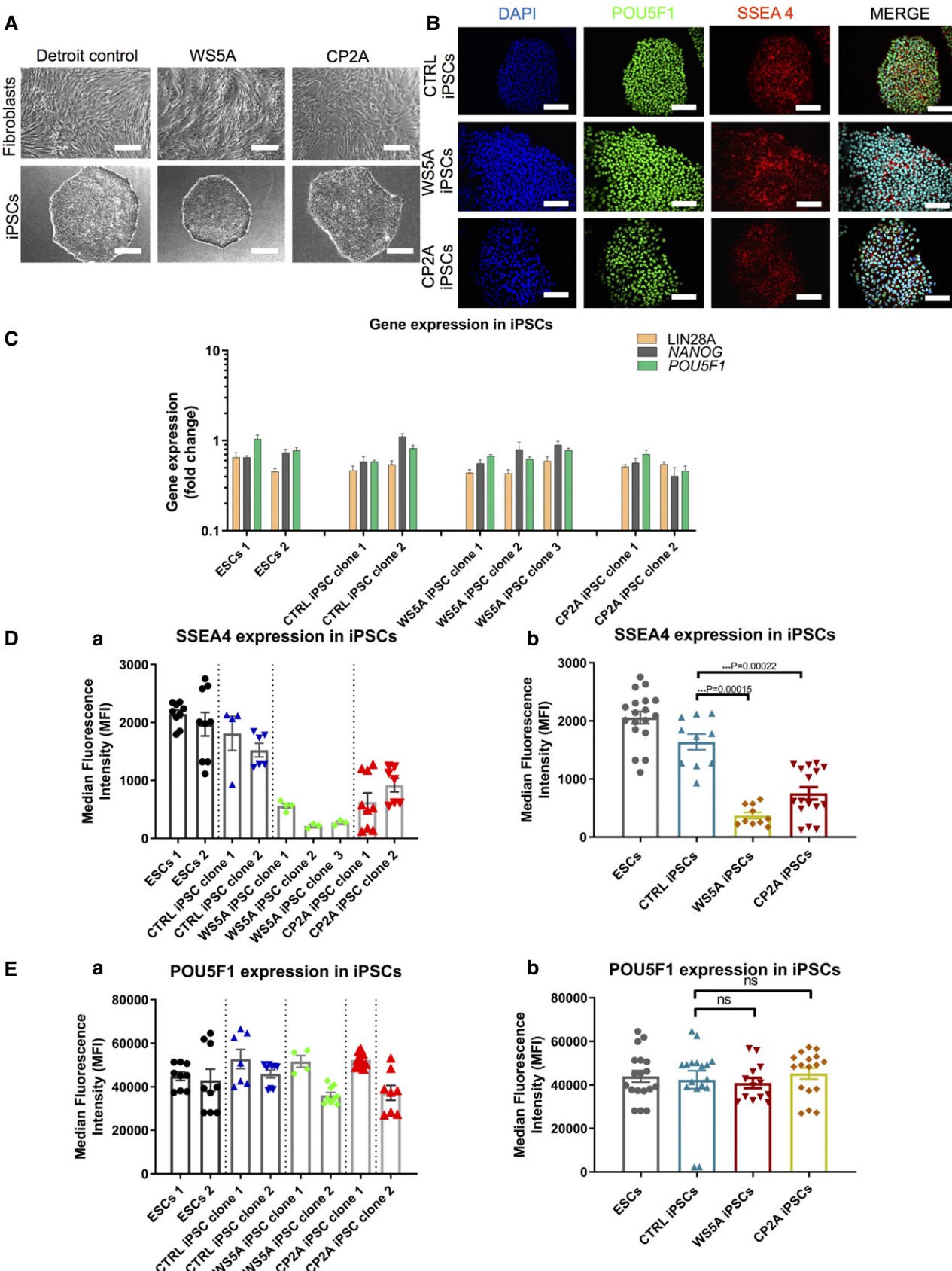

Figure 2.

◀

**Figure 1.  IPSCs generated from patient fibroblasts carrying homozygous and heterozygous *POLG* mutations.**

A  Morphology on phase contrast microscopy for parental fibroblast lines (upper panel) and iPSCs (lower panel) from Detroit 551 control, WS5A, and CP2A POLG patients (scale bars, 50 μm).

B  Immunofluorescence staining of stem cell markers POU5F1 (green) and SSEA4 (red): upper panel—Detroit 551 control iPSCs, middle panel—WS5A iPSCs, and lower panel—CP2A iPSCs (Scale bar, 100 μm). Nuclei are stained with DAPI (blue).

C  RT-qPCR quantification of gene expression for *LIN28A*, *NANOG*, and *POU5F1* for all iPSCs from Detroit 551 control, WS5A, and CP2A POLG patients (*n* = 7, technical replicates per line for ESCs; *n* = 4, technical replicates per clone for control, WS5A, and CP2A iPSCs). The gene expression of the individual clones is assessed with fold change using the comparative $\Delta\Delta C_t$ method by normalizing iPSCs to ESC1.

D, E  Flow cytometric quantification of expression level of SSEA4 (D, *n* = 9, technical replicates per line for ESCs; *n* = 5, technical replicates per clone for control iPSCs; *n* = 3, technical replicates per clone for WS5A iPSCs, *n* = 8, technical replicates per clone for CP2A iPSCs) and POU5F1 (E, *n* = 9, technical replicates per line for ESCs; *n* = 5, technical replicates per clone for control iPSCs; *n* = 3, technical replicates per clone for WS5A iPSCs; *n* = 8, technical replicates per clone for CP2A iPSCs) for ESCs and iPSCs for both ESC control lines and iPSCs generated from Detroit 551 control, WS5A and CP2A fibroblasts.

Data information: The data presented in C–E were generated from 2 distinct ESC lines, 2 iPSC clones from Detroit 551 control, 3 different clones from WS5A patient, and 2 different clones from CP2A patient iPSCs. Data in D and E are presented as individual (a) and combination as a group (b). Data are presented as mean ± SEM for the number of samples. Mann–Whitney *U*-test was used for the data presented. Significance is denoted for *P* values of less than 0.05. \*\*\**P* < 0.001.

Source data are available online for this figure.

lines, G banding analysis was used. We showed that all iPSC lines presented with the same karyotype as their parental human fibroblasts after reprogramming, with no evidence of chromosomal abnormalities (Fig EV1).

Next, we characterized the reprogrammed iPSCs for their pluripotency using immunostaining and flow cytometry for protein expression and RT-qPCR analysis for gene expression. Immunostaining confirmed that all the iPSCs expressed the specific pluripotent markers POU5F1 and SSEA4 (Fig 1B). RT-qPCR analysis showed similar mRNA expression levels in terms of *LIN28A*, *POU5F1,* and *NANOG* in iPSC clones from Detroit 551 control, WS5A, CP2A, and ESC lines (Fig 1C). In addition, we measured expression of pluripotent transcription factors POU5F1, NANOG and pluripotent surface markers SSEA4, TRA-1-60, and TRA-1-81 and quantified these using flow cytometric analysis. Using this technique, we observed that both the ESC and iPSC lines exhibited over 98% of POU5F1-positive cells and over 88% of the cells showed positive staining for SSEA4 (Appendix Fig S1). Interestingly, we detected a lower level of the three pluripotent surface markers SSEA4 (Fig 1D(a and b)), TRA-1-60 (Fig EV2A(a and b)), and TRA-1-81 (Fig EV2B(a and b)) in both WS5A and CP2A iPSCs compared to the two ESC lines and control iPSC line. However, no changes were observed in the expression of POU5F1 (Fig 1E(a and b)). We observed a higher expression of NANOG in WS5A compared to control but not in CP2A lines (Fig EV2C(a and b)). In addition, clonal variations for the protein level and mRNA expression were noticed (Figs 1C and D(a), E(a) and EV2A(a), B(a), C (a)). In order to minimize the phenotypic diversity caused by intraclonal heterogeneity, multiple clones were included in the further analysis.

Next, we demonstrated that the iPSCs we generated retained the potential to differentiate into cell types associated with all three germ layers. We generated hepatocytes (endoderm) with positive expression of albumin and HNF4A (Fig 2A(a)), cardiomyocytes (mesoderm) with positive expression of TNNT2 (Fig 2A(b)) and neurons (ectoderm), specifically dopaminergic neuronal cells with positive expression of Tyrosine hydroxylase (TH) and MAP2 (Fig 2A(c)).

These data confirm that it is possible to generate patient-specific iPSCs with *POLG* mutations by reprogramming of patient fibroblasts.

## POLG iPSCs manifest a partial phenotype

Next, we asked whether iPSCs generated from POLG patients showed changes in mitochondrial structure or function. Since clonal variations were detected between ESCs and other iPSC lines during neuronal differentiation, we compared three homozygous WS5A iPSC clones and two heterozygous CP2A iPSC clones to four clones from two different control iPSC lines to investigate mitochondrial morphology, MMP, and intracellular ATP production. In addition, we quantified the mtDNA using two approaches (i) indirectly using flow cytometry and (ii) PCR quantification. ESCs were excluded for the investigation of the disease-related phenotypes due to the difference between ESC and other iPSCs lines.

Mitochondrial morphology was analyzed by fluorescence microscopy and mitochondrial activity by flow cytometry. First, we double-stained cells with MitoTracker Green (MTG) and tetramethylrhodamine ethyl ester (TMRE) (Agnello *et al*, 2008) and identified a similar appearance of mitochondrial morphology in both patient and control iPSCs under confocal microscopy (Fig 2B).

TMRE, a cationic, lipophilic compound, can be used to measure MMP in live cells as it reversibly accumulates in the highly negatively charged mitochondrial matrix. Thus, membrane potential can be measured dynamically; release of TMRE after mitochondrial depolarization and its reuptake after repolarization can be quantified (Krohn *et al*, 1999). In order to understand the relationship between MMP and the volume of mitochondria present in live cells, we combined MTG and TMRE staining and measured the relative fluorescence intensity of each by flow cytometry. This ratio gives a relative measure of MMP independent of mitochondrial mass that we call specific MMP. To establish this assay, we first had to evaluate the relationship between MTG and MMP fluorescence particularly since MTG fluorescence has been reported to be both independent of (Doherty & Perl, 2017) and sensitive to MMP (Pendergrass *et al*, 2004). We titrated different concentrations of TMRE against 150 nM MTG in control iPSCs. We saw no significant difference in MTG fluorescence in the TMRE concentration range of 5–100 nM (Appendix Fig S2). Higher TMRE concentrations (over 100 nM) caused a decrease in MTG fluorescence (Appendix Fig S2). Thus, we chose 100 nM TMRE and 150 nM MTG to estimate the specific MMP. Mitochondrial volume measured by MTG (Fig 2C), total MMP measured by TMRE alone

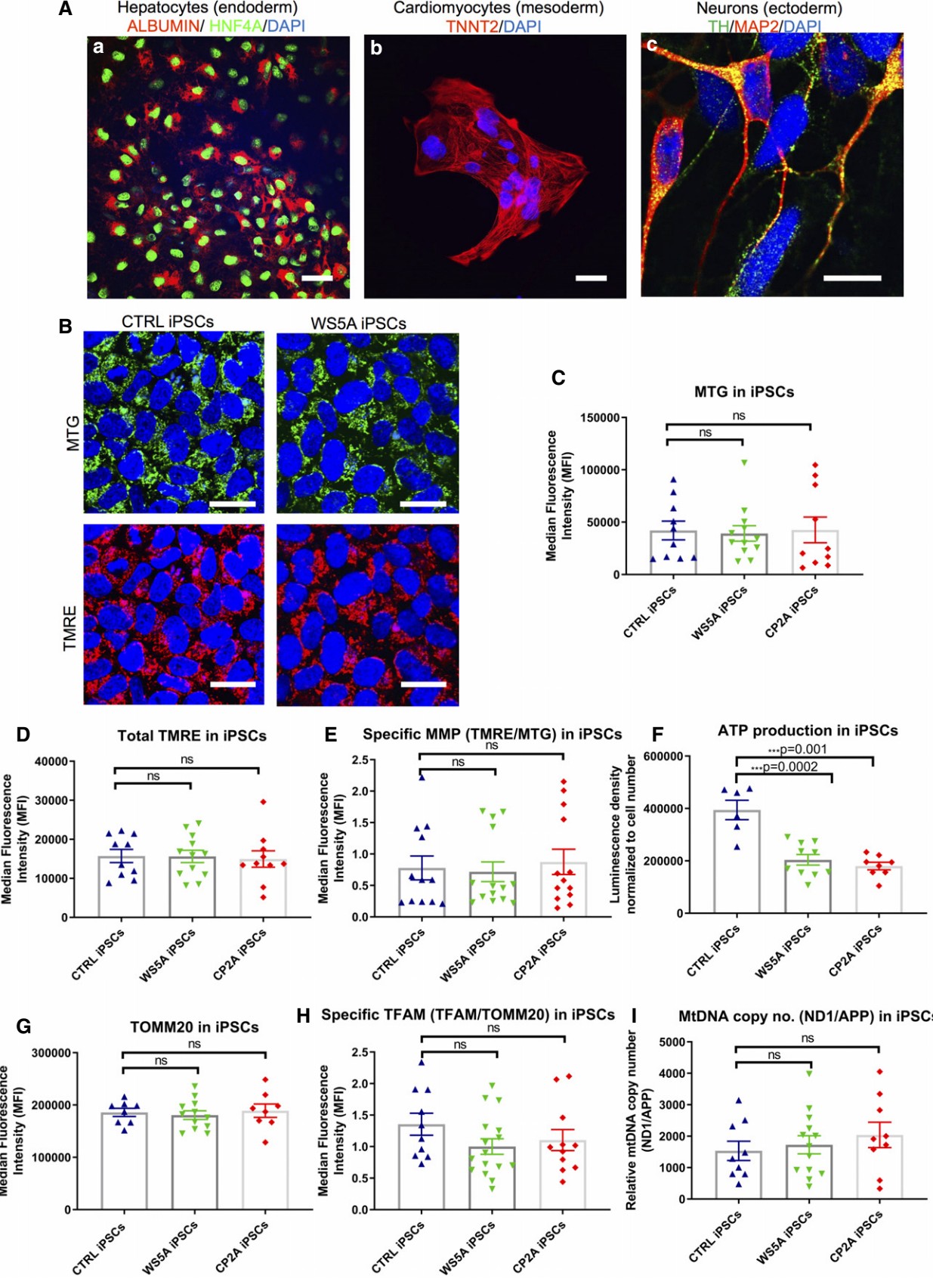

**Figure 2.**

**Figure 2. POLG iPSCs manifested a partial phenotype presenting with energy depletion.**

A   Representative confocal images of iPSC lineage-specific differentiation into germ layers of endoderm-derived hepatocytes with positive expression of ALBUMIN (red) and HNF4A (green) (a) (scale bar, 100 μm), mesodermal-derived cardiomyocytes with positive expression of TNNT2 (red) (b) (scale bar, 100 μm), and ectodermal-derived dopaminergic neurons with positive expression of TH (green) and MAP2 (red) (c) (scale bar, 10 μm). Nuclei are stained with DAPI (blue).

B   Confocal images of mitochondrial morphology for iPSC lines with co-staining of MTG (upper panel) and TMRE (lower panel) (scale bars, 25 μm). Nuclei are stained with DAPI (blue).

C–E   Flow cytometric analysis of iPSCs generated from Detroit 551, WS5A, and CP2A fibroblasts for mitochondrial volume (MTG) (C, $n = 6$, technical replicates per clone for control and CP2A; $n = 5$, technical replicates per clone for WS5A), total MMP (TMRE) (D, $n = 6$, technical replicates per clone for control and CP2A; $n = 5$, technical replicates per clone for WS5A) and specific MMP (E, $n = 6$, technical replicates per clone for control and CP2A; $n = 5$, technical replicates per clone for WS5A) calculated by dividing median fluorescence intensity (MFI) for total TMRE expression by MTG.

F   Intracellular ATP production in iPSCs generated from Detroit 551, WS5A, and CP2A fibroblasts ($n = 3$, technical replicates per clone for control and WS5A; $n = 4$, technical replicates per clone for CP2A).

G, H   Flow cytometric analysis of TOMM20 expression level (G, $n = 4$, technical replicates per clone) and specific TFAM level (total TFAM/TOMM20) (H, $n = 4$, technical replicates per clone).

I   Relative mtDNA copy number in Detroit 551, WS5A and CP2A iPSCs by RT-qPCR analysis using ND1 and APP ($n = 5$, technical replicates per clone for control; $n = 5$, technical replicates per clone for WS5A and CP2A). Values are presented as $Log_2$ of the ratio between the expression values of ND1 in relation to APP.

Data information: The data presented in C–I were generated from 2 different iPSC clones from Detroit 551 control, 3 different clones from WS5A patient iPSCs, and 2 different clones from CP2A patient iPSCs. Data are presented as mean ± SEM for the number of samples. Mann–Whitney $U$-test was used for the data presented in C and E. Two-sided Student's $t$-test was used for the data presented in D and F–I. Significance is denoted for $P$ values of less than 0.05. ***$P < 0.001$.
Source data are available online for this figure.

(Fig 2D), and specific MMP calculated by TMRE/MTG using the combined assay (Fig 2E) showed no differences between patient WS5A and CP2A and control iPSCs. We next assessed ATP production per cell using a live cell luminescence assay and found significantly lower ATP levels in both WS5A and CP2A iPSCs compared to control iPSCs (Fig 2F).

We investigated mtDNA copy number using two different approaches: first, with flow cytometry to assess the level of mitochondrial transcription factor A (TFAM), which binds mtDNA in molar quantities and second, using RT-qPCR to quantify the mtDNA copy number. We performed flow cytometric quantification using antibodies against TFAM and mitochondrial import receptor subunit TOM20 (TOMM20), a mitochondrial outer membrane protein, in order to correlate this to mitochondrial mass. We found no difference in TOMM20 expression (Fig 2G) and TFAM protein levels corrected for mitochondrial content in patients versus control (Fig 2H). Lastly, quantification of mtDNA copy number was performed by RT-qPCR using the relative method. Mitochondrial and nuclear DNA quantities were measured amplifying genomic regions of ND1 and APP genes, respectively. The ratio of mtDNA/nDNA (ND1/APP) was calculated and showed no difference between iPSC clones in WS5A and CP2A patient and control (Fig 2I).

Our findings reveal an energy failure with loss of ATP production in POLG-specific iPSCs, but no change in mitochondrial volume, membrane potential, or mtDNA level. This suggests that the pluripotent stage is not suitable for modeling mitochondrial disease caused by *POLG* mutation.

## Neural induction and derivation of neural stem cells carrying *POLG* mutations

Primary NSCs serve as a source of different types of neurons and glial cells *in vivo* and play a vital role in development, repair, and the functional modulation of human adult neurogenesis. In addition, NSC dysfunction may contribute to an assortment of neurological disorders (Li *et al*, 2018). We therefore explored whether NSCs derived from these two patients showed defective mitochondrial function.

We employed a modified dual SMAD protocol (Fig 3A) to differentiate NSCs from both control and patient iPSCs. A three-stage induction protocol (Fig 3A and B) was applied. Briefly, we utilized the iPSCs and ESCs with high quality and showing no sign of differentiation to initiate the neural induction. After 5 days neural induction using a chemically defined medium (CDM) supplemented with SB431542, AMPK inhibitor Compund C and N-acetylcysteine (NAC), iPSCs (Fig 3B(a)) progressed to a neural epithelial stage exhibiting clear neural rosette structures (Fig 3B(b)). Neurospheres were generated by lifting neural epithelium at day 5, and these were round, well-defined spheres of small to medium size in suspension culture (Fig 3B(c)). NSCs were then produced by dissociating neurospheres into single cells and then replating in monolayers. NSCs in monolayers showed a clear neural progenitor appearance (Fig 3B (d)).

We confirmed the specific stages of neural induction with immunostaining: iPSCs showed SSEA4 and POU5F1 expression (Fig 3B(e)); neural epithelial cells showed rosette structures that uniformly expressed PAX6 and NESTIN (Fig 3B(f)); neurospheres showed positive NESTIN expression (Fig 3B(g)); and NSCs stained positively for PAX6 (Fig 3B(h)). Using flow cytometric quantification, we demonstrated that POU5F1 expression decreased during the induction process, while the neural progenitor markers NESTIN and PAX6 increased during neural induction over the 5 days (Fig 3C). To minimize intraclonal variation, we included two different ESC lines, four independent clones from Detroit 551 fibroblasts, one clone from CCD-1079Sk, one clone from control AG05836, three independent clones from WS5A, and two independent clones generated from CP2A fibroblasts. Each line was subjected to neural induction, on three independent occasions, generating a total of 33 NSC lines which were used in further investigations.

Next, we used immunostaining and flow cytometry to investigate relevant protein markers and RT-qPCR to evaluate gene expression to characterize the NSC lines. Immunostaining demonstrated that iPSC-derived NSCs were positive for NESTIN and PAX6 (Fig 3D). RT-qPCR analysis for neural progenitor markers *SOX2, NESTIN,* and *PAX6* revealed similar patterns of gene expression for all NSCs

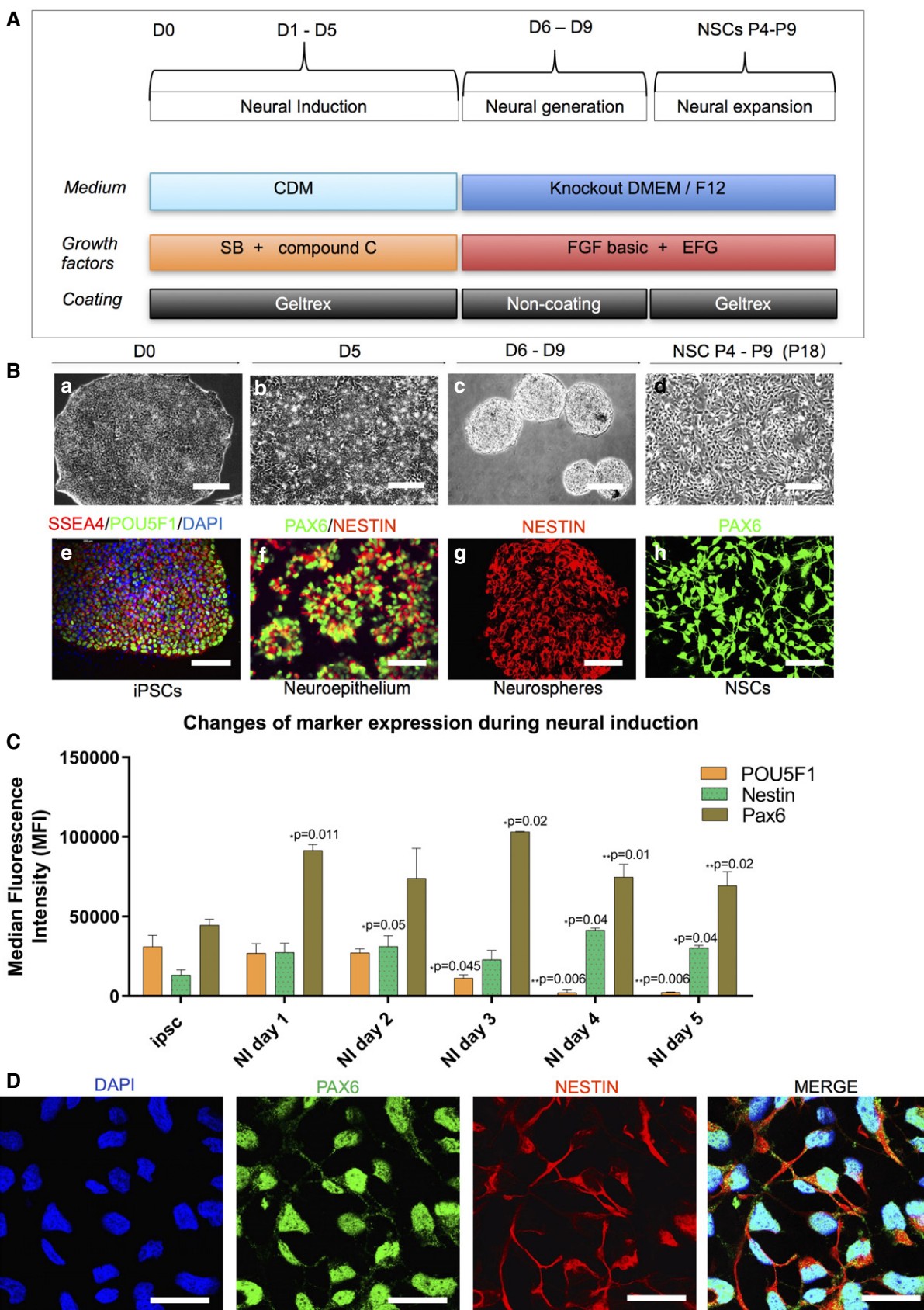

**Figure 3.**

**Figure 3. Validation of patient-specific NSCs.**

A  Schematic of dual differentiation and seeding paradigm for NSCs via the dual SMAD protocol.
B  Representative phase contrast images (upper panel) and immunostaining for specific stages during neural induction from iPSCs to NSCs. Upper panel displays the morphology in culture of different cell types during neural induction to NSCs from iPSCs including iPSCs (a); neuroepithelium with rosette-like structures (b); neurospheres with defined round shapes in suspension culture (c); and NSCs in monolayers (d) (scale bars, 50 μm). Lower panel demonstrates the representative phase contrast images for the immunostaining corresponding to the specific stages in the upper panel: iPSCs with positive staining of SSEA4 (red) and POU5F1 (green) (e) (scale bar, 50 μm); neuroepithelium with rosette-like structures with positive staining of PAX6 (green) and NESTIN (red) (f) (scale bar, 50 μm); neurospheres with positive staining of NESTIN (red) (g) (scale bar, 100 μm); NSCs with positive staining of PAX6 (green) (h) (scale bar, 50 μm).
C  Flow cytometric assessment of stemness marker POU5F1 and NSC markers NESTIN and PAX6 during neural induction from day 0 to day 5 (*n* = 3, technical replicates).
D  Representative images of the immunofluorescent labeling for NSC markers SOX2 (green) and NESTIN (red) (scale bar, 25 μm) from Detroit 551 control iPSC-derived NSCs. Nuclei are stained with DAPI (blue).

Data information: The data presented in C were generated from one iPSC clones (clone 1) from Detroit 551 control. Data are presented as mean ± SEM for the number of samples. Mann–Whitney *U*-test was used to test the significant difference in the expressions of days 1–5 compared to day 0. Significance is denoted for *P* values of less than 0.05. **P* < 0.05; ***P* < 0.01.

Source data are available online for this figure.

(Fig 4A). Since quantification using immunostaining is challenging, we performed flow cytometry to analyze the purity of our NSC cell populations. All NSCs showed positive protein expression of NSC markers PAX6 (Fig 4B) and NESTIN (Fig 4C), with over 94% of the population positive for NESTIN (Appendix Fig S3). In addition, our data showed that the protein expression level for PAX6 (Fig EV3A) exhibited a peak on passage 7 and dropped afterward, and the pluripotency marker POU5F1 (Fig EV3B) remained unchanged during long-term cultivation. Thus, all the NSCs used for further investigation were limited to passages 4–9.

We confirmed that the NSCs generated from patients maintained the same *POLG* gene profile of the parental cells by demonstrating that they retained the same mutation as the original fibroblasts (Fig 4D). Thereafter, we showed that these NSCs had the potential to differentiate into glial subtypes and neurons, including astrocytes which expressed glial fibrillary acidic protein (GFAP, Fig 4E(a)), S100 calcium-binding protein β (S100β) (Appendix Fig S4), excitatory amino acid transporter 1 (EAAT1), and glutamine synthetase (Appendix Fig S5); oligodendrocytes via expression of galactocerebroside (GALC, Fig 4E(b)), oligodendrocyte transcription factor (OLIG2) and myelin basic protein (MBP) (Appendix Fig S6), and dopaminergic (DA) neurons which were positive for TH and tubulin beta III (TUJ1) (Fig 4E(c)).

These findings indicate that it is possible to generate patient-specific, POLG-mutated NSCs with reasonable purity and efficiency. We also confirm that our iPSC-derived NSCs faithfully retained the original genotype (Ma *et al*, 2015; Kang *et al*, 2016) and carried the sequencing with the relevant *POLG* exons.

### POLG NSCs showed a greater impairment of mitochondrial function compared to iPSCs and parental fibroblasts

We applied the same experimental approaches as described above to investigate mitochondrial function in NSCs. We compared a panel of different NSCs generated from three homozygous WS5A iPSC clones and two heterozygous CP2A iPSC clones to three clones from Detroit 551 iPSCs, one clone from CRL2097, and one clone from AG05836 control, to investigate mitochondrial morphology, MMP, and intracellular ATP production. In addition, we assessed the original patient fibroblast lines to determine whether a phenotype was present.

We assessed mitochondrial morphology by transmission electron microscopy and found similar morphology in both patient and control NSCs, displaying normal and well-developed cristae (Fig 5A).

Using the TMRE/MTG double staining approach described above, we found that mitochondrial volume was less in WS5A, but similar in CP2A NSCs compared to control lines (Fig 5B). While the total MMP (Fig 5C) appeared decreased in mutant NSCs as compared to control, no significant difference was detected in the specific MMP level, i.e. MMP per mitochondrial volume (total TMRE/MTG) (Fig 5D). In fibroblasts, similar mitochondrial volumes were found in mutant and control cells based on MTG level (Fig 5F). However, a significantly increased total (Fig 5G) and specific MMP (Fig 5H) was seen in fibroblasts from both WS5A and CP2A mutants versus control lines.

Direct measurement of intercellular ATP production revealed a significant defect in both WS5A and CP2A NSCs compared to control (Fig 5E). Interestingly, this was also seen in the parental fibroblasts (Fig 5I) and iPSCs (Fig 2F). Fibroblasts, however, appeared capable of maintaining and indeed increasing their membrane potential (Lorenz *et al*, 2017), possibly by hyperpolarization, while NSCs did not.

Our data indicate a cell lineage/tissue specificity in response to *POLG* mutation. Energy depletion is seen in both NSCs and fibroblasts, but the effect on MMP differed. This differential response suggests that neurons are selectively vulnerable to the mitochondrial changes induced by *POLG* mutation.

### POLG NSCs manifest mtDNA depletion, but not mtDNA deletion

We assessed mtDNA both qualitatively and quantitatively. As previously, we used both flow cytometric analysis of TFAM and qPCR. We detected a significant decrease of specific TFAM protein level (TFAM/TOMM20) in both WS5A and CP2A NSCs compared to control (Fig 6B), whereas no changes were observed at TOMM20 expression level (Fig 6A). Quantification based on RT-qPCR measurements of the ND1/APP ratio also revealed a significant reduction of mtDNA copy number in patient WS5A and CP2A NSCs (Fig 6C). In contrast, TOMM20 (Fig 6D) and TFAM levels (Fig 6E), as well as mtDNA copy number (Fig 6F), were not significantly changed in patient fibroblasts. These findings further demonstrate the differential response between patient NSCs (Fig 6A–C), iPSCs (Fig 2G–I), and parental fibroblasts (Fig 6D–F). The fact that mtDNA depletion arises in neural lineage indicates that NSCs are a

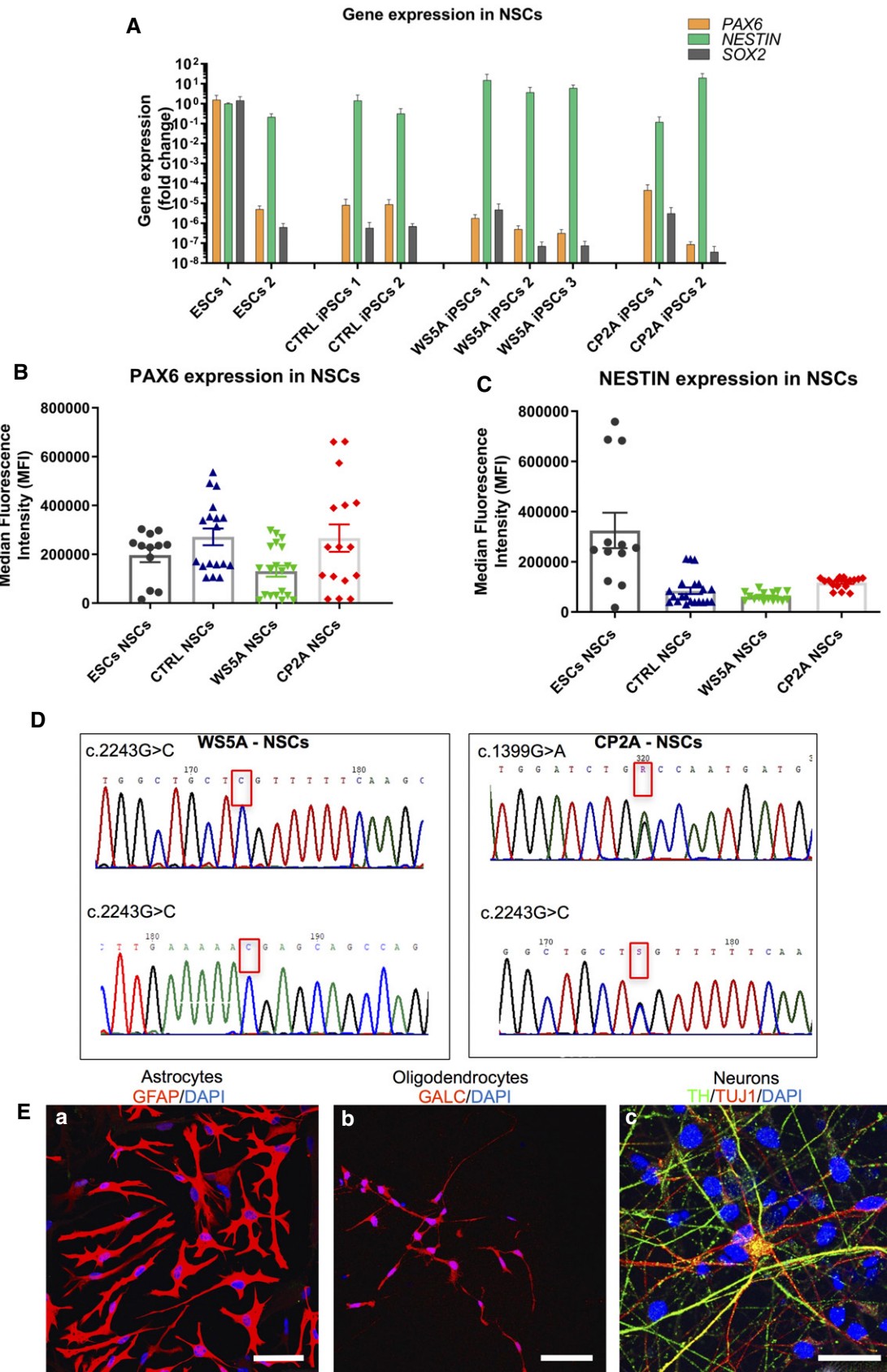

**Figure 4.**

**Figure 4. NSCs retained the original genotype and sequence.**

A    Quantification of gene expression for NSC markers *PAX6, NESTIN,* and *SOX2* for all NSCs from RT-qPCR analysis. The expression of the neural stem cell markers is assessed with fold change using the comparative $\Delta\Delta C_t$ method by normalizing NSCs to iPSCs ($n = 3$, technical replicates per ESC line or iPSC clone).

B, C   Quantification of protein expression level for PAX6 (B, $n = 6$, technical replicates per line for ESCs; $n = 7$, technical replicates per clone for control; $n = 8$, technical replicates per clone for WS5A; $n = 9$, technical replicates per clone for CP2A) and NESTIN (C, $n = 6$, technical replicates per line for ESCs; $n = 7$, technical replicates per clone for control; $n = 8$, technical replicates per clone for WS5A; $n = 9$, technical replicates per clone for CP2A) for iPSC-derived NSCs using flow cytometry.

D    Sequencing chromatogram showing the homozygous c.2243G>C variation in *POLG* in WS5A iPSC-derived NSCs and the heterozygous c.1399G>A and c.2243G>C variation in *POLG* in CP2A iPSC-derived NSCs.

E    Representative confocal images showing glial and neuronal lineages derived from NSCs. (a) Immunostaining of NSC-derived astrocytes with GFAP (red) staining (scale bar, 50 μm). (b) Immunostaining of oligodendrocytes showing GALC (red)-positive labeling (scale bar, 50 μm). (c) Dopaminergic neurons showing TH (green) and TUJ1 (red)-positive staining (scale bar, 25 μm). Nuclei are stained with DAPI (blue).

Data information: The data points in A–C represent NSCs generated from 2 ESC lines, 3 different control iPSCs including 2 different clones from Detroit 551 control, one clone from control AG05836, 3 different clones from WS5A patient, and 2 different clones from CP2A patient iPSCs. Data are presented as mean ± SEM for the number of samples.

Source data are available online for this figure.

relevant model system for mtDNA-associated disease with *POLG* mutation.

Since we previously identified mtDNA deletions in post-mortem neurons from patients (Tzoulis *et al*, 2014), we used long range PCR to investigate whether qualitative mtDNA damage also occurred in NSCs. Deletions were not detectable in the mtDNA of patient NSCs and nor were they found in the parental fibroblasts (Fig 6G). Thus, the accumulation of mtDNA deletions appears time dependent, both in aging tissues (Kazachkova *et al*, 2013) and in neurons of POLG patients (Tzoulis *et al*, 2014).

These findings indicate that NSCs but not fibroblasts or iPSC faithfully replicate the mtDNA depletion found in post-mortem neurons. Deletions of mtDNA, which only accumulate slowly over time, were not found.

## POLG dopaminergic (DA) neurons confirm mtDNA findings in NSCs

As DA neurons of the substantia nigra were one of the most affected cell types in POLG disease (Tzoulis *et al*, 2014), we generated this specific neuronal type to investigate whether we could replicate the pathological findings seen in NSCs.

We applied DA neuron differentiation based on the dual SMAD method with the addition of FGF-8 and Sonic hedgehog (SHH) agonist purmorphamine (PM), modified from a previous report (Stacpoole *et al*, 2011). This procedure included, Phase I: the first stage of differentiation from stem cells to neural rosette-like colonies, and the second stage of differentiation from neural rosette-like colonies to neurospheres by dual inhibition of SMAD signaling with SB431542 and dorsomorphin/Compound C. Phase II and III: caudalization of neurospheres with addition of FGF-8 and further ventralization to dopaminergic progenitors using the combination of FGF-8 and the SHH agonist PM. Phase IV: DA neuron maturation from DA progenitors by adding BDNF and GDNF. Using this protocol, both control and POLG mutant iPSCs efficiently differentiated into mature DA neurons as shown by both TH and MAP2 expression (Fig 6H), and the yield of TH-positive neurons was over 97% in both control and POLG cultures (Fig 6I).

We then investigated alterations of TFAM level and mtDNA copy number in both WS5A and CP2A DA neurons. We found that mutant DA neurons displayed similar defects to those seen in NSCs

(Fig 6B and C), with both lower TFAM level (Fig 6J) and mtDNA copy number (Fig 6K) than control DA neurons.

Our findings confirm that different cell types respond differently to the same bioenergetic challenge. However, considering neurons are difficult to acquire and display limited expansion potential, it would be preferential to use NSCs as they are feasible to expand, bank, and can potentially enable high-throughput drug screening. Since we found that NSCs manifest POLG disease phenotypes, NSCs were considered as a precise and reliable cell type and applied for further studies of disease mechanisms.

## Respiratory chain complex I loss, NAD⁺/NADH redox changes and ROS excess occur in POLG NSCs, but not iPSCs

We showed previously that respiratory chain complex I was lost in POLG affected frontal and cerebellar neurons (Tzoulis *et al*, 2014), and we confirmed these findings in this current study. We studied occipital neurons collected from fresh frozen brain tissue taken from patients with *POLG* mutations ($n = 5$) and controls ($n = 5$). The *POLG* mutations were as follows: A467T/G303R (AL-1B); A467T/A467T (AT-1A); A467T/A467T (AT-2A); W748S/W748S (WS-10A), and W748S/W748S (WS-3A). In agreement with our previous study, immunohistochemistry demonstrated clear complex I deficiency in occipital neurons of POLG patients (Fig 7A(a–c)). Further, relative quantification of mtDNA from microdissected occipital neurons showed significantly decreased mtDNA copy number in the patient neurons compared to controls (Fig 7A(d)).

We then investigated whether this also occurred in POLG cells. To quantify complex I level, we used Anti-NDUFB10, the same antibody as was used in post-mortem studies. Co-staining with TOMM20 was used to correlate complex level to mitochondrial mass. We found a similar level of complex I in both patient and control iPSCs (Fig 7F). Levels of complex II (Fig 7G) and IV (Fig 7H) were also similar in iPSCs. In NSCs, however, we observed a significant decrease in mitochondrial complex I assessed by both immunostaining (Fig 7B) and flow cytometry (Fig 7C). This was not seen in either iPSCs (Fig 7F) or fibroblasts (Fig 7I). Although no significant changes in mitochondrial complex II (Fig 7D) or IV subunits (Fig 7E) were found in mutant NSCs or fibroblasts (Fig 7J and K) as compared to control lines, lower complex IV subunit was found in WS5A NSCs compared to control (Fig 7E). These data

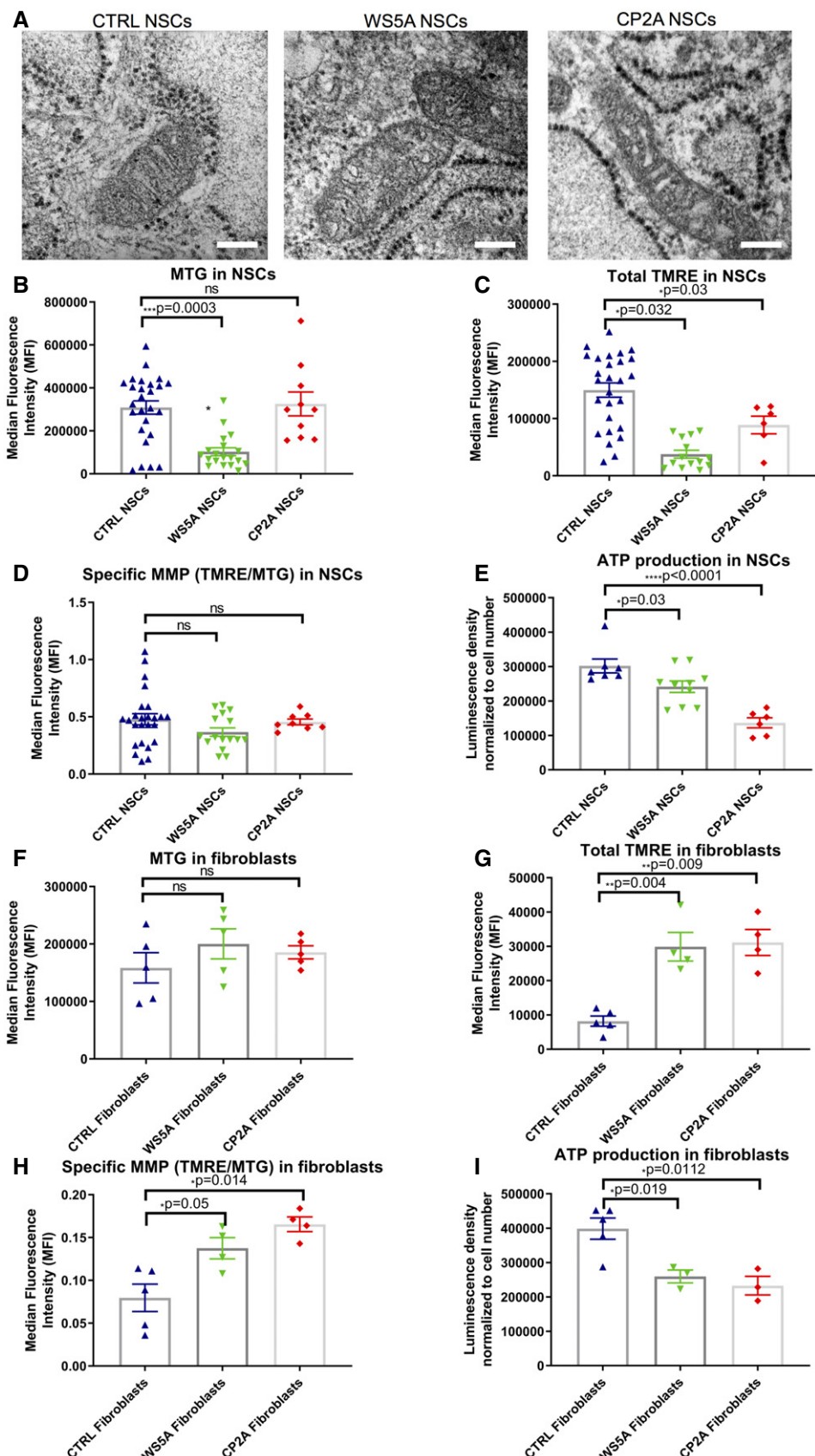

**Figure 5.**

**Figure 5. POLG NSCs exhibited impaired mitochondrial function, which was partly presented in fibroblasts.**

A Representative transmission electron microscopy images of mitochondrial structures in Detroit 551 control, WS5A, and CP2A NSCs (scale bar, 400 nm).

B–D Flow cytometric analysis of mitochondrial volume (MTG) (B, $n = 5$, technical replicates per clone), total MMP (TMRE) (C, $n = 5$, technical replicates per clone), and specific MMP (D, $n = 5$, technical replicates per clone), calculated by dividing median fluorescence intensity (MFI) for total TMRE expression by MTG in NSCs generated from control lines, WS5A, and CP2A iPSCs.

E Intracellular ATP production in NSCs ($n = 3$, technical replicates per clone).

F–H Flow cytometric analysis of mitochondrial volume (MTG) (F, $n = 6$, technical replicates for control; $n = 5$, technical replicates for WS5A and CP2A), total MMP (TMRE) (G, $n = 5$, technical replicates for control; $n = 4$, technical replicates for WS5A and CP2A), and specific MMP (TMRE/MTG) (H, $n = 5$, technical replicates for control; $n = 4$, technical replicates for WS5A and CP2A) in Detroit 551, WS5A, and CP2A fibroblasts.

I Intracellular ATP production in Detroit 551, WS5A, and CP2A fibroblasts ($n = 6$, technical replicates for control; $n = 3$, technical replicates for WS5A and CP2A).

Data information: The data points in B–D represent NSCs generated from 5 different control iPSCs including 3 different clones from Detroit 551 control, one clone from CCD-1079Sk, one clone from control AG05836, 3 different clones from WS5A patient iPSCs, and 2 different clones from CP2A patient iPSCs. The data points in E represent NSCs derived 3 different clones from Detroit 551, 3 different clones from WS5A, and 2 different clones from CP2A iPSCs. Data are presented as mean ± SEM for the number of samples. Mann–Whitney $U$-test was used for the data presented in B and C. Two-sided Student's $t$-test was used for the data presented in D–I. Significance is denoted for $P$ values of less than 0.05. *$P < 0.05$; **$P < 0.01$; ***$P < 0.001$; ****$P < 0.0001$.

Source data are available online for this figure.

---

indicate that, similar to both POLG patient tissues, complex I defect is present in NSCs.

Complex I function is essential for the re-oxidization of NADH and maintenance of the NAD$^+$/NADH ratio, which is an important indicator of redox status and major modulator of intermediary metabolism (Houtkooper et al, 2010). To investigate the effect of POLG mutation on NAD$^+$ metabolism, we measured both NAD$^+$ and NADH levels using liquid chromatography–mass spectrometry (LC-MS). As expected, we found a significant decrease in the NAD$^+$/NADH ratio in both patient NSCs compared with control (Fig 8A). While the levels of NAD$^+$ and NADH varied in the patient NSCs (Fig 8B and C), the ratio showed disturbed NAD$^+$ homeostasis in both. To confirm these findings, we performed colorimetric analysis using a commercial NAD$^+$/NADH assay. Similar to the findings from LC-MS, we found a significant decrease of NAD$^+$/NADH ratio in both patient NSCs versus control (Appendix Fig S8). However, we found an increase in the NAD$^+$/NADH ratio (Fig 8D) and reduced level of NAD$^+$ (Fig 8E) and NADH (Fig 8F) in both patient iPSCs compared with control, though only the CP2A line reached significance (Fig 8D–F).

The mitochondrial respiratory chain is considered to be a major source of intracellular ROS, and the dysfunction of complex I is particularly implicated in excess ROS production (Hayashi & Cortopassi, 2015), which potentially can damage both mitochondria and mtDNA (Cui et al, 2012). To investigate whether the presence of POLG mutation led to increased ROS production, we used dual staining with 2′,7′-dichlorodihydrofluorescein diacetate (DCFDA) and MitoTracker Deep Red (MTDR). We found no major differences in ROS production (DCFDA) in patient WS5A and CP2A iPSCs compared to controls (Fig EV4A). To ensure that this reflected the mitochondrial mass in each cell type, we divided total ROS by a measure of mitochondrial mass (MTDR) to give specific ROS and again found no difference between patient and control iPSCs (Fig EV4B). Interestingly, a trend toward reduced total (Fig EV4C) and specific ROS level (Fig EV4D) was detected in patient fibroblasts. In contrast, both total ROS (Fig EV4E) and specific ROS (Fig 8G), i.e., the ratio of ROS to mitochondrial volume defined by MTDR, appeared significantly higher in mutant NSCs compared to control, indicating the potential for oxidative damage in mutant NSCs. To confirm that the increased ROS was of mitochondrial origin, we used the ROS-sensitive fluorescent dye MitoSox and

quantified both total and the ratio of ROS to mitochondrial volume defined by MTG, i.e., specific ROS. Both patient NSCs showed a significant increase in the mean intensity of MitoSox fluorescence at specific level (Fig 8H), suggesting the high ROS production generated from mitochondria itself. However, the total level of mitochondrial ROS was lower in WS5A patient but higher in CP2A patient compared to control (Fig EV4F).

These data suggest that complex I loss has functional consequences. The disturbance of NAD$^+$ homeostasis will have major implications for neurons that are heavily dependent on glucose metabolism and thus reoxidation of NADH generated by the citric acid cycle. Increased ROS production leads potentially to a more indiscriminate damage. Both consequences are likely to contribute to the pathological mechanism of POLG-related diseases.

## UCP2/SirT1-regulated cellular senescence and BNIP3 pathway-mediated mitophagy are involved in the pathogenesis of neuronal POLG diseases

In view of the changes in NAD$^+$ metabolism and the known link between mitochondrial dysfunction with increased ROS production and aging (Fang et al, 2019), we investigated cellular senescence and mitophagy in our NSCs. We performed staining of β-galactosidase and flow cytometric measurements of cellular senescence marker p16INK4. We observed significantly increased activity for β-galactosidase in both patient WS5A and CP2A NSCs compared to control NSCs (Fig 8I and J). This finding was further corroborated with flow cytometry against p16INK4 (Fig EV5).

Next, we investigated the molecular pathways linked to initiating the process of senescence. Previous studies have shown that overproduction of mitochondrial uncoupler protein 2 (UCP2) irreversibly reduces the MMP and induced a senescent-like morphology, leading to irreversible metabolic changes, loss of cellular ATP, and high ROS (Nishio & Ma, 2016). Sirtuin 1 (SirT1) is the most conserved mammalian NAD$^+$-dependent protein deacetylase and is increasingly recognized as a crucial regulator of a variety of cellular processes, including energy metabolism, stress response, and senescence (Li, 2013). In addition, SirT1 has been shown to repress the UCP2 gene by binding directly to the UCP2 promoter (Bordone et al, 2006). To investigate a possible pathogenic role for UCP2 and SirT1 in POLG-induced senescence, we examined the expression of UCP2

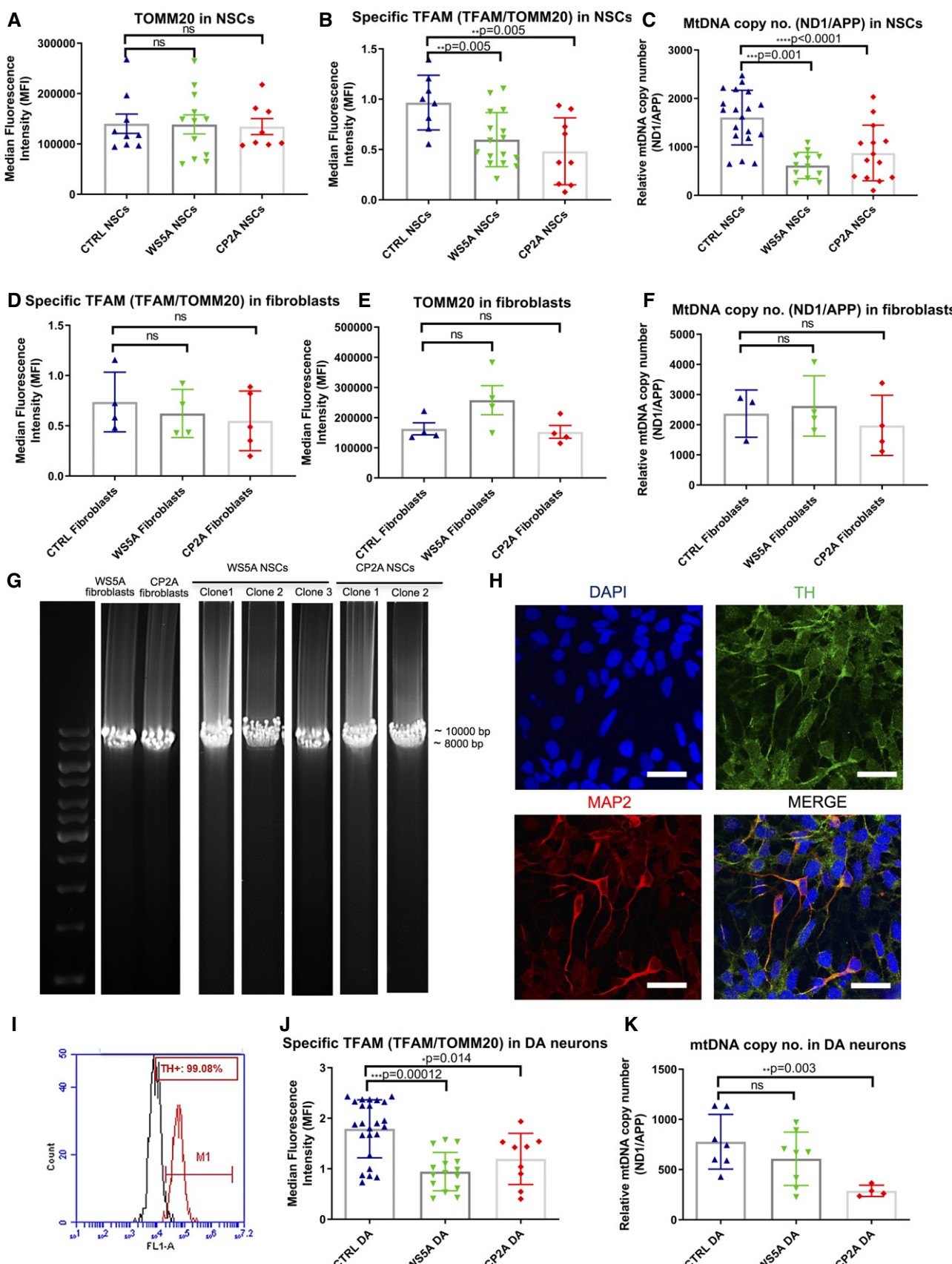

**Figure 6.**

◄

**Figure 6.  POLG NSCs exhibited decreased TFAM level and mtDNA depletion, but not mtDNA deletion, consistent with iPSC-derived DA neurons.**

A, B    Flow cytometric analysis of TOMM20 (A, $n = 3$, technical replicates per clone for control; $n = 4$, technical replicates per clone for WS5A and CP2A) and TFAM protein expression (B, $n = 3$, technical replicates per clone for control and CP2A; $n = 5$, technical replicates per clone for WS5A) in NSCs. Expressed as specific TFAM level (total TFAM/TOMM20).

C       Relative mtDNA copy number analyzed by RT–qPCR using primers and fluorogenic probes for regions of ND1 and nuclear gene APP in all NSCs ($n = 4$, technical replicates per clone for control; $n = 5$, technical replicates per clone for WS5A; $n = 7$, technical replicates per clone for CP2A).

D, E    Flow cytometric analysis of TOMM20 expression level (D, $n = 4$, technical replicates per clone) and TFAM protein expression (E, $n = 4$, technical replicates for control and WS5A; $n = 5$, technical replicates for CP2A) in fibroblasts. Expressed as specific TFAM level (total TFAM/TOMM20).

F       Relative mtDNA copy number analyzed by RT–qPCR for regions of ND1 and nuclear gene APP in fibroblasts ($n = 3$, technical replicates for control; $n = 4$, technical replicates for WS5A and CP2A).

G       Long-PCR for detection of mtDNA deletions in NSCs and their parental fibroblasts from WS5A and CP2A patients.

H       Representative images of confocal microscopy with immunofluorescence labeling of DA neuron-specific marker TH (green) and neuron-specific marker MAP2 (red) for iPSC-derived DA neurons (scale bars, 25 μm). Nuclei are stained with DAPI (blue).

I       Histogram of the positive cell population stained with DA marker TH in iPSC-derived DA neurons using flow cytometric analysis.

J       Flow cytometric measurement of specific TFAM level (total TFAM/TOMM20) in iPSC-derived DA neurons from Detroit 551 control, WS5A, and CP2A POLG patients ($n = 8$, technical replicates per clone for control; $n = 5$, technical replicates per clone for WS5A; $n = 4$, technical replicates per clone for CP2A).

K       Relative mtDNA copy number analyzed by RT–qPCR for regions of ND1 and nuclear gene APP in all iPSC-derived DA neurons ($n = 3$, technical replicates per clone).

Data information: The data points in A, B represent NSCs generated from 3 different control iPSCs including 2 different clones from Detroit 551 control, 1 clone from control AG05836, 3 clones from WS5A patient iPSCs, and 2 clones from CP2A patient iPSCs. Data points in C represent NSCs from 5 different controls, including 3 clones from Detroit 551 control, 1 clone from control AG05836, 1 clone from control CCD-1079Sk, 3 clones from WS5A patient iPSCs, and 2 clones from CP2A patient iPSCs. The data points in J and K represent DA neurons generated from 3 clones from Detroit 551 control, 3 clones from WS5A patient iPSCs, and 2 clones from CP2A patient iPSCs. Data are presented as mean ± SEM for the number of samples. Mann–Whitney $U$-test was used for the data presented in A and J. Two-sided Student's $t$-test was used for the data presented in B–F and K. Significance is denoted for $P$ values of less than 0.05. *$P < 0.05$; **$P < 0.01$; ***$P < 0.001$; ****$P < 0.0001$. Source data are available online for this figure.

and the phosphorylated phospho-SirT1 (Ser47) by western blot. We found upregulation of UCP2 expression and reduced phosphorylated SirT1 in both WS5A and CP2A NSCs compared with control (Fig 8K and L(a and c)). These data suggested that decreased SirT1 activity and increased UCP2 expression might be involved in cellular senescence observed in *POLG* mutation.

Mitophagy/autophagy appeared able to both promote and inhibit senescence and senescence-associated phenotypes (Korolchuk *et al*, 2017). To investigate whether an activation of mitophagy was involved in POLG-related disorders, we studied the mitophagy-related proteins PINK1, Parkin, BNIP3, and LC3B using western blotting. Immunoblotting analysis showed an increased level of the autophagosome marker microtubule-associated protein 1 light chain 3β (LC3B-II) with increased ratio of LCBII/LC3BI in CP2A NSCs indicating autophagy activation (Fig 8K and L(b)). PINK1 and Parkin expression were similar to the control group (Fig 8K and L(d and e)), while BNIP3 accumulation was also observed in CP2A, but not WS5A (Fig 8K and L(f)). Our data showed that *POLG* mutation increase cellular senescence and is associated with increased mitophagy.

Our data suggests that *POLG* mutation can induce cellular senescence and that this may be modulated via upregulation of UCP2. We also find that mitophagy is an active component of neuronal POLG pathogenesis, particularly in the compound heterozygote model.

## Discussion

Despite the presence of mitochondria in most cells, mitochondrial disorders manifest strikingly different tissue involvement (Chinnery, 2015). POLG-related disorders, for example, show major involvement of the nervous system and liver. Modeling brain diseases have been hampered by the limited availability of human tissue and lack of faithful disease models. These present major obstacles to our understanding of disease mechanisms and the development of effective treatments. Reprogramming of patient cells combined with their differentiation to the affected cell type has revolutionized the field (Kanherkar *et al*, 2014; Tabar & Studer, 2014). Here, we reprogrammed patient fibroblasts carrying the two common *POLG* mutations (c.2243G>C; p.W748S and c.1399G>A; p.A467T) to iPSCs and then differentiated these to NSCs. We found that patient-specific NSCs replicated the findings of mtDNA depletion and complex I deficiency identified in post-mortem tissues. While others have generated iPSCs from patients with *POLG* mutations (Zurita *et al*, 2016; Chumarina *et al*, 2019) or investigated valproate toxicity in iPSC-derived POLG patient hepatocytes (Li *et al*, 2015), we believe our study is the first to confirm that the pathological changes seen in post-mortem studies are faithfully replicated in iPSC-derived NSCs. More importantly, using iPSC-derived NSCs we were able to identify cellular and molecular mechanisms involved in the disease process including the overproduction of ROS associated with suppressed complex I-driven respiration and defective NAD$^+$ metabolism leading to cellular senescence via UCP2 upregulation and SirT1 downregulation pathway and mitophagy activation. These findings provided compelling evidence of the multiple cellular and molecular mechanisms contributing to the decline of neuron pool in POLG-related diseases and open the way for the development of novel diagnostic and therapeutic interventions.

In the present study, we found that iPSCs harboring *POLG* mutations appeared to be able to maintain their MMP, mitochondrial mass, and mtDNA replication at similar levels to control. They also appeared capable of regulating their ROS homeostasis. This suggested either that the mutant phenotypes were rescued at this stage, as was suggested by an earlier study of iPSCs with mtDNA mutations (Ma *et al*, 2015), or that, similar to fibroblasts, these cells derived proportionally more of their energy from glycolysis and maintained ROS levels by lowered respiratory chain activity. However, both mutated iPSCs and fibroblasts showed significantly lower ATP levels. In addition, ATP depletion was associated with

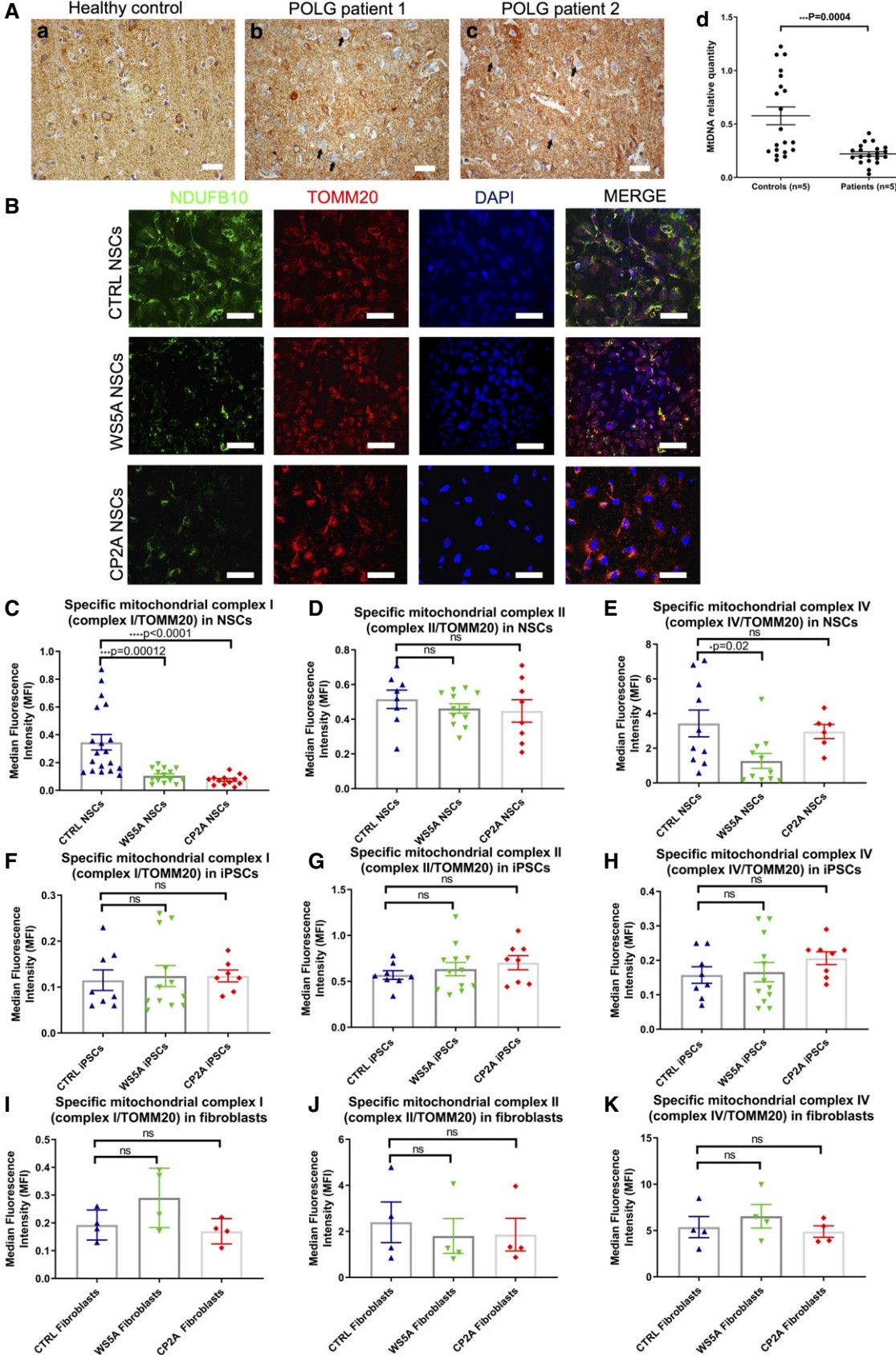

Figure 7.

◄

**Figure 7. POLG mutations induced defects in respiratory chain complex I.**

A   Complex I immunohistochemistry in the occipital cortex of a neurologically healthy control (a) and two patients with POLG disease (b and c) (Scale bar, 20 μm). Patients have numerous complex I-negative neurons (examples marked by arrows). (d) mtDNA relative quantification in microdissected neurons from the occipital cortex of patients with POLG diseases ($n = 5$) and neurologically healthy controls ($n = 5$). Each point represents the mean value of three technical replicates from a pooled sample of 10 neurons. For the purposes of comparison, a control sample has been arbitrarily set to one. The medians of the two groups are compared by Mann–Whitney $U$-test. Data are presented as mean (horizontal bars) $\pm$ SEM (vertical bars).

B   Representative confocal images of immunostaining for mitochondrial complex I subunit NDUFB10 (green) and TOMM20 (red) in control, WS5A, and CP2A NSCs (scale bars, 50 μm). Nuclei are stained with DAPI (blue).

C–E   Flow cytometric measurements of mitochondrial complex I (C, $n = 4$, technical replicates per clone), II (D, $n = 3$, technical replicates per clone for control; $n = 4$, technical replicates per clone for WS5A and CP2A) and IV (E, $n = 3$, technical replicates per clone for control and CP2A; $n = 4$, technical replicates per clone for WS5A) protein level in iPSC-derived NSCs. Expressed as specific complex I, II, and IV level (total complex I, II, IV level/TOMM20).

F–H   Flow cytometric measurements of mitochondrial complex I (F, $n = 4$, technical replicates per clone), II (G, $n = 4$, technical replicates per clone) and IV (H, $n = 4$, technical replicates per clone) protein level in iPSCs. Expressed as specific complex I, II, and IV level.

I–K   Flow cytometric measurements of mitochondrial complex I (I, $n = 4$, technical replicates), II (J, $n = 4$, technical replicates), and IV (K, $n = 4$, technical replicates) protein level in parental fibroblasts expressed as specific complex I, II, and IV level.

Data information: The data points in C represent NSCs generated from 5 different controls, including 3 clones from Detroit 551 control, 1 clone from control AG05836, 1 clone from control CCD-1079Sk, 3 clones from WS5A patient iPSCs, and 2 clones from CP2A patient iPSCs. The data points in D, E represent NSCs generated from 3 clones from Detroit 551 control, 3 clones from WS5A patient iPSCs, and 2 clones from CP2A patient iPSCs. The data points in F–H represent 2 clones from Detroit 551 iPSCs, 3 clones from WS5A patient iPSCs, and 2 clones for CP2A iPSCs. Data are presented as mean $\pm$ SEM for the number of samples. Mann–Whitney $U$-test was used for the data presented in F and J. Two-sided Student's $t$-test was used for the data presented in C–E, G–I and K. Significance is denoted for $P$ values of less than 0.05. *$P < 0.05$; ***$P < 0.001$; ****$P < 0.0001$.

Source data are available online for this figure.

mitochondrial hyperpolarization in fibroblasts. These findings are similar to those observed in another stem cell model of mitochondrial disease (Lorenz et al, 2017) and suggest either that these cells compensated for lower ATP production by reversing the proton flow in the $F_1F_o$ ATPase (Abramov et al, 2010) or the presence of other mechanisms such as downregulating oxygen consumption through complex II (Forkink et al, 2014) are active.

We successfully generated patient-specific NSCs and DA neurons. Combined with neurons being difficult to acquire and display limited expansion potential, NSCs provide a powerful tool to unravel disease mechanism and, potentially, enable high-throughput drug screening. Further, one can perform assays of mitochondrial function in "live cells". From our previous (Tzoulis et al, 2014) and present studies in NSCs, we observed the following potential mechanisms. Confirmation of the loss of complex I in neurons from the occipital cortex and verification of this at the early progenitor stage and in patterned DA neurons. A key finding from our patient studies (Tzoulis et al, 2014) was mtDNA depletion was present in neurons from infants under 1 year of age, and this was also observed in our patient NSCs. Depletion of mtDNA was not, however, present in patient fibroblasts or the derived iPSCs. Interestingly, while depletion was clearly detected by qRT-PCR, we also observed lowered mtDNA levels using an indirect method based on flow cytometric measurement of TFAM. Methodologically, this is of major interest as it suggested that we could use this in live cells as a surrogate measure of mtDNA level. The presence of mtDNA depletion did not appear to impair cell growth; however, unlike the situation in patients, we did not expose our cells to any form of stress. Deletions of mtDNA were not observed in any of the above cell types, suggesting that these were generated over time in post mitotic tissues. The lack of qualitative mtDNA damage supported our earlier conclusion that these changes were cumulative and representative of "accelerated aging".

Since complex I is a major site of ROS production, it has been suggested that loss of this complex is part of a response aimed at reducing the production of these damaging species (Palin et al, 2013). Further, Pryde et al (2016) provided evidence that complex I

degradation was mediated through protease-dependent inactivation. Interestingly, complex I loss was found to affect the whole brain in Parkinson's disease (Flones et al, 2018), not just the substantia nigra (Palin et al, 2013; Flones et al, 2018), supporting the view that loss of this complex was not itself the cause of neuronal loss, but a secondary event. Irrespective of whether neuronal complex I deficiency and elevated ROS are pathological events or a compensatory response, they are clearly important features associated with the POLG disease process in neurons and changes in level of these may be useful for monitoring treatment or other interventions. Here, we found a clear overproduction of ROS. It was attractive to postulate a causal relationship between these findings. The relationship between ROS and cell death and increased ROS and complex I loss remains unclear. Abrupt increases in ROS levels, i.e., above a physiological threshold, may be harmful and may among other events trigger cell death (Davila & Torres-Aleman, 2008). Oxidative stress has been linked to many neurodegenerative diseases, including Alzheimer's (Leuner et al, 2012) and Parkinson's (Schapira, 2008), but its role in mitochondrial disease remains unclear.

Cellular senescence is as a complex stress response by which proliferative cells lose the ability to divide and enter cell cycle arrest. Mounting evidence also showed accumulation of senescent cells with age, a process that may contribute to age-related phenotypes and pathologies (Byun et al, 2015). Dysfunctional mitochondria can induce cellular senescence both in in vitro culture (Moiseeva et al, 2009) and in vivo (Kang et al, 2013; Wiley et al, 2016). Some studies have implicated mitochondrial ROS in this process since ROS are capable of damaging nuclear DNA and thus activating a DNA damage response to induce senescence (Moiseeva et al, 2009). Redox changes, reflected by a lower $NAD^+$/NADH ratio, were also likely to induce a senescence arrest (Lee et al, 2012). We showed that dysfunctional mitochondria due to POLG mutation could drive ROS overproduction and lower the $NAD^+$/NADH ratio, raising the possibility that both were involved in generating the senescence response in POLG NSCs.

A previous study reported that the $NAD^+$-dependent deacetylase SirT1 played an important role in the acceleration of cellular

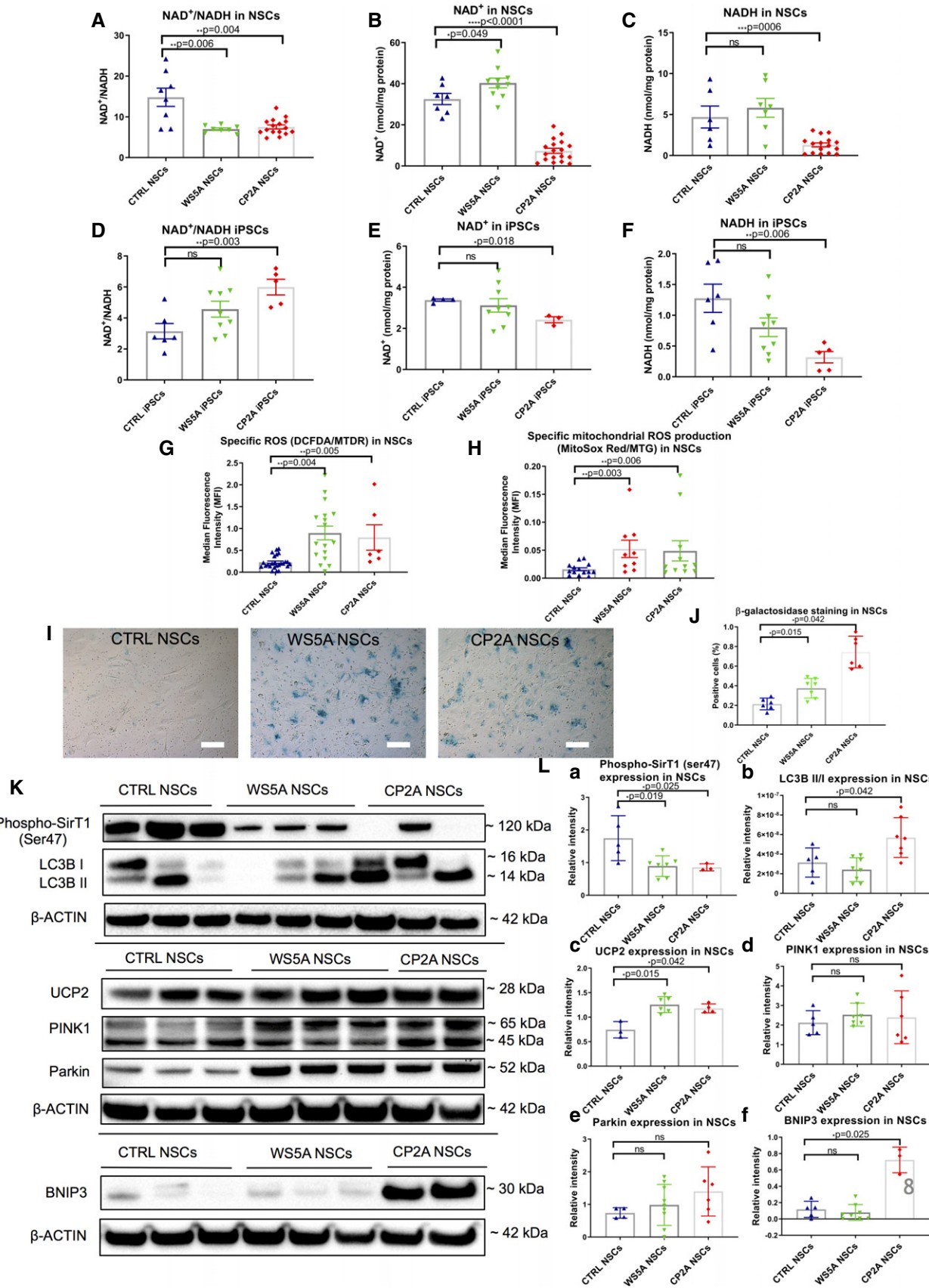

Figure 8.

**Figure 8. NAD+ metabolism, ROS overproduction, exhibition of a senescence phenotype through UCP2/SirT1 and mitophagy activation via BNIP3.**

A–C   LC-MS-based metabolomics for quantitative measurements of NAD+/NADH ratio (A, $n = 3$, technical replicates per clone), NAD+ (B, $n = 3$, technical replicates per clone), and NADH (C, $n = 3$, technical replicates per clone) level in NSCs.

D–F   LC-MS-based metabolomics for quantitative measurement of NAD+/NADH ratio (D, $n = 3$, technical replicates per clone), NAD+ (E, $n = 3$, technical replicates per clone), and NADH (F, $n = 3$, technical replicates per clone) level in iPSCs.

G       Intracellular ROS production measurements of the specific ROS level calculated by total ROS/MTDR in control, WS5A, and CP2A iPSC-derived NSCs using DCFDA and MTDR ($n = 6$, technical replicates per clone for control; $n = 5$, technical replicates per clone for WS5A; $n = 3$, technical replicates per clone for CP2A).

H       Flow cytometric measurements of mitochondrial ROS production at the specific ROS level in Detroit 551, WS5A, and CP2A iPSC-derived NSCs calculated by total ROS (MitoSox Red)/MTG ($n = 4$, technical replicates per clone for control and CP2A; $n = 3$, technical replicates per clone for WS5A).

I, J    Representative images of senescence β-galactosidase staining (scale bars, 20 μm) (I) and quantification by calculating the percentage of positively stained cells by division of the negative cells from I (J, $n = 3$, technical replicates per clone).

K, L    Representative images (K) and quantification (L) for western blotting with Phospho-SirT1 (Ser47), LC3B, UCP2, PINK1, Parkin, BNIP3, and β-ACTIN. Three independent experiments are included.

Data information: The data points in A–F and H represent iPSCs or iPSC-derived NSCs from 2 different clones from Detroit 551 control, 3 different clones from WS5A patient iPSCs, and 2 different clones from CP2A patient iPSCs. The data points in G represent NSCs generated from 5 different control iPSCs including 4 clones from Detroit 551 control and one clone from control AG05836, 3 clones from WS5A patient iPSCs, and 2 clones from CP2A patient iPSCs. The data points in J represent NSCs generated from 2 different clones from Detroit 551 control, 2 different clones from WS5A patient iPSCs, and 2 different clones from CP2A patient iPSCs. The data points in L represent NSCs generated from 2 to 3 different clones from Detroit 551 control and 3 different clones from WS5A patient iPSCs and 2–3 different clones from CP2A patient iPSCs. Data are presented as mean ± SEM for the number of samples. Mann–Whitney $U$-test was used for the data presented in G, L, e, and L, f. Two-sided Student's $t$-test was used for the data presented in A–F, H, J, and L, a–d. Significance is denoted for $P$ values of less than 0.05. *$P < 0.05$; **$P < 0.01$; ***$P < 0.001$; ****$P < 0.0001$.

Source data are available online for this figure.

senescence induced by oxidative stress via p53 acetylation (Furukawa *et al*, 2007). Further, it has been reported that SirT1 represses expression of the UCP2 gene by binding directly to the UCP2 promoter (Amat *et al*, 2007). UCP2, a mitochondrial transporter present in the inner mitochondrial membrane, plays an important role in uncoupling oxidative phosphorylation and decreasing mitochondrial $O_2$ consumption by regulating the MMP. UCP2 overproduction following total loss of cellular ATP was found to induce irreversible metabolic changes and senescent-like morphology (Nishio & Ma, 2016). Here, we showed that SirT1 signaling decreased, while UCP2 signaling increased in POLG-mutated cells, suggesting that senescence may indeed contribute to neuronal loss in POLG-related diseases.

Mitophagy plays a role in maintaining mitochondrial health throughout life and preventing age-related disease. Mitophagy can either specifically eliminate damaged or dysfunctional mitochondria or clear all mitochondria during specialized developmental stages. Accumulating evidence suggests that mitophagy is associated with neurodegenerative disorders, including Parkinson's (Ryan *et al*, 2015), Huntington's (Khalil *et al*, 2015), and Alzheimer's disease (Fang *et al*, 2019). Recent progress in mitophagy studies revealed that mitochondrial priming was mediated either by the PINK1-Parkin signaling pathway (Shiba-Fukushima *et al*, 2012) or by the mitophagic receptors Nix (Vo *et al*, 2019) and BNIP3 (Tang *et al*, 2019). Our finding on the upregulation of BNIP3 protein levels and occurrence of mitochondrial autophagosomes in NSCs with heterozygous *POLG* mutations suggested active degradation through mitophagy. This evidence suggested that BNIP3-mediated autophagy/mitophagy may be involved in POLG-related pathogenesis.

There are two technical questions pertaining to our study that require discussion. Firstly, isogenic controls were not used. We recognize that the current state of the art is to compare patient samples to gene-corrected isogenic controls, usually made by CRISPR-based gene editing. In our studies, however, we considered that the use of multiple, age-matched controls ($n = 5$) remained a viable alternative. This choice was, in part, driven by the presence of a compound heterozygous patient (CP2A patient). Further, we

would point out that many studies still use age/gender-matched controls from healthy individuals as disease comparators as exemplified by the recent study of another *POLG* mutation, p.Q811R (Chumarina *et al*, 2019). Nevertheless, in cases of loss of function mutations and to minimize background-specific confounding factors, we agree that gene-corrected isogenic controls or a well-executed rescue experiment should be conducted.

Secondly, we used retroviral reprogramming to generate most of the cell lines studied. Since the first experiments using integrating retroviral vectors, various approaches to deliver the reprogramming genes have been described, most notably newer integration-free and viral-free methods (Stadtfeld & Hochedlinger, 2010; Bellin *et al*, 2012; Robinton & Daley, 2012). These new techniques avoid insertional mutagenesis and transgene reactivation and can minimize variability between reprogrammed cell lines. While non-integrating methods can benefit both disease modeling and the future use in cell transplantation therapies, many excellent studies of disease modeling have used the retroviral system (Bellin *et al*, 2012). This may, in part, be explained by the limited protocol efficiencies of the new methods. Among the newer techniques, episomal plasmids and Sendai virus are the more commonly used (Okita *et al*, 2011; Chumarina *et al*, 2019), but reprogramming using synthetic mRNAs is also possible. As the non-integrating reprogramming technology improves, these technologies will be preferred in future studies.

In conclusion, our previous studies on patient tissues showed that POLG mutations lead to neuronal mtDNA depletion and, given sufficient time, point mutations and deletions (Tzoulis *et al*, 2013, 2014). Our current studies confirm that there is loss of ATP and membrane potential that we believe leads to changes in redox potential and excess ROS production. We also find activation of mitophagy. Our hypothesis is that neuronal death occurs by two mechanisms: Firstly, the initiation of seizures in already stressed neurons exceeds the capacity of this neuron to maintain cellular energy production and thus cellular integrity and leads to acute neuronal death. Secondly, the loss of ATP with changes in redox potential and ROS production lead to neuronal dysfunction that eventually leads to chronic neuronal loss (Fig 9).

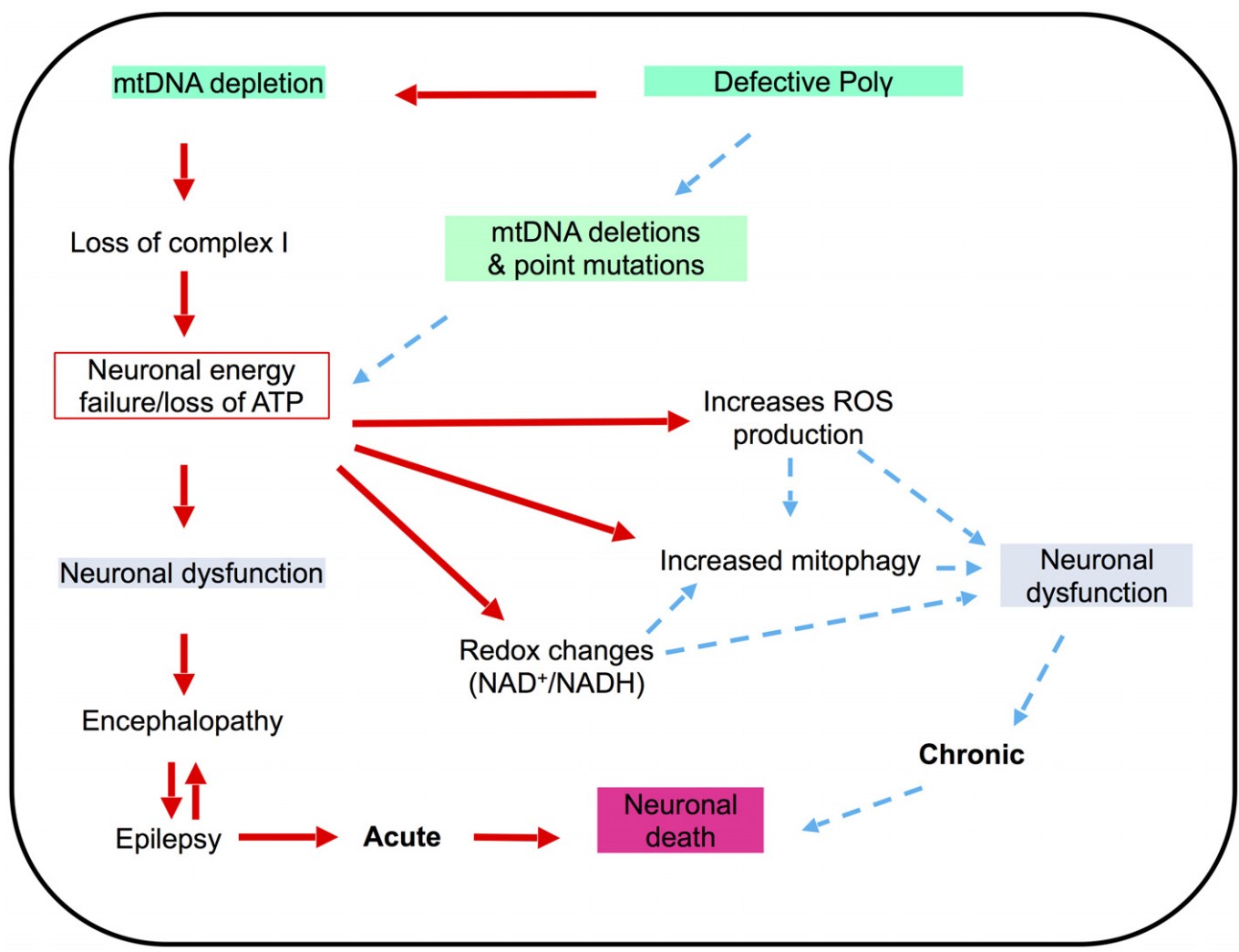

**Figure 9. Summary of the possible disease mechanisms in neuronal cells with *POLG* mutations.**

We believe that our studies are the first that show that it is possible to recapitulate the biochemical and molecular findings associated with the common mutations in POLG. Further, we believe that these studies demonstrate that iPSC-derived neural stem cells provide a robust model system in which to study tissue-specific mitochondrial disease manifestations. Since NSCs can be grown in large numbers in smaller formats, we hope to use this system to establish a high-throughput screening system in order to identify therapies for these devastating diseases. Our studies indicate further that the health potential of targeting NAD$^+$ homeostasis will inform clinical study design to identify nutraceutical approaches for combating POLG disease.

## Materials and Methods

### Ethics approval

The project was approved by the Western Norway Committee for Ethics in Health Research (REK nr. 2012/919). Tissues were acquired with written informed consent from all patients, and the experiments conformed to the principles set out in the WMA Declaration of Helsinki and the Department of Health and Human Services Belmont Report.

### Derivation of iPSCs

Skin fibroblasts from one homozygous c.2243G>C, p.W748S/W748S (WS5A) and one compound heterozygous c.1399G>A/c.2243G>C, p.A467T/W748S (CP2A) patient were collected by punch biopsy and cultured in DMEM/F12, GlutaMAX™ (Thermo Fisher Scientific, cat. no. 10565018) with 10% ($v/v$) FBS (Sigma-Aldrich, cat. no. 12103C), 20 mM glutamine (Sigma-Aldrich, cat. no. G7513), 10 mM sodium pyruvate (Invitrogen, cat. no. 11360 070), and 0.5 mM uridine (Sigma-Aldrich, cat. no. U 3003). Detroit 551 (control 1, ATCC® CCL 110™, human normal fetal female fibroblast), CCD-1079Sk (control 2, ATCC® CRL-2097™, human normal new-born male fibroblast), and AG05836 (control 3, RRID:CVCL2B58, 44-year-old female fibroblasts) were used as controls. Control fibroblasts were grown in

DMEM/F12, GlutaMAX™ (Thermo Scientific, cat. no. 35050061) with 10% (v/v) FBS.

We reprogrammed Detroit 551 and CCD-1079Sk control fibroblasts, and patient fibroblasts using retroviral vectors encoding POU5F1, SOX2, Klf4, and c Myc as previously described (Siller et al, 2016). AG05836 control fibroblasts were reprogrammed by Sendai virus vectors. We also employed two human embryonic stem cell lines (hESCs) as controls: The hESC line 360 (male) was obtained from the Karolinska Institute, Sweden, (Strom et al, 2010) and H1 (male) from WiCell Research Institute (Thomson et al, 1998).

Both iPSC and hESC lines were maintained under feeder-free conditions using Geltrex (Invitrogen, cat. no. A1413302) in E8 medium (Invitrogen, cat. no. A1517001) in 6-well plates (Thermo Scientific, cat. no. 140675). Each line was passaged with 0.5 mM EDTA (Invitrogen, cat. no. 15575038) at 70–80% confluency. The E8 was changed every day, and the cells passaged every 3–4 days. All the cells were monitored for mycoplasma contamination regularly using MycoAlert™ Mycoplasma Detection Kit (Lonza, cat. no. LT07-218).

### Neural induction

Neural induction was based on a previously described method (Stacpoole et al, 2011) with minor modifications. Briefly, 70% confluent iPSCs were split and seeded onto feeder-free Geltrex-coated plates in E8 medium. After 24 h, the medium was changed to neural induction medium prepared by addition of 10 μM SB431542 (Tocris Bioscience, cat. no. 1614), 10 μM N Acetyl L cysteine (NAC, Sigma-Aldrich, cat. no. A7250), and 2 μM AMPK inhibitor Compound C (EMD Millipore, cat. no. US1171261 1MG) to Chemically Defined Medium (CDM) containing 50% Iscove's Modified Dulbecco's Medium (IMDM, Invitrogen, cat. no. 21980 065), 50% F12 Nutrient Mixture (Ham) liquid with GlutaMAX™, 5 mg/ml bovine serum albumin (BSA) Fraction V (Europa bioproducts ITD, cat. no. EQBAC62 1000), 1% (v/v) Lipid 100 X (Invitrogen, cat. no. 11905 031), 450 μM 1 thioglycerol (Sigma-Aldrich, cat. no. M6145 25ML), 7 μg/ml insulin (Roche cat. no. 11376497001), and 15 mg/ml transferrin (Roche, cat. no. 10652202001). Medium was changed daily until cells reached the neural epithelial stage at day 5. The cells were then detached by incubation with collagenase IV (Invitrogen, cat. no. 17104 019) at 37°C for 1 min. After incubation, the cells were washed with DPBS without calcium and magnesium (DPBS$^{-/-}$) before addition of StemPro™ NSC medium supplemented with 1× GlutaMAX™, bFGF, and EGF (Thermo Fisher, cat. no. A1050901). Spheres were then generated by scraping a grid pattern at the bottom of each well using a pipette tip, and the cell suspensions transferred into 100 mm Corning® non-treated culture dishes (Sigma-Aldrich, cat. no. CLS430591) and incubated at 37°C on an orbital shaker (Fisher Scientific, cat. no. SGM 250 030K). After 2–3 days, the spheres were collected and dissociated into single cells by incubation with TrypLE™ Express at 37°C for 10 min followed by trituration. After neutralization with DMEM with 10% (v/v) FBS, the cell pellet was reconstituted in StemPro NSC medium and seeded on Geltrex-coated 6-well plates as monolayer NSCs. All the NSCs used for further analysis was limited to passages 4–9 in order to maintain them as primary lines.

### Mitochondrial volume and MMP measurement

To measure mitochondrial volume and MMP, cells were double-stained with 150 nM MitoTracker Green (MTG) (Invitrogen, cat. no. M7514) and 100 nM TMRE (Abcam, cat. no. ab113852) for 45 min at 37°C. Cells treated with 100 μM FCCP (Abcam, cat. no. ab120081) were used as negative control. Stained cells were washed with PBS, detached with TrypLE™ Express, and neutralized with media containing 10% FBS. The cells were immediately analyzed on a FACS BD Accuri™ C6 flow cytometer (BD Biosciences, San Jose, CA, USA). The data analysis was performed using Accuri™ C6 software. For each sample, more than 40,000 events were analyzed, and doublets or dead cells excluded.

### Mitochondrial DNA quantification and deletion assessment

DNA was extracted using a QIAGEN DNeasy Blood and Tissue Kit (QIAGEN, cat. no. 69504) according to the manufacturer's protocol. MtDNA quantification and depletion assessment were performed using RT-qPCR and long range polymerase chain reaction (Long-PCR) as previously described (Tzoulis et al, 2014).

### Hepatocyte differentiation

Hepatocyte differentiation from iPSCs was optimized for patient and control lines using a previously established approach (Siller et al, 2015, 2016; Siller & Sullivan, 2017). Terminal differentiation to hepatocyte-like cells was performed using the optimal conditions identified above and following a previously reported small molecule driven protocol (Siller et al, 2015; Mathapati et al, 2016).

Immunohistochemistry for identifying the hepatocytes was performed as previously described (Siller et al, 2015, 2016; Mathapati et al, 2016; Siller & Sullivan, 2017). Images were obtained using a Zeiss LSM700 Confocal microscope (Carl Zeiss Meditec AG, Germany).

### Cardiomyocyte differentiation

Cardiomyocyte differentiation was done utilizing a well-established protocol for generating cardiomyocytes through sequential activation and inhibition of Wnt signaling pathway (Lian et al, 2013).

### Karyotype analyses

Human G banding karyotyping was performed using standard methods. For each cell line, 20 chromosomes were analyzed from live or fixed cells in metaphase. The analysis was performed using G banding and Leishman stain, and the cells were analyzed according to the Clinical Cytogenetics Standards and Guidelines published by the American College of Medical Genetics (Meisner & Johnson, 2008).

### Glial cells and DA neuron differentiation

For astrocyte differentiation, NSCs were plated on poly-D-lysine (PDL)-coated coverslips (Neuvitro, cat. no. GG-12-15-PDL). The following day, the cells were washed with DPBS$^{-/-}$ and cultured in astrocyte differentiation medium: DMEM/F-12, GlutaMAX™ supplemented with 1× N2 (Invitrogen, cat. no. 17502-048), 1× B27

(Invitrogen, cat. no. 17504044-10 ml), 200 ng/ml insulin-like growth factor-I (IGF-I) (Sigma-Aldrich, cat. no. I3769-50UG), 10 ng/ml heregulin 1β (Sigma-Aldrich cat. no. SRP3055-50UG), 10 ng/ml activin A (PeproTech, cat. no. 120-14E), 8 ng/ml FGF2 (PeproTech, cat. no. 100-18B), and 1% FBS. The medium was changed every other day for the first week, every 2 days for the second week, and every 3 days for the third and fourth week.

After 28 days of differentiation, the cells were cultured in maturation medium AGM™ Astrocyte Growth Medium BulletKit™ (Lonza, cat. no. CC-3186), including Astrocyte Basal Medium supplemented with ascorbic acid, rhEGF, Gentamicin Sulfate/Amphotericin (GA-1000), insulin, L-glutamine, and FBS for one more month.

For glial oligodendrocyte differentiation, NSCs were plated at different cell densities onto surfaces coated with PDL or 0.01% poly-L-ornithine (Sigma-Aldrich, cat. no. P4957) and 5 μg/ml laminin (Sigma-Aldrich, cat. no. L2020). After 2 days, differentiation was performed according to commercial protocol (Invitrogen). Briefly, oligodendrocyte differentiation medium was supplemented with 1× Neurobasal medium (Invitrogen, cat. no.: 21103049) with 2% B27 Serum Free Supplement, 2 mM GlutaMAX™, and 30 ng/ml triiodothyronine (T3) (Sigma-Aldrich, cat. no. D6397). Differentiating oligodendrocyte cells were kept in these conditions, and the medium was changed every other day.

For DA neuron differentiation, neurospheres generated from neural induction were maintained in CDM supplemented with 100 ng/ml FGF8b (R&D systems, cat. no. 423-F8) over a period of 7 days to initiate DA progenitor induction. The following 7 days, the medium was changed to CDM supplemented with 1 μM PM (EMD Millipore, cat. no. 540220-5MG) and 100 ng/ml FGF-8. Termination of the suspension cultures was performed by dissociating the spheres into single cells by incubation with TrypLE™ Express followed by trituration and subsequent plating into monolayers. The DA neurons were matured in DA medium: CDM supplemented with 10 ng/ml BDNF (PeproTech, cat. no. 450-02) and 10 ng/ml GDNF (PeproTech, cat. no. 450-10) on poly-L-ornithine (Sigma-Aldrich, cat. no. P4957) and laminin (Sigma-Aldrich, cat. no. L2020)-coated plates.

## Immunocytochemistry and immunofluorescence (ICC/IF)

Cells were fixed with 4% (*v/v*) paraformaldehyde (PFA, VWR, cat. no. 100503 917) and blocked using blocking buffer containing 1× PBS, 10% (*v/v*) normal goat serum (Sigma-Aldrich, cat. no. G9023) with 0.3% (*v/v*) Triton™ X-100 (Sigma-Aldrich, cat. no. X100-100ML). The cells were then incubated with primary antibody solution overnight at 4°C and further stained with secondary antibody solution (1:800 in blocking buffer) for 1 h at RT. IPSCs were stained for pluripotent markers using the primary antibodies rabbit Anti-SOX2 (Abcam, cat. no. ab97959, 1:100), rabbit Anti-POU5F1 (Abcam, cat. no. ab19857, 1:100), and mouse Anti-SSEA4 [MC813] (Abcam, cat. no. ab16287, 1:200). NSCs were stained with rabbit Anti-PAX6 (Abcam, cat. no. ab5790, 1:100), mouse Anti-NESTIN (10c2) (Santa Cruz Biotechnology, cat. no. sc23927, 1:50), rabbit Anti-SOX2, and mitochondrial complex I subunit rabbit Anti-NDUFB10 (Abcam, cat. no. ab196019, 1:1,000). Astrocytes and oligodendrocytes were stained with chicken Anti-GFAP (Abcam cat. no. ab4674, 1:400), rabbit Anti-S100β (Abcam cat. no. ab196442,

1:100), rabbit Anti-EAAT1 (Abcam cat. no. ab416, 1:200), mouse Anti-Glutamine Synthetase (Abcam cat. no. ab64613, 1:200), rabbit Anti-GALC (Abcam cat. no. ab2894, 1:200), rabbit Anti-OLIGO2 (Abcam cat. no. ab42453, 1:1,000), and rabbit Anti-MBP (Abcam cat. no. ab62631, 1:500) respectively. The antibodies used for DA neuron staining were rabbit Anti-tyrosine hydroxylase (Abcam, cat. no. ab75875, 1:100), mouse Anti-Beta III Tubulin (Abcam, cat. no. ab78078, 1:1,000), and chicken Anti-MAP2 (Abcam, cat. no. ab5392, 1:100). The secondary antibodies used were Alexa Flour® goat Anti-rabbit 488 (Thermo Fisher Scientific, cat. no. A11008, 1:800), Alexa Flour® goat Anti-rabbit 594 (Thermo Fisher Scientific, cat. no. A11012, 1:800), Alexa Flour® goat Anti-mouse 594 (Thermo Fisher Scientific, cat. no. A11005, 1:800), and Alexa Flour® goat Anti-chicken 594 (Thermo Fisher Scientific, cat. no. A11042, 1:800). After incubation with secondary antibodies, the coverslips were mounted onto cover slides using Prolong Diamond Antifade Mountant with DAPI (Invitrogen, cat. no. P36962).

For staining of neurospheres, spheres were spread directly onto cover slides and left at RT until completely dry and then fixed with 4% (*v/v*) PFA. After two washes with PBS, the spheres were covered in PBS with 20% sucrose, sealed with parafilm, and incubated overnight at 4°C. The spheres were blocked with blocking buffer for 2 h at RT, and the primary antibodies were added to the samples overnight at 4°C. After washing the samples for 3 h in PBS with a few changes of buffer, incubation with secondary antibodies (as described above) was conducted overnight at 4°C in a humid and dark chamber. Coverslips were mounted using Fluoromount-G (Southern Biotech, cat. no. 0100 01) before imaging was performed using the Leica TCS SP5 or SP8 confocal microscope (Leica Microsystems, Germany).

## ATP generation assay

The Luminescent ATP Detection Assay Kit (Abcam, cat. no. ab113849) was used to investigate intracellular ATP production. Cells were cultured in a Corning® 96-well flat, clear bottom, white wall plate (Life Sciences, cat. no. 3601), and ATP measurements were performed according to the manufacturer's protocol when cells had reached 90% confluence. The kit irreversibly inactivates ATP degrading enzymes (ATPases) during the lysis step and measures the luminescent signal corresponding to the endogenous levels of ATP. Three to six replicates were measured for each sample. Luminescence intensity was monitored using the Victor® XLight Multimode Plate Reader (PerkinElmer). To normalize the value with cell number, cells grown on the same 96-well plates were incubated with Janus Green cell normalization stain after manufacturer's instructions (Abcam, cat. no. ab111622). OD 595 nm was measured using the Labsystems Multiskan® Bichromatic plate reader (Titertek Instruments, USA).

## ROS production

Intracellular ROS production was measured by flow cytometry using dual staining of 30 μM DCFDA (Abcam, cat. no. b11385) and 150 nM MTDR (Invitrogen, cat. no. M22426), which enabled us to assess ROS level related to mitochondrial volume. Mitochondrial ROS production was quantified using co-staining of 10 μM Mito-SOX™ red mitochondrial superoxide indicator (Invitrogen, cat. no.

M36008) and 150 nM MTG to evaluate ROS level in relation to mito-chondrial volume. Stained cells were detached with TrypLE™ Express and neutralized with media containing 10% FBS. The cells were immediately analyzed on a FACS BD Accuri™ C6 flow cytome-ter. For each sample, more than 40,000 events were recorded, and doublets or dead cells excluded before data analysis was performed using the Accuri™ C6 software.

## NAD⁺ metabolism measurement using LC-MS analysis

Cells were washed with PBS and extracted by addition of ice-cold 80% methanol followed by incubation at 4°C for 20 min. Thereafter, the samples were stored at −80°C overnight. The following day, samples were thawed on a rotating wheel at 4°C and subsequently centrifuged at 16,000 $g$ and 4°C for 20 min. The supernatant was added to 1 volume of acetonitrile, and the samples were stored at −80°C until analysis. The pellet was dried and subsequently recon-stituted in lysis buffer (20 mM Tris–HCl (pH 7.4), 150 mM NaCl, 2% SDS, 1 mM EDTA) to allow for protein determination (BCA assay).

Separation of the metabolites was achieved with a ZIC-pHILC column (150 × 4.6 mm, 5 μm; Merck) in combination with the Dionex UltiMate 3000 (Thermo Scientific) liquid chromatography system. The column was kept at 30°C. The mobile phase consisted of 10 mM ammonium acetate pH 6.8 (Buffer A) and acetonitrile (Buffer B). The flow rate was kept at 400 μl/min, and the gradient was set as follows: 0 min 20% Buffer B, 15 min to 20 min 60% Buffer B, and 35 min 20% Buffer B. Ionization was subsequently achieved by heated electrospray ionization facilitated by the HESI-II probe (Thermo Scientific) using the positive ion polarity mode and a spray voltage of 3.5 kV. The sheath gas flow rate was 48 units with an auxiliary gas flow rate of 11 units and a sweep gas flow rate of 2 units. The capillary temperature was 256°C, and the auxiliary gas heater temperature was 413°C. The stacked-ring ion guide (S-lens) radio frequency (RF) level was at 90 units. Mass spectra were recorded with the QExactive mass spectrometer (Thermo Scientific), and data analysis was performed with the Thermo Xcalibur Qual Browser. Standard curves generated for NAD⁺ and NADH were used as reference for metabolite quantification.

## NAD⁺ metabolism measurement using colorimetric analysis

The ratio of NAD⁺/NADH was quantified with a commercial NAD⁺/NADH Quantitation Colorimetric Kit (BioVision, cat. no. K337) according to the manufacturer's instructions and as previ-ously described (Gomes *et al*, 2012).

## Protein level assessment of TFAM and mitochondrial respiratory chain complexes

Cells were detached with TrypLE™ Express, pelleted, and fixed in 1.6% (*v/v*) PFA at RT for 10 min, before permeabilization with ice-cold 90% methanol at −20°C for 20 min. The cells were blocked using buffer containing 0.3 M glycine, 5% goat serum, and 1% BSA in PBS. For TFAM expression, cells were stained with Anti-TFAM antibody conjugated with Alexa Fluor® 488 (Abcam, cat. no. ab198308, 1:400) and Anti-TOMM20 antibody conjugated with Alexa Fluor® 488 (Santa Cruz Biotechnology, cat. no. sc 17764

AF488, 1:400), separately. Staining of mitochondrial respiratory chain complexes was conducted using primary antibodies Anti-NDUFB10 (Abcam, cat. no. ab196019, 1:1,000), Anti-SDHA [2E3GC12FB2AE2] (Abcam, cat. no. ab14715, 1:1,000), and Anti-COX IV [20E8C12] (Abcam, cat. no. ab14744, 1:1,000), followed by secondary antibody incubation (1:400). The cells were immediately analyzed on BD Accuri™ C6 flow cytometer, and Accuri™ C6 soft-ware was used for data analysis. Both dot plots of SSC-H/SSC-A and FSC-H/FSC-A were used to exclude duplicates. For each sample, more than 40,000 events were recorded.

## Western blotting

Extraction of protein was performed using 1× RIPA lysis buffer (Sigma-Aldrich, cat. no. R0278) supplemented with Halt™ Protease and Phosphatase Inhibitor Cocktail (Invitrogen, cat. no. 78444). Protein concentration was determined using BCA protein assay (Thermo Fisher Scientific, cat. no. 23227). The cell protein was loaded into NuPAGE™ 4–12% Bis-Tris Protein Gels (Invitrogen, cat. no. NP0321PK2) and resolved in PVDF membrane (Bio-Rad, cat. no. 1704157) using the Trans-Blot® Turbo™ Transfer System (Bio-Rad, Denmark). Membranes were blocked with 5% non-fat dry milk or 5% BSA in TBST for 1 h at RT. Membranes were then incubated overnight at 4°C with rabbit monoclonal IgG Anti-UCP2 (1:1,000, Cell Signalling, cat. no. 89326), rabbit polyclonal IgG Anti-Phospho-SirT1 (Ser47) (1:2,000, Cell Signalling, cat. no. 2314), rabbit poly-clonal IgG Anti-PINK1 (1:500, Proteintech, cat. no. 23274-1-AP), rabbit polyclonal IgG Anti-Parkin (1:500, Proteintech, cat. no. 14060-1-AP), rabbit polyclonal IgG Anti-LC3B (1:3,000, Abcam, cat. no. ab51520), rabbit polyclonal Anti-BNIP3 (1:1,000, Abcam, cat. no. ab10962) or rabbit polyclonal IgG Anti-UCP2 (1:500, Protein-tech, cat. no. 11081-1-AP), and mouse monoclonal IgG Anti-β-ACTIN antibody conjugated to HRP (1:5,000, Abcam, cat. no. ab49900) as a control. After washing in TBST, membranes were incubated with donkey Anti-mouse monoclonal antibody or swine Anti-rabbit monoclonal antibody conjugated to HRP secondary anti-body (Jackson ImmunoResearch, 1:1,000), for 1 h at RT. Super Signal West Pico Chemiluminescent Substrate (Thermo Fisher Scien-tific, cat. no. 34577) was used as enzyme substrate according to manufacturer's recommendations. The membranes were visualized in SynGene scanner (VWR, USA).

## β-galactosidase staining assay

β-galactosidase activity was detected by using a Senescence β-galac-tosidase Staining Kit (Cell Signaling, cat. no. 9860) according to the manufacturer's instructions. The cells were seeded in complete media into 6-well plates with coverslips. After 24 h-incubation, the cells were fixed with 1× fixative solution and incubated with the β-galactosidase staining solution overnight at 37°C in a dry incubator without $CO_2$. Afterward, cells were observed, and images were captured under Nikon TE2000 fluorescence microscope.

## Tissue studies

Formalin-fixed, paraffin-embedded (FFPE), and fresh frozen brain tissues were available from patients with POLG disease ($n = 5$) and neurologically healthy controls ($n = 5$) who were demographically

matched. Informed consent was obtained from all subjects and that the experiments conformed to the principles set out in the WMA Declaration of Helsinki and the Department of Health and Human Services Belmont Report. The *POLG* mutations for these patients are A467T/G303R (AL-1B); A467T/A467T (AT-1A); A467T/A467T (AT-2A); W748S/W748S (WS-10A); and W748S/W748S (WS-3A). Patient IDs used in our previous publications were shown in parenthesis to allow comparison with results of our earlier work. There were no statistically significant differences in post-mortem interval or length of fixation between patient and control tissue. Samples were dissected at autopsy and either snap-frozen immediately in isopentane, which had been cooled in liquid nitrogen, and stored at −80°C, or fixed in formaldehyde and later embedded in paraffin blocks according to standard procedures.

Mitochondrial complex I immunohistochemistry was performed on 4-μm sections of formalin-fixed, paraffin-embedded tissue from the primary occipital cortex of three patients and two neurologically healthy controls as previously described (Tzoulis *et al*, 2014).

Neurons for mtDNA analysis were microdissected from frozen sections of primary occipital (Brodmann area 17) from five patients and five age-matched controls. Microdissection and cell lysis were carried out as previously described (Tzoulis *et al*, 2013). Only cells that could be positively identified as neurons, with a visible nucleus and normal morphological characteristics, were used. For each area, there was no significant size difference between neurons of patients and corresponding controls. A total of 400 neurons were picked from 5 patients ($n = 200$) and five age-matched controls ($n = 200$). Neurons were microdissected, avoiding carryover of glia or other cells, and pooled in 3–5 groups of 10 cells per individual. MtDNA copy number quantification was performed in microdissected neurons by qRT-PCR, using TaqMan fluorogenic probes and a 7500 fast sequence detection system (ABI) as previously described (Tzoulis *et al*, 2013).

## Gene expression

Total RNA was isolated using MagMAX™ 96 Total RNA isolation Kit (Thermo Fisher Scientific, cat. no. AM1830). High-throughput MagMAX™ Express 96 was employed to extract RNA from the cell lysate.

EXPRESS One Step Superscript™ RT-qPCR Kit (Thermo Fisher Scientific, cat. no. 11781 200) and TaqMan™ Probes were recruited to perform cDNA synthesis and RT-qPCR in one step. Applied Biosystems 7500 Fast Real-Time PCR machine (Thermo Fisher Scientific) was used to perform qRT-PCR. TaqMan™ probes (Life Technologies) for *POU5F1* (Hs00999634), *NANOG* (Hs04260366), *PAX6* (Hs01088114), *NESTIN* (Hs04187831), and *LIN28A* (Hs00702808) were used. The average CT values of three technical replicates were normalized to the geometric mean of endogenous control gene, *Actin Beta* (ACTB: Hs01060665). The expression of the iPSC markers was assessed with fold change by using the comparative $\Delta\Delta C_t$ method by normalizing the gene level from ESC1. The expression of the NSC markers was assessed with fold change by using the comparative $\Delta\Delta C_t$ method by normalizing NSCs to iPSCs.

## DNA sequencing for *POLG* mutations

Forward and backward oligonucleotide primers were used to amplify the 7 exons and 13 exons of the *POLG* gene, as reported

### The paper explained

#### Problem

Mitochondrial diseases are the most common with inborn errors of metabolism and mutations in *POLG*, the gene encoding the catalytic subunit of the mitochondrial DNA polymerase gamma, the most common subgroup. These diseases are often associated with catastrophic involvement of the brain, and currently, there are no cures and no robust models to study disease mechanisms in neuronal tissue. We used neural stem cells (NSCs) produced from patient induced pluripotent stem cells (iPSCs) to study disease mechanisms.

#### Results

We generated iPSCs containing two founder mutations (W748S homozygous; W748S/A467T compound heterozygous) and differentiated these into NSCs. These neural precursors manifested features that faithfully replicated the molecular and biochemical changes found in patient post-mortem brain tissue, namely mtDNA depletion and loss of complex I. We also confirmed the same phenotypes in dopaminergic neurons generated from these iPSCs. POLG-driven mitochondrial dysfunction also led to neuronal ROS overproduction, increased cellular senescence, and disturbed NAD$^+$ metabolism, a feature that reflected the loss of complex I.

#### Impact

This is the first model of POLG disease that replicates exactly what is seen in patient tissues. Using this system, it was possible to examine the consequences of POLG-induced loss of mtDNA and complex I and show how *POLG* mutations affects NAD$^+$ metabolism and cellular fate. We believe that iPSC-derived NSCs provide a robust model system in which to study tissue specific mitochondrial disease manifestations, and we hope to use this system to establish a high-throughput screening system in order to identify therapies for these devastating diseases.

elsewhere (Hakonen *et al*, 2005). Automated nucleotide sequencing was performed using the Applied Biosystems™ BigDye® Terminator v3.1 Cycle Sequencing Kit (Invitrogen, cat. no. 4337454) and analyzed on an ABI3730 Genetic Analyzer with sequencing analyzer software ChromasPro (Technelysium Pty Ltd, Australia). The primers designed for seven exons in *POLG 1* gene were: forward, 5′ TGTAAAACGACGGCCAGTGAAAGAACTGAG GCTCCGAG 3′ and reverse, 5′ CAGGAAACAGCTATGACCCCTACAGAGCCA GTCCACT 3′. The primers designed for 13 exons in *POLG1* gene were: forward, 5′ TGTAAAACGACGGCCAGTATTTCCCAGCTG ATGACGAC 3′ and reverse, 5′ CAGGAAACAGCTATGACCTGCCACCCGACT TTCATTAG 3′. DNA Chromatogram was aligned with the best matching human sequences in NCBI Trace.

## Data analysis

In order to minimize the phenotypic diversity caused by intraclonal heterogeneity which is a common issue for iPSC-related studies, multiple clones in each line were included in the all the analysis and more than three biological repeats were conducted for each clone to ensure adequate power to detect a prespecified effect size. Data were presented as mean ± standard error of the mean (SEM) for the number of samples ($n \geq 3$ per clone). Distributions were tested for normality using the Shapiro–Wilk test. Outliers were detected using interquartile range (IQR) and Tukey's Hinges test. Mann–

Whitney *U*-test was used to assess statistical significance for variables with non-normal distribution, while two-sided Student's *t*-test was applied for normal distributed variables. Data were analyzed with SPSS software (SPSS v.25, IBM), and figures were produced by GraphPad Prism software (Prism 7.0, GraphPad Software, Inc.). Significance was denoted for *P* values of less than 0.05.

## Data availability

This study includes no data deposited in external repositories.

**Expanded View** for this article is available online.

## Acknowledgements

The authors encourage all laboratory members for discussions and critical reading of the manuscript. We are grateful to the Molecular Imaging Centre, Flow Cytometry Core Facility and Genomics Core Facility in University of Bergen in Norway. This work was supported by funding from the Norwegian Research Council (project no. 229652), Rakel og Otto Kr.Bruuns legat and Meltzer (project no. 809432). GJS was partly supported by the Norwegian Research Council through its Centres of Excellence funding scheme (project number 262613).

## Author contributions

KXL, LAB, and GJS conceptualized. KXL involved in methodology. KXL, CKK, SM, GHV, GAZ, AK, CT, LEH, MZ, RMP, ZZ, and NB investigated. KXL and CKK wrote original draft. KXL, CKK, MZ, YH, RS, GJS, and LAB involved in writing, review, and editing. JF performed statistical analysis. LAB, KXL, and GJS acquired funding. GJS, MZ, and LAB provided resources. KXL and LAB supervised. All authors agree to the authorship.

## Conflict of interest

The authors declare that they have no conflict of interest.

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
