## [Review Process File · EMBO Molecular Medicine]

Disease-specific phenotypes in iPSC-derived neural stem cells with POLG mutations

Kristina Xiao Liang, Cecilie Kristiansen, Sepideh Mostafavi, Guro Vatne, Gina Zantingh, Atefeh Kianian, Charalampos Tzoulis, Lena Høyland, Mathias Ziegler, Roberto Perez, Jessica Furriol, Zhuoyuan Zhang, Novin Balafkan, Yu Hong, Richard Siller, Gareth Sullivan, and Laurence Bindoff

DOI: [10.15252/emmm.202012146](https://doi.org/10.15252/emmm.202012146)

Corresponding authors: Kristina Xiao Liang (xiao.liang@uib.no) , Laurence Bindoff (laurence.albert.bindoff@helse-bergen.no)

Review Timeline:

Submission Date:	12th Feb 20
Editorial Decision:	16th Mar 20
Appeal Received:	27th Mar 20
Editorial Decision:	8th Apr 20
Revision Received:	23rd Jun 20
Editorial Decision:	9th Jul 20
Appeal Received:	13th Jul 20
Editorial Decision:	16th Jul 20
Revision Received:	30th Jul 20
Accepted:	31st Jul 20

Editor: Celine Carret

Transaction Report:

16th Mar 2020

Decision on your manuscript EMM-2020-12146

Dear Dr. Liang,

Thank you for the submission of your manuscript "Disease-specific phenotypes in iPSC-derived neural stem cells with POLG mutations". We have now heard back from the three referees whom we asked to evaluate your manuscript.

As you will see, all referees find some interest in the study and agree that such a model would be valuable to the field. However, although ref. #1 is rather supportive of publication, ref. #2 and #3 are less so and raise serious issues regarding the conclusiveness of the data and pinpoint several technical problems too that preclude a solid interpretation of the experimental evidence provided. As clear and conclusive insight into a novel clinically relevant observation is key for publication in EMBO Molecular Medicine, and together with the fact that we only accept papers that receive enthusiastic support upon initial review, I am afraid that we cannot offer to consider the manuscript further.

We hope that the referee comments will be helpful to you as you prepare your manuscript for submission elsewhere. Thank you again for your interest and we hope that you will continue to consider sending your work to EMBO Molecular Medicine in the future.

Yours sincerely,

Celine Carret

Celine Carret, PhD
Senior Editor
EMBO Molecular Medicine

***** Reviewer's comments *****

Referee #1 (Remarks for Author):

Liang et al reports the modeling the POLG diseases using human iPSCs and their neural derivatives. POLG is one of catalytic subunit of DNA polymerase gamma that catalyze the replication and repair of mitochondrial DNA. POLG mutations are known to cause diverse phenotypes, and investigating the role of POLG is critical to understand the diseases and to develop potential therapeutics. Authors derived iPSC lines from patients with homozygous POLG mutation (WS5A), or compound heterozygous mutation (CPA2A). Comparison of mitochondrial numbers, gene expression, and functions were made among different types of cells from the iPSCs. Authors found that neural stem cells (NSCs) and dopaminergic neurons (DA) displayed a major phenotypes in mitochondrial functional defect. Overall, the manuscript is well written, and logical and data is strong. However, there are some concerns to address before the publication.

1. There are some expression data that were presented in a non-conventional way. In Figure 1C, the expression of Sox2 in different clones is presented strangely. How the SOX2 level was normalized against which samples is not clear. Please describe how the fold change was calculated, or make correction. When characterizing the pluripotency of the iPSC lines, authors presented the median intensity of markers, SSEA4 and TRA160/TRA181. Since it is well known that the different clones express different levels of the markers, it is advised that authors remove the data and rather show the immunostaining data and FACS data showing the presence of the markers.

2. It is understandable that authors derived multiple iPSC lines from the patients of POLG mutations. However, when the data was presented it is not clear which cell lines were used. Authors should describe that in figure legends, or if adding the information in the legend is not practical, authors make a supplementary table to display the iPSC or hESC clones that were used for the given experiments. It is not good to describe, "...A-C are generated from two distinct ESC lines and 2 or 3 different clones of each line..."(Legend Figure 4BC). "...NSCs generated from Detroit 551, WS5A, and CP2A iPSCs..." (Legend Figure 5C).

3. In neuronal differentiation, there seem to be a difference between ESC and other iPSCs lines (Figure 4C). If there is not, authors should describe that there is no significant difference. In addition, the experiments on comparison between POLG mutant and wild type, ESC data was not presented and only one control iPSCs were used. If the control line itself is outlier, the whole conclusions in the data are not significant. If it is too much work to repeat all the experiments on NSC and DA cells from other control iPSC lines or ESC lines, authors may perform the experiments on fibroblasts, such as Figure 5. The findings will support one of major conclusion of the manuscript that POLG mutations have cell type specific effect on mitochondrial activity without mitochondrial DNA replication and numbers.

4. Some data need quantification and perform statistical analysis. In Figure EV10, the staining of the MtpHagy Dye does not support any conclusion. Authors need to take random pictures of the slides and quantify to examine whether there is statistically significant difference between normal and POLG mutant NSCs.

5. Authors should include a model to describe how POLG mutations cause the defect in mitochondrial activity and mitophagy at one of the last figures.

Referee #2 (Comments on Novelty/Model System for Author):

The value of the model is severely impacted by several major technical drawbacks. Please see my detailed explanation in my review, especially point 1 and 2 about isogenic controls and use of retrovirus for reprogramming.

Referee #2 (Remarks for Author):

The manuscript by Liang et al. describes the generation of iPSCs from patients with pathogenic POLG mutations and the characterization of these iPSCs and differentiated cells of the neuronal lineage. The POLG enzyme replicates mitochondrial DNA, and mutations in patients cause a variety

of symptoms in different tissues. Neurological defects are particularly common in mutation carriers. The authors describe some potentially disease-related phenotypes in neural stem cells derived from patient iPSCs related to mitochondrial dysfunction, which are novel, as previous reports dealing with POLG mutations in iPSCs did not report similar findings. Since there is currently no good human system besides patient material to study POLG mutations, the authors intent to introduce their iPSCs as a model to study disease mechanisms.

Overall the paper is well written and a human model for POLG mutations would be very useful for the community, but the manuscripts value is severely impacted by several (and in my opinion too many) major technical drawbacks:

1. Although the authors generate two independent iPSC lines with different mutations, they fail to compare those to appropriate isogenic controls. The currently used controls are one iPSC line generated from a completely unrelated fibroblast line from ATCC, which is not matched to patient lines in any way, as well as two ESC lines. Comparing to such unmatched and unrelated controls is clearly no longer state of the art and can severely affect the reliability of the phenotypes. Observed differences could either be due to the POLG mutations or some genetic background artefact in patient or control lines. The current state of the art is to compare to gene-corrected isogenic controls. If this is not possible for a specific reason (that needs to be clearly stated), comparison of multiple independent patient lines showing the same phenotypes to age-/gender- etc. matched lines, preferentially from unaffected siblings, is required. In case of loss of function, a well-executed rescue experiment would also be conceivable.

In this context, the different expression levels of SSEA4 and TRA160 shown in Figure 1 already indicate differences in line behavior at high level, so there could be more issues under the hood.

2. The second major drawback is the use of retroviral reprogramming, which was state of the art 10 years ago and has been long replaced by non-integrating reprogramming methods (Sendai, mRNA, episomal...). Earlier reports of iPSCs with POLG mutations used Sendai (Zurita et al. 2016, Chumarina et al. 2019), I therefore do not understand why this was not done here. Integration of retroviral DNA can mutate genes, leading to unspecific phenotypes that appear to be caused by the pathogenic mutations. This drawback, together with the lack appropriate isogenic controls, makes the whole study unreliable.

3. Why did the authors generate 3 controls, but compare only to one of them in their assays?

4. I am not a big fan of the quality of images and graphs. Most of them are quite blurry, especially in Fig. 1. The main Figures are unlabeled, making it hard to keep track. Nomenclature of supplementary figures is misleading (SI vs. EV)

5. EB-based NSC differentiation is no longer state of the art, current protocols utilize differentiation in fully defined media and attached in monolayer, which is much less prone to heterogeneity of the NSCs.

In this context, what the purity of NSCs? Nestin is high, but this is broadly expressed in many NSCs. What percentage of cells expresses Pax6, and other markers, such as FoxG1 and Sox2, how variable is this over multiple differentiations? Showing a low variability of differentiated cells is crucial to be able to see significant phenotypic differences. Many of the bar graphs show two or three populations, which could indicate heterogeneity.

6. Fig. 4e: GFAP is not sufficient as a marker to demonstrate Astrocyte identity, as it is also expressed in NSCs. At least some other AS markers, including S100B, GLT, GLAST should also be shown to make this conclusion. Similarly, GALC is not sufficient as a marker to demonstrate Oligodendrocyte identity. At least some other OL markers, including O4 and MBP should be shown.

7. Fig. 6h: TH and MAP2 stainings appear very poor, making successful DA neuron differentiation questionable. Both stainings appear to stain most cells in the images, with some more strongly stained cells present. Both the strongly and weakly stained cells appear quite immature in their morphology, with a relatively flat cell body and broad neurites that look more like processes from

radial-glia-like NPCs. I believe that most of this staining is background. The authors need to do a better job of proving that they indeed generate high amounts of pure dopaminergic neurons, and that they are mature, e.g. by functional assays.

8. Fig. 6k: mtDNA copy number is only (barely) significantly changed in one of the two iPSC lines, but not the other one. This makes me wonder how reproducible and robust this phenotype actually is. Also, there is clearly two populations in the control with only the upper one significantly changed. Is there a reason why in this and many other assays only one of the two patient lines shows an effect?

9. Fig. 7c: The variation in complex I in NSCs is very large, I am wondering how this can be 4-star significant, when at least half of the measurements of control are at the same level than the mutants.

10. Text and Fig. 8: NAD result are inconsistent between NSCs to iPSCs, making this whole section hard to understand.

11. Fig. 8k: Band intensities of WS5A-NSCs in Phospho-SirT1 blot do not fit to quantification. First band is much stronger than in Ctrl, two other bands weaker than CTRL. This high variation is not reflected in the error bar in L-b.

Some more suggestions to improve the impact of the manuscript:

1. Most assays are done in NSCs, but not in much more disease-relevant cell types, such as DA or cortical neurons. I think it would make the paper much stronger if clear phenotypes were shown in cell types present in the adult human body and affected by the disease.
2. It would be very interesting and a great opportunity to address selective vulnerability of affected and spared tissues by studying iPSC-derived cell types affected vs. unaffected in disease.
3. Performing stress treatments e.g. on DA neurons could be very interesting to elicit more compelling phenotypes and also see variations in vulnerability of mutant cells.
4. I would also suggest addressing why the neuronal lineage is particularly prone to loss of mitochondrial copy number.

Taken together, despite its potential value for the field of POLG mutations and mitochondrial biology in general, I believe that the manuscript in its current form does not provide enough convincing data at the required technical level to be ready for publication in a high-impact journal.

Referee #3 (Remarks for Author):

This is an interesting study reporting the generation of iPSCs carrying POLG mutations. These iPSCs could represent a useful tool to advance our understanding of the pathogenic mechanisms of POLG mutations.

Unfortunately however it is this reviewer's opinion that the level of insights into disease mechanisms is very limited in this study. There are no insights into the reason why POLG mutations affect neurons specifically. I also could not see clear experiments dissecting the mechanisms underlying the neuronal cell loss. Finally there is no suggestion of potential therapeutic strategies.

iPSCs can be a great resource but in order to be of interest to a broader audience, more insightful findings should be present. I think that the results would better fit in a more specialized journal.

As a service to authors, EMBO provides authors with the possibility to transfer a manuscript that one journal cannot offer to publish to another EMBO publication. The full manuscript and if applicable, reviewers reports are automatically sent to the receiving journal to allow for fast handling and a prompt decision on your manuscript. For more details of this service, and to transfer your manuscript to another EMBO title please click on Link Not Available

Dear Editor,

EMBO Molecular Medicine

We were disappointed that the manuscript was rejected by EMBO Molecular Medicine (EMM-2020-12146), particularly on the basis of what we consider inappropriately negative comments. We believe that all of the questions and comments raised by the reviewers could have been answered. We have provided answers to the questions raised by the reviewers and modified the manuscript according to the changes indicated. We have also added some new data that we consider relevant. We believe that the comments have resulted in a much better manuscript and we hope it is now reconsidered and finally acceptable for publication in your journal.

Please find also below a point-by-point response to the reviewers' comments typed in **BOLD**.

Referee #1 (Remarks for Author):

Liang et al reports the modeling the POLG diseases using human iPSCs and their neural derivatives. POLG is one of catalytic subunit of DNA polymerase gamma that catalyze the replication and repair of mitochondrial DNA. POLG mutations are known to cause diverse phenotypes, and investigating the role of POLG is critical to understand the diseases and to develop potential therapeutics. Authors derived iPSC lines from patients with homozygous POLG mutation (WS5A), or compound heterozygous mutation (CPA2A). Comparison of mitochondrial numbers, gene expression, and functions were made among different types of cells from the iPSCs. Authors found that neural stem cells (NSCs) and dopaminergic neurons (DA) displayed a major phenotype in mitochondrial functional defect. Overall, the manuscript is well written, and logical and data is strong. However, there are some concerns to address before the publication.

1. There are some expression data that were presented in a non-conventional way. In Figure 1C, the expression of Sox2 in different clones is presented strangely. How the SOX2 level was normalized against which samples is not clear. Please describe how the fold change was calculated or make correction. When characterizing the pluripotency of the iPSC lines, authors presented the median intensity of markers, SSEA4 and TRA160/TRA181. Since it is well known that the different clones express different levels of the markers, it is advised that authors remove the data and rather show the immunostaining data and FACS data showing the presence of the markers.

Answer: As pointed out by the reviewer, the expression of *SOX2* in different clones is presented higher expression than the other genes. As showed in the Figure A below, our immunostaining data identified the presence of *SOX2*. Therefore, we have removed the *SOX2* group to make it less problematic without influencing the reliability of the data. In the present manuscript, the average CT values of three technical replicates were normalized to the geometric mean of endogenous control gene, Actin Beta (ACTB: Hs01060665). The expression of the iPSC markers was assessed with fold change by using the comparative $\Delta\Delta C_t$ method and normalizing to the gene level from ESC1. The expression of the NSC markers was assessed with fold change by using the comparative $\Delta\Delta C_t$ method by normalizing NSCs to iPSCs, as described in the Materials and Methods section.

We agree with the point that it is well known that different clones express different levels of the markers we described that clonal variations for the protein level and mRNA expression were noticed (Fig. 1C, D, a; E, a; SI3A, a; SI3B, a and SI3C, a). In order to minimize the phenotypic diversity caused by intra-clonal heterogeneity, multiple clones were included in the further analysis. When characterizing the pluripotency of the iPSC lines, we measured the expression of pluripotent transcript factors POU5F1 (Figure 1E), Nanog (Figure 1F) and pluripotent surface marker markers SSEA4, TRA-1-60 and TRA-1-81 using flow cytometric analysis. Using this technique, we observed lower levels of the three pluripotent surface markers - SSEA4, TRA-1-60 and TRA-1-81 (Figure S1B) in both WS5A and CP2A iPSCs as compared to the 2 ESCs and control iPSC line. However, no change was observed in the expression of POU5F1 (Figure 1f) or Nanog (Figure S1C). Although the mechanisms for those findings need to be investigated in the future, we believe this phenotype may related to the mutation. Therefore, we still keep them in the revised manuscript.

2. It is understandable that authors derived multiple iPSC lines from the patients of POLG mutations. However, when the data was presented it is not clear which cell lines were used. Authors should describe that in figure legends, or if adding the information in the legend is not practical, authors make a supplementary table to display the iPSC or hESC clones that were used for the given experiments. It is not good to describe, "...A-C are generated from two distinct ESC lines and 2 or 3 different clones of each line..."(Legend Figure 4BC). "...NSCs generated from Detroit 551, WS5A, and CP2A iPSCs..." (Legend Figure 5C).

Answer: We thank to the reviewer for the suggestion and we have made a supplementary table (Table EV) to display the iPSC or hESC clones that were used for the given experiments.

3. In neuronal differentiation, there seem to be a difference between ESC and other iPSCs lines (Figure 4C). If there is not, authors should describe that there is no significant difference. In addition, the experiments on comparison between POLG mutant and wild type, ESC data was not presented and only one control iPSCs were used. If the control line itself is outlier, the whole conclusions in the data are not significant. If it is too much work to repeat all the experiments on NSC and DA cells from other control iPSC lines or ESC lines, authors may perform the experiments on fibroblasts, such as Figure 5. The findings will support one of major conclusion of the manuscript that POLG mutations have cell type specific effect on mitochondrial activity without mitochondrial DNA replication and numbers.

Answer: As rightly pointed by the reviewer, we do indeed see a difference between ESC and other iPSCs lines during neuronal differentiation. As showed below in Figure B, there is significant higher expression in iPSC of specific TFAM (total TFAM/TOMM20) using flow cytometric analysis. Therefore, in the experiments on comparison between POLG mutant and wild type, ESC data was not presented and only one control iPSCs was used. In the revised version, we have included one more control NSCs line from iPSC line derived from AG05836 fibroblasts (RRID: CVCL_2B58) and performed several assays of total MMP, specific MMP, MitoSox ROS and mtDNA copy number.

4. Some data need quantification and perform statistical analysis. In Figure EV10, the staining of the Mtpagy Dye does not support any conclusion. Authors need to take random pictures of the slides and quantify to examine whether there is statistically significant difference between normal and POLG mutant NSCs.

Answer: As mentioned by the reviewer, the staining of the Mtpagy Dye does not support any conclusion and we believe that the Western Blotting data are sufficient to provide the evidence of mitophagy, therefore we have deleted the images of Mtpagy Dye in the revised manuscript.

5. Authors should include a model to describe how POLG mutations cause the defect in mitochondrial activity and mitophagy at one of the last figures.

Answer: We thank to the reviewer for bringing up the idea and we have included a model to describe how POLG mutations cause the defect in mitochondrial activity and mitophagy at the end in the revised manuscript, see the Figure below as well. From our earlier studies, we know that POLG mutations lead neuronal mtDNA depletion and, given sufficient time, point mutations and deletions. Our current studies confirm that there is loss of ATP and membrane potential that we believe leads to changes in redox potential and excess ROS production. We also find activation of mitophagy. Our hypothesis is that neuronal death occurs by two mechanisms: firstly, the initiation of seizures in already stressed neurons exceeds the capacity of these neuron to maintain cellular energy production and thus cellular integrity and leads to acute neuronal death. Secondly, the loss of ATP with changes in redox potential and ROS production, lead to neuronal dysfunction that eventually leads to chronic neuronal loss.

Figure 9 in revised manuscript

Referee #2 (Comments on Novelty/Model System for Author):

The value of the model is severely impacted by several major technical drawbacks. Please see my detailed explanation in my review, especially point 1 and 2 about isogenic controls and use of retrovirus for reprogramming.

Referee #2 (Remarks for Author):

The manuscript by Liang et al. describes the generation of iPSCs from patients with pathogenic POLG mutations and the characterization of these iPSCs and differentiated cells of the neuronal lineage. The POLG enzyme replicates mitochondrial DNA, and mutations in patients cause a variety of symptoms in different tissues. Neurological defects are particularly common in mutation carriers. The authors describe some potentially disease-related phenotypes in neural stem cells derived from patient iPSCs related to mitochondrial dysfunction, which are novel, as previous reports dealing with POLG mutations in iPSCs did not report similar findings. Since there is currently no good human system besides patient material to study POLG mutations, the authors intent to introduce their iPSCs as a model to study disease mechanisms.

Overall the paper is well written and a human model for POLG mutations would be very useful for the community, but the manuscripts value is severely impacted by several (and in my opinion too many) major technical drawbacks:

1. Although the authors generate two independent iPSC lines with different mutations, they fail to compare those to appropriate isogenic controls. The currently used controls are one iPSC line

generated from a completely unrelated fibroblast line from ATCC, which is not matched to patient lines in any way, as well as two ESC lines. Comparing to such unmatched and unrelated controls is clearly no longer state of the art and can severely affect the reliability of the phenotypes. Observed differences could either be due to the POLG mutations or some genetic background artefact in patient or control lines. The current state of the art is to compare to gene-corrected isogenic controls. If this is not possible for a specific reason (that needs to be clearly stated), comparison of multiple independent patient lines showing the same phenotypes to age-/gender-etc. matched lines, preferentially from unaffected siblings, is required. In case of loss of function, a well-executed rescue experiment would also be conceivable.

Answer: We fully agree with the reviewer that gene-corrected isogenic control iPSC lines would be of great interest. We acknowledge that genome editing techniques such as CRISPR/Cas can repair disease alleles, however, current understanding also raises questions concerning how off-target effects may limit the applications of gene-editing by creating off-target mutations with unknown consequences (Aryal, Wasylishen et al. 2018). In our study, we could not predict how off-target effects would impact mitochondrial function and disease phenotypes and therefore we chose to use differentiated iPSCs generated from a healthy control. Since this has been the accepted method for disease comparison over many decades, we still feel it is relevant here.

In the present study, we show that the previously identified cellular phenotypes (such as mitochondrial DNA deletion and mitochondrial complex I deficiency, shown in the 2 figures below), reported by our group using postmortem tissue (Tzoulis, Tran et al. 2014) are faithfully reproduced in this iPS cell model. This has never been shown before and indicates that we are able faithfully to model the disease by comparing our patient specific lines to a control generated from normal healthy individual. To confirm our NSCs do recapitulate the disease phenotypes, we have also investigated iPSC derived DA neurons and postmortem brain tissue from the patients involved in the present study. We could demonstrate the defective complex I level and mtDNA depletion, consistent with the findings from iPSC derived NSC model.

In the revised version, we have included one more control NSCs line generated from an iPSC line derived from AG05836 fibroblasts (RRID: CVCL_2B58) and performed several assays of total MMP, specific MMP, MitoSox ROS and mtDNA copy number.

Based on our data, together with the additional new data from another control, we believe that our controls are sufficiently reliable to be used as comparisons for the phenotypes seen in our NSC model system.

In this context, the different expression levels of SSEA4 and TRA160 shown in Figure 1 already indicate differences in line behavior at high level, so there could be more issues under the hood.

Answer: Please refer to the answers described above in reviewer #1, question #1.

2. The second major drawback is the use of retroviral reprogramming, which was state of the art 10 years ago and has been long replaced by non-integrating reprogramming methods (Sendai, mRNA, episomal...).

Earlier reports of iPSCs with POLG mutations used Sendai (Zurita et al. 2016, Chumarina et al. 2019), I therefore do not understand why this was not done here. Integration of retroviral DNA can mutate genes, leading to unspecific phenotypes that appear to be caused by the pathogenic mutations. This drawback, together with the lack appropriate isogenic controls, makes the whole study unreliable.

Answer: We agree that several advanced integration free reprogramming methods, such as plasmid, Sendai, mRNA, and episomal systems have been developed and in fact we have now adopted these methodologies to derive new iPSCs in our lab (both Sendai and mRNA). At the time we made these lines, however, retroviral approaches were the only options available. There are advantages and disadvantages for each system. Integration of viral DNA provides an easy method with good efficiency (Mahendra S. Rao and Nasir Malik, 2012) and it is validated for multiple cell types. Non-integrate reprogramming methods show lower efficiency, are validated for only one cell type, and technically challenging. However, it should be stressed that we have fully characterized our reprogrammed iPSCs using multiple approaches, including measurement of the expression of pluripotent transcription factors POU5F1, NANOG and pluripotent surface markers SSEA4, TRA-1-60 and TRA-1-81 using both immunostaining and flow cytometric analysis. We used RT-qPCR analysis to show the mRNA expression levels in terms of LIN28A, POU5F1, SOX2 and NANOG for all the clones. We further validated that our reprogrammed iPSCs retained the potential to differentiate into cell types associated with all three germ layers. While we agree that there will always be technical drawbacks with any reprogramming system, we still feel our study made a solid interpretation of the experimental evidence provided and this critic is somewhat unwarranted.

3. Why did the authors generate 3 controls, but compare only to one of them in their assays?

Answer: As described in the manuscript, we generated two or three individual clones for each line, including control and the two POLG mutants. The figures presented in the manuscript are combined data which were generated from at least three individual lines with more than three biological replicates. As showed below in Figure C, mtDNA copy number was determined from NSCs derived from individual iPSC clones in different lines, including two clones from control 1 (Detroit 551), one clone from control 2 (AG05836), three clones from WS5A patient and two clones from CP2A patient. Figure D was demonstrated in the present manuscript after combined the different clones for each line. All the information was described in figure legends.

Figure C

Figure D

4. I am not a big fan of the quality of images and graphs. Most of them are quite blurry, especially in Fig. 1. The main Figures are unlabeled, making it hard to keep track. Nomenclature of supplementary figures is misleading (SI vs. EV)

All our images were taken using the Leica TCS SP5 or SP8 confocal microscope (Leica Microsystems, Germany). We believe our pictures included in the present manuscript reach the high quality for publication. The issues reviewer raised may cause from the conversion of another format, for example the PDF. We show one example for our original picture below in Figure E. We have corrected the nomenclature of supplementary figures in revised manuscript.

Figure E

5. EB-based NSC differentiation is no longer state of the art, current protocols utilize differentiation in fully defined media and attached in monolayer, which is much less prone to heterogeneity of the NSCs.

Answer: Our current protocol is based on a modified dual SMAD protocol, which allows us to differentiate iPSCs directly into neural epithelial cells, without going through EB stages. We noticed the fact of heterogeneity of the NSCs generated from iPSCs in our research work as it is widely accepted. However, as reviewer mentioned, the heterogeneity of the NSCs is less in our study and minimized by including multiple clones for each line.

In this context, what the purity of NSCs? Nestin is high, but this is broadly expressed in many NSCs. What percentage of cells expresses Pax6, and other markers, such as FoxG1 and Sox2, how variable is this over multiple differentiations? Showing a low variability of differentiated cells is crucial to be able to see significant phenotypic differences. Many of the bar graphs show two or three populations, which could indicate heterogeneity.

Answer: Yes, we provide evidence to show the percentage of PAX6 and SOX2 positive populations in our NSCs by flow cytometric analysis, as showed in Figure F. In addition, we have also demonstrated the mRNA expression for *NESTIN*, *PAX6* and *SOX2*, as well as protein level in *NESTIN* and *PAX6*, as showed in Figure 3 in our manuscript and below as well.

As correctly pointed by reviewers that there is heterogeneity and variability over multiple differentiations. We accept that this is in fact one difficulty in using iPSCs for disease modeling. In our study, we have considered and applied two approaches to minimize the variability of differentiated cells. As we noticed that phenotypic differences may appear during passaging, only NSCs with certain passaging number P4-P9 were used for all the data presented in the paper. In addition, we also include multiple clones for each individual line.

6. Fig. 4e: GFAP is not sufficient as a marker to demonstrate Astrocyte identity, as it is also expressed in NSCs. At least some other AS markers, including S100B, GLT, GLAST should also be shown to make this conclusion. Similarly, GALC is not sufficient as a marker to demonstrate Oligodendrocyte identity. At least some other OL markers, including O4 and MBP should be shown.

Answer: As suggested by the reviewer, we have included S100 β , EAAT1 (GLAST) and glutamine synthesis for astrocytes and OLIG2, MBP for oligodendrocytes in revised manuscript, as showed below as well.

Figure EV6-8 in revised manuscript

7. Fig. 6h: TH and MAP2 stainings appear very poor, making successful DA neuron differentiation questionable. Both stainings appear to stain most cells in the images, with some more strongly stained cells present. Both the strongly and weakly stained cells appear quite immature in their morphology, with a relatively flat cell body and broad neurites that look more like processes from radial-glia-like NPCs. I believe that most of this staining is background. The authors need to do a better job of proving that they indeed generate high amounts of pure dopaminergic neurons, and that they are mature, e.g. by functional assays.

Answer: We understand the concern from the reviewer, however, we feel that we do have sufficient proof to show that our iPSC-derived DA neurons are mature and functional.

Figure F

Figure G

As showed above in Figure F, after 28 days' differentiation and two months maintenance, longer axon formation was observed (Figure F, a). At least 2 biological replicates for each clone were successfully differentiated into DA neurons and no obvious morphological difference was observed between DA neurons derived from control and patient cells (Figure

F, a). Disease and control lines were differentiated simultaneously to assess the efficiency of DA neuron yield and to minimize the differentiation bias which may influence the disease modeling. Immunocytochemistry with a panel of markers for specific DA neural lineage confirmed iPSC-derived DA neurons expressed the TH and MAP2 (Figure F, b). Flow cytometric analysis showed that, DA neurons derived from two POLG patients or control iPSCs lines demonstrated high ($\geq 80\%$) and comparable yields of TH (Figure F, c&d), suggesting suitability for in vitro disease modeling. We further confirmed that the POLG-DA neurons generated from patients-iPSCs maintained the same POLG gene profile of the parental cells by demonstrating that they retained the same mutation as the original fibroblasts (Figure F, e).

As shown above in Figure G, we identified that iPSC-derived DA neurons displayed the co-positive expression and co-localization of Synaptophysin, PSD-95 and MAP2 and the neurons appeared to form synaptic connections (Figure G, a). To determine the physiological phenotype of iPSC-DA neurons, we performed whole-cell current clamp recordings ($n = 10$) (Figure G, b), showing electrophysiological properties (Figure G, c-e).

8. Fig. 6k: mtDNA copy number is only (barely) significantly changed in one of the two iPSC lines, but not the other one. This makes me wonder how reproducible and robust this phenotype actually is. Also, there is clearly two populations in the control with only the upper one significantly changed.

Answer: We acknowledge the reviewer's concern regarding the reproducibility and robustness of mtDNA copy number phenotype. As we described above for the answers in Reviewer #2, question #3, the final figures in the manuscript were generated using a combination of the values from individual clone in each line. We agree that there is heterogeneity in each clone, however, we can still detect the changes between control and mutant from individual clone (as shown above in Figure C). In addition, and importantly for this question, we also observed lowered mtDNA levels using an indirect method based on flow cytometric measurement of TFAM. Methodologically, this is of major interest as it suggested that we could use this in live cells as a surrogate measure of mtDNA level. Therefore, we believe our data is reproducible and robust.

Is there a reason why in this and many other assays only one of the two patient lines shows an effect?

Answer: In our manuscript, we demonstrated the same disease phenotypes for both patient lines, including loss of mtDNA and complex I, neuronal ROS overproduction, increased cellular senescence, as well as disturbed NAD⁺ metabolism. Interestingly, we found activated mitophagy only in cells with compound heterozygous POLG mutations (CP2A patient), which may relate to the different type of mutation. We accept that more underlying mechanisms need to be addressed in future experiments.

9. Fig. 7c: The variation in complex I in NSCs is very large, I am wondering how this can be 4-star significant, when at least half of the measurements of control are at the same level than the mutants.

We acknowledge the fact that the variation in complex I in NSCs is large and there are two outliers in control lines (red row in Figure H, a). After removing these two outliers, we show the 3/4-star significant using GraphPad Prism software (Prism 7.0, GraphPad Software, Inc.) by Mann-Whitney U test.

Figure H

10. Text and Fig. 8: NAD result are inconsistent between NSCs to iPSCs, making this whole section hard to understand.

Answer: Our data showed differences between NSCs to iPSCs in relation to expression of disease phenotype. As showed in the manuscript, we found Complex I loss in patient NSCs but not in iPSCs from the same patient. Complex I function is essential for the re-oxidization of NADH and maintenance of the NAD^+/NADH ratio, an important indicator of redox status and major modulator of intermediary metabolism (Houtkooper, Canto et al. 2010). As expected, we found a significant decrease in the NAD^+/NADH ratio in both patient NSCs compared with control (Fig. 8A). While the levels of NAD^+ and NADH varied in the patient NSCs (Fig. 8B & C), the ratio showed disturbed NAD^+ homeostasis in both. We confirmed these finding using a commercial NAD^+/NADH assay. In keeping with the absence of complex I loss, changes in NAD^+/NADH were not seen POLG iPSCs (Fig. 8D-F).

11. Fig. 8k: Band intensities of WS5A-NSCs in Phospho-SirT1 blot do not fit to quantification. First band is much stronger than in Ctrl, two other bands weaker than CTRL. This high variation is not reflected in the error bar in L-b.

Answer: The high variation is not reflected in the quantification, due to the removal of this value which is detected as an outlier using GraphPad Prism software.

Some more suggestions to improve the impact of the manuscript:

1. Most assays are done in NSCs, but not in much more disease-relevant cell types, such as DA or cortical neurons. I think it would make the paper much stronger if clear phenotypes were shown in cell types present in the adult human body and affected by the disease.

Answer: We would beg to disagree with this point. We have succeeded in differentiating patient-iPSCs into both NSCs and dopaminergic (DA) neurons, however, as is well-known, neurons are harder to acquire and display limited expansion potential. As long as they express the disease phenotype, NSCs provide a better tool for studying disease mechanism and have a much greater potential for drug screening being easier to culture, maintain, freeze and refreeze, as well as count. Further, we can perform assays of mitochondrial function in “live cells”.

In addition, primary NSCs provide a continued source of neurons and glial cells in the brain that further serve as a foundation for development, repair, and functional modulations of human adult neurogenesis. It is, therefore, not surprising that dysfunction of NSCs contribute to an assortment of neurological disorders (Li, Chao et al. 2018). Since NSCs can be easily obtained from iPSCs, they provide an alternative model to primary NSCs with the possibility of application in disease-relevant phenotype studies and in drug development for disease therapy that target the specific phenotypes identified. Furthermore, we provide in our study evidence to show that NSCs recapitulate the disease phenotypes as shown in patient

specific DA neurons generated from iPSCs and patient postmortem brain tissue. Therefore, we are quite convinced that NSCs provide a good tool to study POLG related diseases. In addition, since NSC can be grown in large numbers in smaller formats, we hope to use this system to establish a high-throughput screening system in order to identify therapies for these devastating diseases.

2. It would be very interesting and a great opportunity to address selective vulnerability of affected and spared tissues by studying iPSC-derived cell types affected vs. unaffected in disease.

Answer: We are not sure what this reviewer means by unaffected cell type. We performed the analysis and showed the selective vulnerability of NSCs and DA-neurons and compared these to fibroblasts that were unaffected by disease. If the reviewer is suggesting making comparisons to cell types that do not express selective vulnerability to POLG mutation, we would have to ask which cell types are meant. We have looked at many tissues and find that *POLG* mutation leads to cellular defects in all, despite the fact that some do not express this clinically.

3. Performing stress treatments e.g. on DA neurons could be very interesting to elicit more compelling phenotypes and also see variations in vulnerability of mutant cells.

Answer: We fully agree that performing stress treatments e.g. on either NSCs or DA neurons could be very interesting to elicit more compelling phenotypes. We have started to treat the cells using different compounds in a further project, but we believe it is also very important to understand first the pathogenic mechanisms of POLG mutations, which is the main aim of the present study. Besides, to our knowledge, our studies are the first that show it is possible to recapitulate the neuronal molecular and biochemical defects associated with POLG mutation in a human stem cell model.

4. I would also suggest addressing why the neuronal lineage is particularly prone to loss of mitochondrial copy number.

Answer: This is clearly a question that we wish to address. From our own and other studies (often on postmortem tissues), we know that neurons develop mtDNA depletion as do hepatocytes. Skeletal muscle cells, however, do not manifest depletion, but instead show multiple deletions. Being able to demonstrate that we can recapitulate the same phenotype seen in PM neurons in a tractable cell-based model and showing how these cells respond with loss of ATP, increased ROS etc., is, we feel, a major step forward and one that should enable us to explore further the question raised by the reviewer.

Referee #3 (Remarks for Author):

This is an interesting study reporting the generation of iPSCs carrying *POLG* mutations. These iPSCs could represent a useful tool to advance our understanding of the pathogenic mechanisms of *POLG* mutations.

Unfortunately however it is this reviewer's opinion that the level of insights into disease mechanisms is very limited in this study. There are no insights into the reason why *POLG* mutations affect neurons specifically.

Answer: We disagree with this reviewer concerning the lack of insights generated by our studies. We are the first that show that it is possible to recapitulate the biochemical and molecular findings in neural stem cells associated with the common *POLG* mutations. Further, we believe that our studies demonstrate that iPSC-derived neural stem cells can provide a robust model system in which to study tissue specific mitochondrial disease manifestations. Our studies suggested that *POLG* mutations result in a compromised ETC, leading to both a reduction in OXPHOS (and thus, less ATP), redox changes as well as increased levels of ROS (which “leak” out as a result of the less efficient ETC). This, in turn, can further damage the mitochondria, leading to a vicious feedback loop that often results in neural cell loss. Other pathogenic mutations interfere with the critical processes of mitochondrial mitophagy (the autophagic disposal of redundant or damage mitochondria), with potentially devastating consequences for the patient. Lastly, we show, for the first time that pathways including cellular senescence and mitophagy are activated by the presence of *POLG* mutation and are likely contributors to the neuronal death seen in these diseases. Thus, we believe that our studies may provide important insights into disease mechanisms.

I also could not see clear experiments dissecting the mechanisms underlying the neuronal cell loss.

Answer: As clearly demonstrated in the manuscript, we have performed plenty of experiments to dissect the mechanisms underlying the neuronal cell loss, as highlight below,

- **Human iPSCs carrying *POLG* mutations can be differentiated into high-yield neural stem cells.**
- **Both fibroblasts and iPSCs from *POLG* patients manifested only a partial disease phenotype.**
- **NSCs with disease caused by *POLG* mutations showed energy failure and mtDNA depletion, similar to findings iPSC-derived DA neurons.**
- ***POLG* NSCs recapitulated the disease phenotypes observed in *POLG* patient post-mortem tissues.**
- ***POLG* NSCs showed loss of mitochondrial complex I and abnormal UCP2/Sirt1 mediated NAD⁺ homeostasis associated with overproduction of intercellular and mitochondrial ROS.**
- **Elevated ROS triggered cell senescence and BNIP3-mediated mitophagy, which contributes to pathological mechanisms in mitochondrial diseases.**

We also made a clear model to describe how POLG mutations cause the defect in mitochondrial activity and mitophagy, leading to the neural cell loss, as showed in Figure 9 in the revised manuscript and answer to reviewer#1, question #5.

Finally there is no suggestion of potential therapeutic strategies.

Answer: As suggested by reviewer, we suggested the potential therapeutic strategies in POLG-related diseases based on our present studies. As described in revised manuscript " Our studies indicate further that the health potential of targeting NAD⁺ homeostasis will inform clinical study design to identify nutraceutical approaches for combating POLG disease."

8th Apr 2020

Dear Dr. Liang,

Thank you for the submission of your manuscript to EMBO Molecular Medicine and for asking us to reconsider our decision. Thank you also for already making some changes in your article and providing a point-by-point response.

After discussing with my colleagues, we agreed to inquire with an external advisor whether your paper should be further considered in EMBO Molecular Medicine. I am happy to report that this advisor was positive about the study and recommended revision.

Please see below some of our advisor's comments:

"I found this paper very interesting indeed, as it demonstrates that neuronal derivatives of Patients' mutant iPSC do display a number of features essential to explain the neuronal specific impairment typical of the disease associated with specific POLG mutations. This is an important achievement that can be extended in the future to other mutations of POLG associated with diverse phenotypes (e.g. encephalomyopathy, PEO). Mouse models of these mutations have so far been disappointing at least for the clinical phenotype, which prevents also their use to test treatment strategies in vivo. Therefore, I think that this is a valuable paper that should appear in EMBO Mol Medicine, although some of the comments (not all) from the reviewers are reasonable and should be answered. I have a couple of minor questions myself.

- 1) the authors do not explain why the membrane potential is increased instead of being decreased in iPSC cells, in spite of the reduction in ATP levels.
- 2) the phenotypes of POLG mutations are not limited to hepatocerebral failure, but encompass a wide spectrum of presentations, including myopathy with PEO and accumulation of multiple mtDNA deletions, it will indeed be interesting to see what happens to mtDNA in muscle cells generated from iPSC from patients with the appropriate mutations.
- 3) Finally, I found the discussion on POLG mutations and aging a little too speculative. It would be interesting, if time allows the authors to do it, to see whether the apparent impairment of Sirtuin1 is directly due to the reduction of NAD⁺/NADH, for instance by exposing the cell cultures to nicotinamide riboside, and obligate precursor of NAD⁺."

Therefore, given these comments and advice, we would like to invite you to further revise your article. Please consider our advisor questions and review your statistical analyses, eventually consulting with a statistician. Removing outliers from an analysis without having pre-established the criteria is not an acceptable process. Similarly, making a bar plot and providing error bars when 1 or 2 samples are used is questionable.

We feel that we can consider a revision of your manuscript if you can further address these issues including our advisor's questions. Please note that it is EMBO Molecular Medicine policy to allow only a single round of revision and that, as acceptance or rejection of the manuscript will depend on another round of review, your responses should be as complete as possible.

EMBO Molecular Medicine has a "scooping protection" policy, whereby similar findings that are

published by others during review or revision are not a criterion for rejection. Should you decide to submit a revised version, I do ask that you get in touch after three months if you have not completed it, to update us on the status.

I look forward to receiving your revised manuscript.

Yours sincerely,

Celine Carret

Celine Carret, PhD
Senior Editor
EMBO Molecular Medicine

*** Instructions to submit your revised manuscript ***

** PLEASE NOTE ** As part of the EMBO Publications transparent editorial process initiative (see our Editorial at <https://www.embopress.org/doi/pdf/10.1002/emmm.201000094>), EMBO Molecular Medicine will publish online a Review Process File to accompany accepted manuscripts.

Advisor's comments:

"I found this paper very interesting indeed, as it demonstrates that neuronal derivatives of Patients' mutant iPSC do display a number of features essential to explain the neuronal specific impairment typical of the disease associated with specific POLG mutations. This is an important achievement that can be extended in the future to other mutations of POLG associated with diverse phenotypes (e.g. encephalomyopathy, PEO). Mouse models of these mutations have so far been disappointing at least for the clinical phenotype, which prevents also their use to test treatment strategies in vivo. Therefore, I think that this is a valuable paper that should appear in EMBO Mol Medicine, although some of the comments (not all) from the reviewers are reasonable and should be answered. I have a couple of minor questions myself.

1. the authors do not explain why the membrane potential is increased instead of being decreased in iPSC cells, in spite of the reduction in ATP levels.

Answers: We would first thank the advisor for their positive comments concerning the importance of our achievement in generating iPSC derived neural stem cells that display *“the neuronal specific impairment typical of the disease associated with specific POLG mutations”*.

The comment concerning membrane potential is very interesting. We would point out, however, that we showed that the mitochondrial membrane potential (MMP) was increased in fibroblasts, not in iPSCs. In Figure 1 and 2 reproduced below, we show ATP depletion in both POLG iPSCs (Figure 1, a) and fibroblasts (Figure 2, a), however, an elevated MMP was seen in fibroblasts (Figure 1, b) not in iPSCs (Figure 1, c).

We discussed briefly a possible explanation for an elevated MMP in POLG fibroblasts in the Results (paragraph 3) and Discussion (paragraph 2). There appears to be three possibilities: the one we mention in our text is hyperpolarization of the mitochondrial inner membrane. This was suggested in a similar study looking at iPSC with an ATPase mutation (Lorenz et al., 2017). Another possibility is that patient fibroblasts are in a hypermetabolic state. This suggestion was made in a study looking at fibroblasts from patients with amyotrophic lateral sclerosis (Konrad et al., 2017). In this study, ALS patient fibroblasts showed a higher mitochondrial membrane potential with no concomitant rise in ATP (but no fall). It was suggested that the hypermetabolic state in ALS reflected an upregulation of respiration and glycolysis as an adaptation to relative energy depletion. Since we showed that ATP levels fall in our iPSC lines, this is either not relevant in POLG disease or the energy depletion is so severe that it exceeded the ATP generating capacity. Lastly, similar to what was seen in fibroblasts from Alzheimer's disease (Sonntag et al., 2017), POLG fibroblasts may switch even more to glycolysis, which in turn, leads to a decrease in ROS, as showed in our manuscript and Figure 3 below. We also have added this explanation in Discussion (paragraph 2).

Figure 1

Figure 2

Figure 3

2) the phenotypes of POLG mutations are not limited to hepatocerebral failure, but encompass a wide spectrum of presentations, including myopathy with PEO and accumulation of multiple mtDNA deletions, it will indeed be interesting to see what happens to mtDNA in muscle cells generated from iPSC form patients with the appropriate mutations.

Answer: We thank the advisor for pointing out this out. We are aware from our own clinical studies that muscle and brain/liver respond differently to the presence of *POLG* mutations and agree entirely that it would be very interesting to investigate muscle cells from patients with appropriate mutations. Differentiating iPSCs into specific cell lineages is, however, a major undertaking and currently our group has been working for several years to establish robust iPSC based neuronal models including NSCs, dopaminergic neurons, astrocytes and motor neurons. We are developing a cardiomyocyte *POLG* model which so far has not shown mtDNA deletions, but this is only after 1-2 months in culture. It is worth pointing out that mtDNA deletions accumulate slowly in somatic cells such as neurons and myocytes and it may, therefore, require months or indeed years before they are seen in culture, if at all.

3) Finally, I found the discussion on *POLG* mutations and aging a little too speculative. It would be interesting, if time allows the authors to do it, to see whether the apparent impairment of Sirtuin1 is directly due to the reduction of NAD⁺/NADH, for instance by exposing the cell cultures to nicotinamide riboside, and obligate precursor of NAD⁺."

Answer: We take the point that the discussion was speculative and did not mean to suggest a proven link. Our data raises the possibility that an altered NAD⁺/NADH ratio could influence cell survival, and while this could be through sirtuin1, we also mention the possibility that this could be through ROS overproduction. Similar to the phenotypes in NSCs, *POLG* mutation drives also mitochondrial dysfunction which lead to ROS overproduction in Dopaminergic (DA) neurons derived from the same patients iPSCs, we have treated DA neurons with N-acetylcysteine amide (NACA) and found that it ameliorates mitochondrial dysfunction and inhibits oxidative damage, as demonstrated below in Figure 4. Since we have included this finding in other manuscript, we will not present it in present manuscript.

In addition, we planned to treat our cells (NSCs and astrocyte which showed a similar phenotypes) with nicotinamide riboside (NRs) to see if this rescued the mitochondrial defects and reversed reduction of the NAD⁺/NADH ratio. Unfortunately, the COVID-19 enforced closure of our university meant that this experiment had to be stopped.

Editor:

Therefore, given these comments and advice, we would like to invite you to further revise your article. Please consider our advisor questions and review your statistical analyses, eventually consulting with a statistician. Removing outliers from an analysis without having pre-established the criteria is not an acceptable process. Similarly, making a bar plot and providing error bars when 1 or 2 samples are used is questionable.

Answer: We have asked a statistician, Jessica Furriol Palmer, to re-analyze all of our data with SPSS software (SPSS v.25, IBM). We have also provided all the bar plot and error bars using at least 3 samples. In the revised manuscript, we have renewed all the Figure with new mean, SEM and p-value after removing the outliers testing the distributions for normality using the Shapiro-Wilk test. In the revised Figure, Mann-Whitney U test was used to assess statistical significance for variables with non-normal distribution, while two-sided Student's t-test was applied for normal distributed variables. Please also see the revised manuscript in Materials and Methods section and Figure legends.

Referee #1:

Liang et al reports the modeling the POLG diseases using human iPSCs and their neural derivatives. POLG is one of catalytic subunit of DNA polymerase gamma that catalyze the replication and repair of mitochondrial DNA. POLG mutations are known to cause diverse phenotypes and investigating the role of POLG is critical to understand the diseases and to

develop potential therapeutics. Authors derived iPSC lines from patients with homozygous POLG mutation (WS5A), or compound heterozygous mutation (CPA2A). Comparison of mitochondrial numbers, gene expression, and functions were made among different types of cells from the iPSCs. Authors found that neural stem cells (NSCs) and dopaminergic neurons (DA) displayed a major phenotype in mitochondrial functional defect. Overall, the manuscript is well written, and logical and data is strong. However, there are some concerns to address before the publication.

1. There are some expression data that were presented in a non-conventional way. In Figure 1C, the expression of Sox2 in different clones is presented strangely. How the SOX2 level was normalized against which samples is not clear. Please describe how the fold change was calculated or make correction. When characterizing the pluripotency of the iPSC lines, authors presented the median intensity of markers, SSEA4 and TRA160/TRA181. Since it is well known that the different clones express different levels of the markers, it is advised that authors remove the data and rather show the immunostaining data and FACS data showing the presence of the markers.

Answer: We thank the reviewer for this point. As suggested, we have removed this data and instead show immunostaining data that identifies the presence of SOX2 (showed in the Figure 5 and below. In the present manuscript, the average CT values of three technical replicates were normalized to the geometric mean of endogenous control gene, Actin Beta (ACTB: Hs01060665). The expression of the iPSC markers was assessed with fold change by using the comparative $\Delta\Delta C_t$ method and normalizing to the gene level from ESC1. The expression of the NSC markers was assessed with fold change by using the comparative $\Delta\Delta C_t$ method by normalizing NSCs to iPSCs, as described in the Materials and Methods section.

Figure 5

We agree with the point that it is well known that different clones express different levels of the markers we described that clonal variations for the protein level and mRNA expression were noticed (Fig. 1C, D, a; E, a; SI3A, a; SI3B, a and SI3C, a). In order to minimize the phenotypic diversity caused by intra-clonal heterogeneity, multiple clones were included in the further analysis. When characterizing the pluripotency of the iPSC lines, we measured the expression of pluripotent transcript factors POU5F1 (Figure 1E), Nanog (Figure 1F) and pluripotent surface marker markers SSEA4, TRA-1-60 and TRA-1-81 using flow cytometric analysis. Using this technique, we observed lower levels of the three pluripotent surface markers - SSEA4, TRA-1-60 and TRA-1-81 (Figure S1B) in both WS5A and CP2A iPSCs as compared to the 2 ESCs and control iPSC line. However, no change was observed in the expression of POU5F1 (Figure 1F) or Nanog (Figure S1C). Although the mechanisms for those findings need to be investigated in the future, we believe this phenotype may related to the mutation. Therefore, we still keep them in the revised manuscript.

2. It is understandable that authors derived multiple iPSC lines from the patients of POLG mutations. However, when the data was presented it is not clear which cell lines were used. Authors should describe that in figure legends, or if adding the information in the legend is not practical, authors make a supplementary table to display the iPSC or hESC clones that were used for the given experiments. It is not good to describe, "...A-C are generated from two distinct ESC lines and 2 or 3 different clones of each line..."(Legend Figure 4BC). "...NSCs generated from Detroit 551, WS5A, and CP2A iPSCs..." (Legend Figure 5C).

Answer: We thank to the reviewer for the suggestion. We have made a supplementary table (Appendix Table S1) to display the iPSC or hESC clones that were used for the given experiments and we have revised the figure legends with more details on data information. Please find the details in the revised manuscript Appendix Table S1 and figure legends.

3. In neuronal differentiation, there seem to be a difference between ESC and other iPSCs lines (Figure 4C). If there is not, authors should describe that there is no significant difference. In addition, the experiments on comparison between POLG mutant and wild type, ESC data was not presented and only one control iPSCs were used. If the control line itself is outlier, the whole conclusions in the data are not significant. If it is too much work to repeat all the experiments on NSC and DA cells from other control iPSC lines or ESC lines, authors may perform the experiments on fibroblasts, such as Figure 5. The findings will support one of major conclusion of the manuscript that POLG mutations have cell type specific effect on mitochondrial activity without mitochondrial DNA replication and numbers.

Answer: As rightly pointed by the reviewer, we do indeed see a difference between ESC and other iPSCs lines during neuronal differentiation. As showed below in Figure 6 for example, there is significant higher specific TFAM (total TFAM/TOMM20) in control iPSC-derived NSCs compared to ESC-derived NSCs using flow cytometric analysis. Therefore, in the experiments on comparison between POLG mutant and wild type, ESC data was not presented and only control iPSCs was used.

In the revised version, we have included two more control NSCs line from iPSC line derived from CCD-1079Sk (control 2, ATCC® CRL-2097™, human normal new-born male fibroblast) and AG05836 (control 3, RRID: CVCL2B58, 44 years-old female fibroblasts). We also performed assays of total MMP, specific MMP, MitoSox ROS and mtDNA copy number for both of the new controls, see below Figure 7 (Orange arrows display the new added lines) and measurement of the specific complex I, II and IV for AG05836 (control 3), see below Figure 8 (Orange arrows display the new added lines). By adding more control lines. we have confirmed our main findings of the manuscript which showing that POLG mutations have effect on mitochondrial activity and mitochondrial DNA replication and numbers, as clearly showed in revised Figure.

Figures 7

Individual clones

Combined as groups

Figures 8

Individual clones

Combined as groups

4. Some data need quantification and perform statistical analysis. In Figure EV10, the staining of the Mtpagy Dye does not support any conclusion. Authors need to take random pictures of the slides and quantify to examine whether there is statistically significant difference between normal and POLG mutant NSCs.

Answer: As mentioned by the reviewer, the staining of the Mtpagy Dye does not support any conclusion and we believe that the Western Blotting data are sufficient to provide the evidence of mitophagy, therefore we have deleted the images of Mtpagy Dye in the revised manuscript.

5. Authors should include a model to describe how POLG mutations cause the defect in mitochondrial activity and mitophagy at one of the last Figure.

Answer: We thank to the reviewer for bringing up the idea and we have included a model to describe how POLG mutations cause the defect in mitochondrial activity and mitophagy at the end in the revised manuscript, see the Figure 9 below as well. From our earlier studies, we know that POLG mutations lead neuronal mtDNA depletion and, given sufficient time, point mutations and deletions. Our current studies confirm that there is loss of ATP and membrane potential that we believe leads to changes in redox potential and excess ROS production. We also find activation of mitophagy. Our hypothesis is that neuronal death occurs by two mechanisms: firstly, the initiation of seizures in already stressed neurons exceeds the capacity of these neuron to maintain cellular energy production and thus cellular integrity and leads to acute neuronal death. Secondly, the loss of ATP with changes in redox potential and ROS production, lead to neuronal dysfunction that eventually leads to chronic neuronal loss.

Figure 9 in revised manuscript

Referee #2 (Comments on Novelty/Model System for Author):

The value of the model is severely impacted by several major technical drawbacks. Please see my detailed explanation in my review, especially point 1 and 2 about isogenic controls and use of retrovirus for reprogramming.

Referee #2 (Remarks for Author):

The manuscript by Liang et al. describes the generation of iPSCs from patients with pathogenic POLG mutations and the characterization of these iPSCs and differentiated cells of the neuronal lineage. The POLG enzyme replicates mitochondrial DNA, and mutations in patients cause a variety of symptoms in different tissues. Neurological defects are particularly common in mutation carriers. The authors describe some potentially disease-related phenotypes in neural stem cells derived from patient iPSCs related to mitochondrial dysfunction, which are novel, as previous reports dealing with POLG mutations in iPSCs did not report similar findings. Since there is currently no good human system besides patient material to study POLG mutations, the authors intent to introduce their iPSCs as a model to study disease mechanisms.

Overall the paper is well written and a human model for POLG mutations would be very useful for the community, but the manuscripts value is severely impacted by several (and in my opinion too many) major technical drawbacks:

1. Although the authors generate two independent iPSC lines with different mutations, they fail to compare those to appropriate isogenic controls. The currently used controls are one iPSC line generated from a completely unrelated fibroblast line from ATCC, which is not matched to patient lines in any way, as well as two ESC lines. Comparing to such unmatched and unrelated controls is clearly no longer state of the art and can severely affect the reliability of the phenotypes. Observed differences could either be due to the POLG mutations or some genetic background artefact in patient or control lines. The current state of the art is to compare to gene-corrected isogenic controls. If this is not possible for a specific reason (that needs to be clearly stated), comparison of multiple independent patient lines showing the same phenotypes to age-/gender- etc. matched lines, preferentially from unaffected siblings, is required. In case of loss of function, a well-executed rescue experiment would also be conceivable.

Answer: We fully agree with the reviewer that gene-corrected isogenic control iPSC lines would be of great interest. We acknowledge that genome editing techniques such as CRISPR/Cas can repair disease alleles, however, current understanding also raises questions concerning how off-target effects may limit the applications of gene-editing by creating off-target mutations with unknown consequences (Aryal, Wasylishen et al. 2018). In our study, we could not predict how off-target effects would impact mitochondrial function and disease phenotypes and therefore we chose to use differentiated iPSCs generated from a healthy control. Since this has been the accepted method for disease comparison over many decades, we still feel it is relevant here.

In the present study, we show that the previously identified cellular phenotypes (such as mitochondrial DNA deletion and mitochondrial complex I deficiency, shown in the 2 Figure below), reported by our group using postmortem tissue (Tzoulis, Tran et al. 2014) are faithfully reproduced in this iPS cell model. This has never been shown before and indicates that we have generated a robust model of POLG disease. To confirm our NSCs do recapitulate the disease phenotypes, we have also investigated iPSC derived DA neurons and postmortem brain tissue from the patients involved in the present study. We could demonstrate the defective complex I level and mtDNA depletion, consistent with the findings from iPSC derived NSC model.

In the revised version, we have included one more control NSCs line generated from an iPSC line derived from AG05836 fibroblasts (RRID: CVCL_2B58), which is reprogrammed using Sendai viral system. We also performed several assays of total MMP, specific MMP, MitoSox ROS and mtDNA copy number.

Based on our data, together with the additional new data from another control, we believe that our controls are sufficiently reliable to be used as comparisons for the phenotypes seen in our NSC model system.

In this context, the different expression levels of SSEA4 and TRA160 shown in Figure 1 already indicate differences in line behavior at high level, so there could be more issues under the hood.

Answer: Please refer to the answers described above in reviewer #1, question #1.

2. The second major drawback is the use of retroviral reprogramming, which was state of the art 10 years ago and has been long replaced by non-integrating reprogramming methods (Sendai, mRNA, episomal...).

Earlier reports of iPSCs with POLG mutations used Sendai (Zurita et al. 2016, Chumarina et al. 2019), I therefore do not understand why this was not done here. Integration of retroviral DNA can mutate genes, leading to unspecific phenotypes that appear to be caused by the pathogenic mutations. This drawback, together with the lack appropriate isogenic controls, makes the whole study unreliable.

Answer: We agree that several advanced integration free reprogramming methods, such as plasmid, Sendai, mRNA, and episomal systems have been developed and in fact we have now adopted these methodologies to derive new iPSCs in our lab (both Sendai and mRNA). At the time we made these lines, however, retroviral approaches were the only options available. There are advantages and disadvantages for each system. Integration of viral DNA provides an easy method with good efficiency (Mahendra S. Rao and Nasir Malik, 2012) and it is validated for multiple cell types. Non-integrate reprogramming methods show lower efficiency, are validated for only one cell type, and technically challenging. However, it should be stressed that we have fully characterized our reprogrammed iPSCs using multiple approaches, including measurement of the expression of pluripotent transcription factors POU5F1, NANOG and pluripotent surface markers SSEA4, TRA-1-60 and TRA-1-81 using both immunostaining and flow cytometric analysis. We used RT-qPCR analysis to show the mRNA expression levels in terms of LIN28A, POU5F1, SOX2 and NANOG for all the clones. We further validated that our reprogrammed iPSCs retained the potential to differentiate into cell types associated with all three germ layers. While we agree that there will always be technical drawbacks with any reprogramming system, we still feel our study made a solid interpretation of the experimental evidence provided and this critic is somewhat unwarranted.

3. Why did the authors generate 3 controls, but compare only to one of them in their assays?

Answer: As described in the manuscript, we generated two or three individual clones for each line, including control and the two POLG mutant lines. The Figure presented in the manuscript are combined data which were generated from at least three individual lines with more than three biological replicates. As showed below in Figure 10, mtDNA copy number was determined from NSCs derived from individual iPSC clones in different lines, including two clones from control 1 (Detroit 551), one clone from control 2 (CCD-1079Sk), one clone from control 3 (AG05836), three clones from WS5A patient and two clones from

CP2A patient. Figure 11 was demonstrated in the present manuscript after combined the different clones for each line. All the information was described in figure legends.

4. I am not a big fan of the quality of images and graphs. Most of them are quite blurry, especially in Fig. 1. The main Figure are unlabeled, making it hard to keep track. Nomenclature of supplementary Figure is misleading (SI vs. EV)

All our images were taken using the Leica TCS SP5 or SP8 confocal microscope (Leica Microsystems, Germany). We believe our pictures included in the present manuscript reach the high quality for publication. The issues reviewer raised may cause from the conversion of another format, for example the PDF. We show one example for our original picture below in Figure 12. We have corrected the nomenclature of supplementary Figure in revised manuscript.

5. EB-based NSC differentiation is no longer state of the art, current protocols utilize differentiation in fully defined media and attached in monolayer, which is much less prone to heterogeneity of the NSCs.

Answer: Our current protocol is based on a modified dual SMAD protocol, which allows us to differentiate iPSCs directly into neural epithelial cells, without going through EB stages. We noticed the fact of heterogeneity of the NSCs generated from iPSCs in our research work as it is widely accepted. However, as reviewer mentioned, the heterogeneity of the NSCs is less in our study and minimized by including multiple clones for each line.

In this context, what the purity of NSCs? Nestin is high, but this is broadly expressed in many NSCs. What percentage of cells expresses Pax6, and other markers, such as FoxG1 and Sox2, how variable is this over multiple differentiations? Showing a low variability of differentiated cells is crucial to be able to see significant phenotypic differences. Many of the bar graphs show two or three populations, which could indicate heterogeneity.

Answer: Yes, we provide evidence to show the percentage of PAX6 and SOX2 positive populations in our NSCs by flow cytometric analysis. This is shown in Figure 13. In addition, we have also investigated the mRNA expression for *NESTIN*, *PAX6* and *SOX2*, as well as protein level in *NESTIN* and *PAX6*, as showed in Figure 3 in our manuscript and below as well.

Figure 3 in manuscript

As correctly pointed by the reviewer, there is heterogeneity and variability over multiple differentiations. We accept that this is in fact one difficulty in using iPSCs for disease modeling. In our study, we have considered and applied two approaches to minimize the variability of differentiated cells. As we noticed that phenotypic differences may appear during passaging, only NSCs with certain passaging number P4-P9 were used for all the data presented in the paper. In addition, we also include multiple clones for each individual line.

6. Fig. 4e: GFAP is not sufficient as a marker to demonstrate Astrocyte identity, as it is also expressed in NSCs. At least some other AS markers, including S100B, GLT, GLAST should also

be shown to make this conclusion. Similarly, GALC is not sufficient as a marker to demonstrate Oligodendrocyte identity. At least some other OL markers, including O4 and MBP should be shown.

Answer: As suggested by the reviewer, we have included S100 β , EAAT1 (GLAST) and glutamine synthesis for astrocytes and OLIG2, MBP for oligodendrocytes in revised manuscript, as showed below as well.

7. Fig. 6h: TH and MAP2 stainings appear very poor, making successful DA neuron differentiation questionable. Both stainings appear to stain most cells in the images, with some more strongly stained cells present. Both the strongly and weakly stained cells appear quite immature in their morphology, with a relatively flat cell body and broad neurites that look more like processes from radial-glia-like NPCs. I believe that most of this staining is background. The authors need to do a better job of proving that they indeed generate high amounts of pure dopaminergic neurons, and that they are mature, e.g. by functional assays.

Answer: We understand the concern from the reviewer; however, we feel that we do have sufficient proof to show that our iPSC-derived DA neurons have good purity, are mature and functional.

Figure 14

Figure 15

As showed above in Figure 14, we identified that iPSC-derived DA neurons are over 90% positive for both TH and DAT. They displayed the co-positive expression and colocalization of pre-synaptic marker Synaptophysin, post-synaptic marker PSD-95 and mature-neuronal marker MAP2. To determine the physiological phenotype of iPSC-DA neurons, we performed whole-cell current clamp recordings (n=10) and showed electrophysiological properties, as demonstrated in Figure 15 (A. Representative current traces (top) under depolarizing voltage steps (bottom) from a holding potential $V_h = -60$ mV. B. Representative potential traces (top) under depolarizing current steps (bottom)). Taken together, these evidence indicate proof that we have generated high amounts of pure DA neurons, and that they are mature and functional.

8. Fig. 6k: mtDNA copy number is only (barely) significantly changed in one of the two iPSC lines, but not the other one. This makes me wonder how reproducible and robust this phenotype actually is. Also, there is clearly two populations in the control with only the upper one significantly changed.

Answer: We acknowledge the reviewer's concern regarding the mtDNA copy number measurements. We agree that there is heterogeneity in each clone, however, if one compares the mtDNA copy number measurements performed in single neurons from postmortem brain, we still see variation and an overlap between levels in control and patients. Despite this, we are still able to detect the changes between control and mutant from individual clone (as showed above in Figure D). In addition, and importantly for this question, we also observed lowered mtDNA levels using an indirect method based on flow cytometric measurement of TFAM. Methodologically, this is of major interest as it suggested that we could use this in live cells as a surrogate measure of mtDNA level. Therefore, we believe our data is reproducible and robust.

Is there are reason why in this and many other assays only one of the two patient lines shows an effect?

Answer: In our manuscript, we demonstrated the same disease phenotypes for both patient lines, including loss of mtDNA and complex I, neuronal ROS overproduction, increased cellular senescence, as well as disturbed NAD⁺ metabolism. Interestingly, we did find activated mitophagy only in cells with compound heterozygous POLG mutations (CP2A patient), which may relate to the different type of mutation. We accept that more underlying mechanism need to be addressed in the future experiments.

9. Fig. 7c: The variation in complex I in NSCs is very large, I am wondering how this can be 4-star significant, when at least half of the measurements of control are at the same level than the mutants.

Answer: We acknowledge the fact that the variation in complex I in NSCs is large and there are two outliers in control lines (red row in Figure 16, a) based on the new SPSS analysis. After removed the outliers, 3-star ($p=0.000473$) and 4-star ($p=0.000017$) significance were identified in WS5A vs. CTRL group and CP2A vs. CTRL group, see below the Figure 16 and the table demonstrating the result from SPSS analysis.

Figure 16

Table

Cell	Outliers?	Number	Colu mn1	Colu mn2	Colu mn3
CTRL	Yes	2.12	2.06		
WS5A	no				
CP2A	no				

Cell	Distribution	Normal?
CTRL	0.004	no
WS5A	0.226	yes
CP2A	0.74	yes

	p-val (M-W)	Colu mn1
WS5A VS. CTRL	0.000129	sign
CP2A VS. CTRL	0.000006	sign

10. Text and Fig. 8: NAD result are inconsistent between NSCs to iPSCs, making this whole section hard to understand.

Answer: Our data showed differences between NSCs and iPSCs in relation to expression of disease phenotype. While iPSC did manifest some aspects of disease, it was only when differentiated further that we saw a more comprehensive disease phenotype. It is not unusual that different cell types manifest differently, and we see this often in mitochondrial disease, particularly when using glycolytic cells such as fibroblasts which tend not to manifest respiratory chain dysfunction. We found Complex I loss in patient NSCs, but not in iPSCs from the same patient. As explained above, this potentially reflects the metabolic differences between these two cell types and not any failure on our part to detect a phenotype. Once complex I deficiency is present, however, we would expect it to impact on

NAD metabolism. Complex I function is essential for the re-oxidization of NADH and maintenance of the NAD^+/NADH ratio, an important indicator of redox status and major modulator of intermediary metabolism (Houtkooper et al., 2010). As expected, we found a significant decrease in the NAD^+/NADH ratio in both patient NSCs compared with control (Fig. 8A). While the levels of NAD^+ and NADH varied in the patient NSCs (Fig. 8B & C), the ratio showed disturbed NAD^+ homeostasis in both. We confirmed these finding using a commercial NAD^+/NADH assay. In keeping with the absence of complex I loss, changes in NAD^+/NADH were not seen POLG iPSCs (Fig. 8D-F).

11. Fig. 8k: Band intensities of WS5A-NSCs in Phospho-SirT1 blot do not fit to quantification. First band is much stronger than in Ctrl, two other bands weaker than CTRL. This high variation is not reflected in the error bar in L-b.

Answer: We agree that first band has much higher intensity compared to the other WS5A-NSCs in Phospho-SirT1. We believe this is caused by clonal variation of iPSCs, which is a well-described (PMID 28453617) and accepted issue. We also observed some variation in our iPSCs and their NSCs. In order to minimize it, we have included multiple clones from each cell types. In all the other western blotting experiments, we observed the similar band density among all three WS5A clones, as showed below in Figure 17. Therefore, we believe this band is an absolute outlier existing only in one-time experiment and it is reasonable to exclude from the quantification. We also replaced this picture with a better image.

Figure 17

Some more suggestions to improve the impact of the manuscript:

1. Most assays are done in NSCs, but not in much more disease-relevant cell types, such as DA or cortical neurons. I think it would make the paper much stronger if clear phenotypes were shown in cell types present in the adult human body and affected by the disease.

Answer: We would beg to disagree with this point. We have succeeded in differentiating patient-iPSCs into both NSCs and dopaminergic (DA) neurons, however, as is well-known, neurons are harder to acquire and display limited expansion potential. As long as they express the disease phenotype, NSCs provide a better tool for studying disease mechanism and have a much greater potential for drug screening being easier to culture, maintain, freeze and refreeze, as well as count. Further, we can perform assays of mitochondrial function in “live cells”.

In addition, primary NSCs provide a continued source of neurons and glial cells in the brain that further serve as a foundation for development, repair, and functional modulations of human adult neurogenesis. It is, therefore, not surprising that dysfunction of NSCs contribute to an assortment of neurological disorders (Li et al., 2018). Since NSCs can be easily obtained from iPSCs, they provide an alternative model to primary NSCs with the possibility of application in disease-relevant phenotype studies and in drug development for disease therapy that target the specific phenotypes identified. Furthermore, we provide in our study evidence to show that NSCs recapitulate the disease phenotypes as shown in patient specific DA neurons generated from iPSCs and patient postmortem brain tissue. Therefore, we are quite convinced that NSCs provide a good tool to study POLG related diseases. In addition, since NSC can be grown in large numbers in smaller formats, we hope to use this system to establish a high-throughput screening system in order to identify therapies for these devastating diseases.

2. It would be very interesting and a great opportunity to address selective vulnerability of affected and spared tissues by studying iPSC-derived cell types affected vs. unaffected in disease.

Answer: We do not fully agree with this omission. We performed the analysis and showed the selective vulnerability of NSCs and DA-neurons and compared these to fibroblasts that were unaffected by disease. If the reviewer is suggesting making comparisons to cell types that do not express selective vulnerability to POLG mutation, we would like to ask which cell types are meant and suggested. We have looked at many tissues and find that *POLG* mutation leads to cellular defects in all, despite the fact that some do not express this clinically.

3. Performing stress treatments e.g. on DA neurons could be very interesting to elicit more compelling phenotypes and also see variations in vulnerability of mutant cells.

Answer: We fully agree that performing stress treatments e.g. on either NSCs or DA neurons could be very interesting to elicit more compelling phenotypes. We have started to treat the cells using different compounds in a further project, but we believe it is also very important to understand first the pathogenic mechanisms of POLG mutations, which is the main aim of the present study. Besides, to our knowledge, our studies are the first that show it is possible to recapitulate the neuronal molecular and biochemical defects associated with POLG mutation in a human stem cell model.

4. I would also suggest addressing why the neuronal lineage is particularly prone to loss of mitochondrial copy number.

Answer: This is clearly a question that we do wish to address. From our own and other studies (often on postmortem tissues), we know that neurons develop mtDNA depletion as do hepatocytes. Skeletal muscle cells, however, do not manifest depletion, but instead show multiple deletions. Being able to demonstrate that we can recapitulate the same phenotype seen in PM neurons in a tractable cell-based model and showing how these cells respond with loss of ATP, increased ROS etc., is, we feel, a major step forward and one that should enable us to explore further the question raised by the reviewer.

Referee #3 (Remarks for Author):

This is an interesting study reporting the generation of iPSCs carrying POLG mutations. These iPSCs could represent an useful tool to advance our understanding of the pathogenic mechanisms of POLG mutations.

Unfortunately, however it is this reviewer's opinion that the level of insights into disease mechanisms is very limited in this study. There are no insights into the reason why POLG mutations affect neurons specifically.

Answer: We disagree with this reviewer concerning the lack of insights generated by our studies. As the advisor points out, we are the first that show that it is possible to recapitulate the biochemical and molecular findings associated with the common *POLG* mutations in a tractable model such as neural stem cells. Further, we believe that our studies demonstrate that iPSC-derived neural stem cells can provide a robust model system in which to study tissue specific mitochondrial disease manifestations. Our studies suggested that POLG mutations result in a compromised ETC, leading to both a reduction in OXPHOS (and thus, less ATP), redox changes as well as increased levels of ROS (which “leak” out as a result of the less efficient ETC). This, in turn, can further damage the mitochondria, leading to a vicious feedback loop that often results in neural cell loss. Other pathogenic mutations interfere with the critical processes of mitochondrial mitophagy (the autophagic disposal of redundant or damage mitochondria), with potentially devastating

consequences for the patient. Lastly, we show, for the first time that pathways including cellular senescence and mitophagy are activated by the presence of POLG mutation and are likely contributors to the neuronal death seen in these diseases. Thus, we believe that our studies may provide important insights into disease mechanisms.

I also could not see clear experiments dissecting the mechanisms underlying the neuronal cell loss.

Answer: As clearly demonstrated in the manuscript, we have performed many experiments to dissect the mechanisms underlying the neuronal cell loss, as highlight below,

- **Human iPSCs carrying *POLG* mutations can be differentiated into high-yield neural stem cells.**
- **Both fibroblasts and iPSCs from POLG patients manifested only a partial disease phenotype.**
- **NSCs with disease caused by *POLG* mutations showed energy failure and mtDNA depletion, similar to findings iPSC-derived DA neurons.**
- **POLG NSCs recapitulated the disease phenotypes observed in POLG patient post-mortem tissues.**
- **POLG NSCs showed loss of mitochondrial complex I and abnormal UCP2/Sirt1 mediated NAD⁺ homeostasis associated with overproduction of intercellular and mitochondrial ROS.**
- **Elevated ROS triggered cell senescence and BNIP3-mediated mitophagy, which contributes to pathological mechanisms in mitochondrial diseases.**

We also made a clear model to describe how POLG mutations cause the defect in mitochondrial activity and mitophagy, leading to the neural cell loss, as showed in Figure 9 in the revised manuscript and answer to reviewer#1 , question #5.

Finally, there is no suggestion of potential therapeutic strategies.

Answer: As suggested by reviewer, we suggested the potential therapeutic strategies in POLG-related diseases based on our present studies. As described in revised manuscript " Our studies indicate further that the health potential of targeting NAD⁺ homeostasis will inform clinical study design to identify nutraceutical approaches for combating POLG disease."

Reference:

Houtkooper, R.H., Canto, C., Wanders, R.J., and Auwerx, J. (2010). The secret life of NAD⁺: an old metabolite controlling new metabolic signaling pathways. *Endocr Rev* 31, 194-223.

Konrad, C., Kawamata, H., Bredvik, K.G., Arreguin, A.J., Cajamarca, S.A., Hupf, J.C., Ravits, J.M., Miller, T.M., Maragakis, N.J., Hales, C.M., et al. (2017). Fibroblast bioenergetics to classify amyotrophic lateral sclerosis patients. *Mol Neurodegener* 12, 76.

Li, L., Chao, J., and Shi, Y. (2018). Modeling neurological diseases using iPSC-derived neural cells : iPSC modeling of neurological diseases. *Cell and tissue research* 371, 143-151.

Lorenz, C., Lesimple, P., Bukowiecki, R., Zink, A., Inak, G., Mlody, B., Singh, M., Semtner, M., Mah, N., Aure, K., et al. (2017). Human iPSC-Derived Neural Progenitors Are an Effective Drug Discovery Model for Neurological mtDNA Disorders. *Cell Stem Cell* 20, 659-674 e659.

Sonntag, K.C., Ryu, W.I., Amirault, K.M., Healy, R.A., Siegel, A.J., McPhie, D.L., Forester, B., and Cohen, B.M. (2017). Late-onset Alzheimer's disease is associated with inherent changes in bioenergetics profiles. *Sci Rep* 7, 14038.

9th Jul 2020

Dear Dr. Liang,

Thank you for the submission of your revised manuscript to our editorial office. We have now received the enclosed three reports on it. As you will see, referees 1 and the advisor (ref. 4 here) are satisfied with the revisions. Unfortunately, referee 2 is not and the reasons still pertained to technical issues not addressed at all.

After intensely discussing the issue, we decided to reject the article because technically the study does not meet major standards in the field. By being so, as referee #2 highlights, the main conclusions of the study can not be ascertained to the POLG mutations without a doubt, possibly raising concerns regarding the conclusiveness of the data and possibly casting doubts whenever the model will be used by others.

This said, because the three referees agree on the high interest of the study and for a much needed model of POLG mutation in mitochondrial diseases, we would, however, have no objection to consider a new manuscript on the same topic if at some time in the near future you obtained data that would considerably strengthen the message of the study and address the referee 2's concerns in full. To be completely clear, however, I would like to stress that if you were to send a new manuscript this would be treated as a new submission rather than a revision and would be reviewed afresh, in particular with respect to the literature and the novelty of your findings at the time of resubmission. If you decide to follow this route, please make sure you nevertheless upload a letter of response to the referees' comments.

At this stage though, I am sorry to have to disappoint you. I nevertheless hope, that the referee comments will be helpful in your continued work in this area and I thank you for considering EMBO Molecular Medicine.

Yours sincerely,

Celine Carret

Celine Carret, PhD
Senior Editor
EMBO Molecular Medicine

***** Reviewer's comments *****

Referee #1 (Comments on Novelty/Model System for Author):

After revision, the manuscript has been improved well.

Referee #1 (Remarks for Author):

After revision, the manuscript has been improved well and ready to get published.

Referee #2 (Comments on Novelty/Model System for Author):

major technical drawbacks remain, see below.

Referee #2 (Remarks for Author):

I was asked to look at a revised version of the manuscript by Liang et al., which describes the generation and characterization of iPSCs from patients with pathogenic POLG mutations. POLG enzyme replicates mitochondrial DNA, and mutations in patients cause neurological defects. The authors describe some potentially disease-related phenotypes in neural stem cells derived from patient iPSCs related to mitochondrial dysfunction, which are novel, as previous reports dealing with POLG mutations in iPSCs did not report similar findings.

I still think that the paper could be a valuable addition to the field, as there is currently no good human system besides patient material to study POLG mutations. However, I criticized several major technical drawbacks of the paper, which in my view largely impair its potential impact, and in the last version I have seen, the authors make clear that they do not intend to resolve them:

1. Lack of isogenic controls: The currently used controls are unmatched to patient lines and, as I explained earlier, comparing to such unmatched and unrelated controls can severely affect the reliability of the phenotypes. To say this clearly, all phenotypes described in the manuscript could be due to genetic background artefacts in patient or control lines. This is why the current state of the art is to compare to gene-corrected isogenic controls, and this is what most people in the field do and expect when publishing findings in iPSCs. The authors do not provide a reason why they do not have those controls, and I do not see any reason to make an exception here, especially for a high-impact journal that publishes high-impact findings.

Instead, the authors claim that CRISPR-based gene correction is hampered with off-targets and imply that it would not be useful for that reason, which is a gross misinterpretation of the field. Off-target-effects are well characterized, occur rarely and there are many good methods to show their absence, so this is no limitation.

Also, for the reasons I described above, the argument that people have compared to unmatched controls for many years and that's why this should still be accepted, is not valid. Generating better (isogenic) controls is established for several years and takes only a few weeks, so I don't see a reason not to use those controls.

Finally, the authors claim that the phenotypes they describe are similar to those found in patient material and that's why no better controls are required. I disagree with this assessment, as the paper also intends to provide a resource to the field. Thus, after publication other people may use the lines to investigate novel disease mechanisms, and at this point the lack of isogenic controls (and the use of retroviral reprogramming, see below) will prevent people from identifying those with the required certainty.

2. Use of retroviral reprogramming: This was state of the art 10 years ago and has been long replaced by non-integrating reprogramming methods (Sendai, mRNA, episomal...). As explained earlier, integration of retroviral DNA can mutate genes, leading to unspecific phenotypes that appear to be caused by the pathogenic mutations. I think this is a no go especially in comparison

with the lack of isogenic controls as there is no way to tell if retroviral transgenes cause any issues. Again, the authors do not provide an understandable reason why they used this form of reprogramming besides higher efficiency and technical challenges, which is not valid as there are several other easy-to-use, high-efficiency, non-integrating methods available. This can e.g. be seen in earlier reports of iPSCs with POLG mutations (Zurita et al. 2016, Chumarina et al. 2019), which used Sendai virus reprogramming. The authors describe in their response that they have validated the lines for pluripotency etc. but this is not the issue here: pluripotency does not exclude that the retrovirus damages a gene that will cause a misleading phenotype in differentiated cells.

Minor points:

When looking over the other minor points, it appeared that the authors would be able to address many, if not all of them in a sufficient way. However, as the major issues in my opinion still do not support acceptance, I will not discuss the minor points here again in detail.

Taken together, this could be a nice and valuable study, but in its current form does not meet major standards in the field, especially for a high-impact journal like EMBO Mol Med. I therefore still cannot recommend publication.

Referee #4 (Comments on Novelty/Model System for Author):

THIS IS THE FIRST PAPER REPRODUCING IN IPSC MUTANT CELLS AND DERIVATIVE NERVE CELLS THE MAJOR MOLECULAR ABNORMALITIES FOUND IN POST-MORTEM BRAINS OF PATIENTS AFFECTED WITH SPECIFIC, SEVERE POLG MUTATIONS LEADING TO SCAE. THE PAPER PAVES THE WAY FOR FURTHER STUDIES USING AD HOC CELLS RATHER THAN (RATHER DISAPPOINTING) MOUSE MODEL TO SHED LIGHT IN THE PATHOGENESIS OF THIS HETEROGENEOUS GROUP OF DISORDERS AND CHALLENGE THIS SYSTEM WITH POTENTIAL THERAPEUTIC APPROACHES.

Referee #4 (Remarks for Author):

THIS IS THE FIRST PAPER REPRODUCING IN IPSC MUTANT CELLS AND DERIVATIVE NERVE CELLS THE MAJOR MOLECULAR ABNORMALITIES FOUND IN POST-MORTEM BRAINS OF PATIENTS AFFECTED WITH SPECIFIC, SEVERE POLG MUTATIONS LEADING TO SCAE. THE PAPER PAVES THE WAY FOR FURTHER STUDIES USING AD HOC CELLS RATHER THAN (RATHER DISAPPOINTING) MOUSE MODEL TO SHED LIGHT IN THE PATHOGENESIS OF THIS HETEROGENEOUS GROUP OF DISORDERS AND CHALLENGE THIS SYSTEM WITH POTENTIAL THERAPEUTIC APPROACHES. THE AMENDMENTS INTRODUCED IN THE PRESENT VERSION OF THE PAPER ARE USEFUL AND IMPROVE THE QUALITY OF THE WORK.

As a service to authors, EMBO provides authors with the possibility to transfer a manuscript that one journal cannot offer to publish to another EMBO publication. The full manuscript and if applicable, reviewers reports are automatically sent to the receiving journal to allow for fast handling and a prompt decision on your manuscript. For more details of this service, and to transfer your manuscript to another EMBO title please click on Link Not Available

Dear Senior Editor Celine Carret,

We are writing concerning the decision on our manuscript (EMM-2020-12146-V3). We were extremely disappointed that this was rejected particularly since the decision was based solely on technical criteria that we feel do not preclude our conclusions and which, in our opinion, are incorrectly applied.

We carefully responded to all of the comments made by the reviewers following the first round of evaluation including the comments repeated again by referee #2. We would take issue with the comments made by this reviewer:

1. Isogenic controls:

We agree that the current state of the art is to use gene-corrected isogenic controls, and that this is becoming the norm when publishing findings in iPSCs. We disagree that this experimental design applies to all iPSC studies, particularly in the case of disease modeling where the presence of many different mutations makes the generation of individual controls impracticable. In studies of POLG mutations, there have been no publications so far using isogenic controls to study compound heterozygous mutations such as that used in our study (CP2A patient).

This has been stated in other studies - "Unfortunately, the high efficiency of genome cutting, and repair makes the introduction of heterozygous alleles by standard CRISPR/Cas9 technique impossible." (EauClaire and Webb 2019).

In contrast, many studies have used age/gender matched controls from healthy individuals to compare with disease samples. This point was previously raised by referee #2 who comments concerning isogenic controls - "If this is not possible for a specific reason (that needs to be clearly stated), comparison of multiple independent patient lines showing the same phenotypes to age-/gender- etc. matched lines". This we have done. Further, others have applied the strategy of using healthy controls: for example, Margarita Chumarina et al, reported using healthy iPSC control for the comparison of patient iPSC sample with POLG1 mutation (Chumarina, Russ et al. 2019). In our revised manuscript, we increased the number of controls and used a total of 7 different healthy control clones including age matched controls which confirmed our disease modeling results.

Thus, we believe our study meets the highest current standards for an iPSC study.

2. Use of retroviral reprogramming:

Reprogramming approaches to generate iPSCs have developed very quickly. Retroviral reprogramming is one of the traditional ways to produce iPSCs even though newer tools have been established. Furthermore, many current researchers still use this traditional approach to generate iPSCs for disease modelling. Our co-author, Gareth Sullivan, who provided all the iPSC lines in our study has published several papers using the same control retrovirally reprogrammed iPSC lines. These are presented in the following publications (Siller, Greenhough et al. 2015, Mathapati, Siller et al. 2016, Siller, Naumovska et al. 2016, Siller and Sullivan 2017). Moreover, a recent study in Nature Communications (Araki, Hoki et al. 2020) reported using reprogrammed stem cells including two retroviral iPSC lines.

We would also add that our iPSCs are very well characterized and show the appropriate pluripotent stem cell markers both at protein and mRNA level, as well as the ability to differentiate into all three germ layers. Lastly, as a supplementary experiment in our revised version, we included one iPSC line made with Sendai virus and carried out all the experiments in the disease modelling part to confirm our findings in comparison to patient NSCs.

Thus, we fail to understand the suggestion that our results, obtained using 7 healthy controls and patient samples generated by retroviral and Sendai reprogramming, are invalid because "integration of retroviral DNA can mutate genes". This is counterintuitive: If the virus always integrated in the same site, it would produce the same phenotype in all lines; if it inserted in different sites, this would produce multiple and different phenotypes. It is worth restating, therefore, that our findings in patient iPSC matched exactly what we found in post-mortem studies, a point made forcibly by referee #4. Therefore, we believe that retroviral reprogramming is still a useful tool and one that is widely accepted in stem cell research field and different high-impact journals.

Other points made by referee#2:

1. "the authors make clear that they do not intend to resolve them":

We disagree. We have made every attempt to answer the comments made by this referee. We provided more controls that we differentiated into NSCs, and we repeated all the assays to show that the phenotype seen in our patient cells was real.

2. "Generating better (isogenic) controls is established for several years and takes only a few weeks"

We completely disagree with this comment. This suggestion is based solely on the CRISPR-editing step. Following editing the cells must be extensively characterized and thereafter differentiated. Using iPSCs to study neurological diseases such as POLG disease is a very time-consuming research. Our studies, which included generating iPSC, characterizing all the different clones (we have in total 33 clones), differentiating them into NSCs, characterizing NSCs, exploring the diseases phenotypes, investigating the disease mechanisms, and repeating all the experiments have taken between 3-4 years to complete.

We understand that the main aim of a high-impact journal such as EMBO is to publish cutting-edge research that will remain relevant despite technical advances. In our opinion, in contrast to the other referees, referee #2 is making inappropriate demands that reflect more a blinkered belief in fashionable technology rather than the search for scientific veracity.

We sincerely believe that our work will provide a benchmark for future research into mitochondrial disease caused by POLG mutations and ask that you reconsider the decision.

Thank you and looking forward to your response.

Yours sincerely,
Kristina Xiao Liang & Laurence Bindoff

16th Jul 2020

Dear Dr. Liang,

I had time to review your appeal and decided to send the manuscript, referee #2 comments and your response to it to an editorial advisor which expertise this time relied on stem cells and iPSCs.

Please review this advisor comment below:

"[...] as a stem cell biologist I would say the referee [#2] is perfectly correct. Retroviral reprogramming is no longer used for exactly the reasons stated and in addition, many people will not be able to use them for standard experiments as a resource because a higher lab safety level is required in many countries for retroviral lines. A stem cell journal would like[ly] not accept without isogenic controls since there is no such thing as a " healthy control" (you have to sequence and check for all disease associated predisposition factors for the disease or phenotype for example since no one knows what they will develop in later life). So the stem cell biology is definitely not state of the art. I'm pretty sure another [stem cell] journal [...] wouldn't accept it.

However, a more medically oriented journal like EMBO Mol Med might be more interested because of novel insights it gives on a disease or condition, independent of its value or otherwise as a resource. So if this is the case and you like the novelty, I would suggest you do go ahead but make sure the authors very clearly state the shortcomings of their study and do not emphasise the resource aspects."

We do like the novelty of the findings and appreciate the need for such a model for the field of mitochondrial diseases. As you can read for yourself, the stem cell aspect of the work is suboptimal. Still, we would like to invite an editorial revision of this work for publication as we believe that indeed, as our advisor said, your work provide novelty and insights directly relevant to human disease. This said, we need you to clearly state and discuss the shortcomings of your study that are 1) the lack of isogenic controls and 2) the viral reprogramming and make sure to refrain from offering this model as a resource for the community.

Please submit your revised manuscript within two weeks. I look forward to seeing a revised form of your manuscript as soon as possible.

Yours sincerely,
Celine Carret

Celine Carret, PhD
Senior Editor
EMBO Molecular Medicine

Editor's comments:

"We do like the novelty of the findings and appreciate the need for such a model for the field of mitochondrial diseases. As you can read for yourself, the stem cell aspect of the work is suboptimal. Still, we would like to invite an editorial revision of this work for publication as we believe that indeed, as our advisor said, your work provides novelty and insights directly relevant to human disease. This said, we need you to clearly state and discuss the shortcomings of your study that are 1) the lack of isogenic controls and 2) the viral reprogramming and make sure to refrain from offering this model as a resource for the community."

Answers: We would first thank the editor for the positive comments concerning the novelty of the findings and importance on our achievement in generating iPSC derived neural stem cells as a model for the field of mitochondrial diseases. As suggested, we have discussed in the revised manuscript the technical shortcomings of our study including 1), lack of isogenic controls; 2) using the viral reprogramming, as described on manuscript, page 12-13-
"There are two technical questions pertaining to our study that require discussion. Firstly, isogenic controls were not used. We recognise that the current state of the art is to compare patient samples to gene-corrected isogenic controls, usually made by CRISPR-based gene editing. In our studies, however, we considered that the use of multiple, age matched controls (n=5) remained a viable alternative. This choice was, in part, driven by the presence of a compound heterozygous patient (CP2A patient). Further, we would point out that many studies still use age/gender matched controls from healthy individuals as disease comparators as exemplified by the recent study of another POLG mutation, p.Q811R (Chumarina, Russ et al. 2019). Nevertheless, in cases of loss of function mutations, and to minimize background-specific confounding factors, we agree that gene-corrected isogenic controls or a well-executed rescue experiment should be conducted. Secondly, we used retroviral reprogramming to generate most of the cell lines studied. Since the first experiments using integrating retroviral vectors, various approaches to deliver the reprogramming genes have been described, most notably newer integration-free and viral-free methods (Stadtfeld and Hochedlinger 2010, Bellin, Marchetto et al. 2012, Robinton and Daley 2012). These new techniques avoid insertional mutagenesis and transgene reactivation and can minimize variability between reprogrammed cell lines. While non-integrating methods can benefit both disease modelling and the future use in cell transplantation therapies, many excellent studies of disease modelling have used the retroviral system (Bellin, Marchetto et al. 2012). This may, in part, be explained by the limited protocol efficiencies of the new methods. Among the newer techniques, episomal plasmids and Sendai virus are the more commonly used (Okita, Matsumura et al. 2011, Chumarina, Russ et al. 2019), but reprogramming using synthetic mRNAs is also possible. As the non-integrating reprogramming technology improves, these technologies will be preferred in future studies."

Referee's comments:

We are grateful for Referee #1, #2 and #4 for their positive comments on our manuscript. Here, we also carefully responded to the comments made by the reviewers following the first round of evaluation including the comments repeated by Referee #2.

1). Lack of isogenic controls: The currently used controls are unmatched to patient lines and, as I explained earlier, comparing to such unmatched and unrelated controls can severely affect the reliability of the phenotypes. To say this clearly, all phenotypes described in the manuscript could be due to genetic background artefacts in patient or control lines. This is why the current state of the art is to compare to gene-corrected isogenic controls, and this is what most people in the field do and expect when publishing findings in iPSCs. The authors do not provide a reason why they do not have those controls, and I do not see any reason to make an exception here, especially for a high-impact journal that publishes high-impact findings.

Instead, the authors claim that CRISPR-based gene correction is hampered with off-targets and imply that it would not be useful for that reason, which is a gross misinterpretation of the field. Off-target-effects are well characterized, occur rarely and there are many good methods to show their absence, so this is no limitation.

Also, for the reasons I described above, the argument that people have compared to unmatched controls for many years and that's why this should still be accepted, is not valid. Generating better (isogenic) controls is established for several years and takes only a few weeks, so I don't see a reason not to use those controls.

Finally, the authors claim that the phenotypes they describe are similar to those found in patient material and that's why no better controls are required. I disagree with this assessment, as the paper also intends to provide a resource to the field. Thus, after publication other people may use the lines to investigate novel disease mechanisms, and at this point the lack of isogenic controls (and the use of retroviral reprogramming, see below) will prevent people from identifying those with the required certainty.

Answers: We discuss in our revised manuscript that the lack of isogenic controls is a technical drawback in our study. We agree that the current state of the art is to compare the patient samples to gene-corrected isogenic controls, usually made by genome editing such as CRISPR-based gene correction. We disagree, however, that this experimental design applies to all iPSC studies, particularly the case of disease modeling where the presence of different mutations makes the generation of individual controls impracticable. In studies of POLG mutations, there have been no publications so far using isogenic controls to study compound heterozygous mutations such as our patient CP2A.

In contrast, many studies have used age/gender matched controls from healthy individuals to compare with disease samples (Chumarina, Russ et al. 2019, Grunwald, Stock et al. 2019, Castro-Viñuelas, Sanjurjo-Rodríguez et al. 2020). This point was previously raised by referee #2 who commented concerning isogenic controls - "*If this is not possible for a*

*specific reason (that needs to be clearly stated), comparison of multiple independent patient lines showing the same phenotypes to age-/gender- etc. matched lines". This we have done. Further, others have applied the strategy of using healthy controls: for example, Margarita Chumarina et al, reported using healthy iPSC control for the comparison of patient iPSC sample with *POLG1* mutation (Chumarina, Russ et al. 2019). In our revised manuscript, we increased the number of controls and used a total of 7 different healthy control clones including age matched controls to confirm our disease modeling results.*

Thus, we believe our study meets the current standards for an iPSC study. Nevertheless, in cases of loss of function mutations, and to minimize background-specific confounding factors, we agree that gene-corrected isogenic controls or a well-executed rescue experiment should be conducted. As recommended, we have added a discussion of the weakness of our study in the revised manuscript - "*There are two technical questions pertaining to our study that require discussion. Firstly, isogenic controls were not used. We recognise that the current state of the art is to compare patient samples to gene-corrected isogenic controls, usually made by CRISPR-based gene editing. In our studies, however, we considered that the use of multiple, age matched controls (n=5) remained a viable alternative. This choice was, in part, driven by the presence of a compound heterozygous patient (CP2A patient). Further, we would point out that many studies still use age/gender matched controls from healthy individuals as disease comparators as exemplified by the recent study of another *POLG* mutation, p.Q811R (Chumarina, Russ et al. 2019). Nevertheless, in cases of loss of function mutations, and to minimize background-specific confounding factors, we agree that gene-corrected isogenic controls or a well-executed rescue experiment should be conducted.*"

2). Use of retroviral reprogramming: This was state of the art 10 years ago and has been long replaced by non-integrating reprogramming methods (Sendai, mRNA, episomal...). As explained earlier, integration of retroviral DNA can mutate genes, leading to unspecific phenotypes that appear to be caused by the pathogenic mutations. I think this is a no go especially in comparison with the lack of isogenic controls as there is no way to tell if retroviral transgenes cause any issues. Again, the authors do not provide an understandable reason why they used this form of reprogramming besides higher efficiency and technical challenges, which is not valid as there are several other easy-to-use, high-efficiency, non-integrating methods available. This can e.g. be seen in earlier reports of iPSCs with *POLG* mutations (Zurita et al. 2016, Chumarina et al. 2019), which used Sendai virus reprogramming. The authors describe in their response that they have validated the lines for pluripotency etc. but this is not the issue here: pluripotency does not exclude that the retrovirus damages a gene that will cause a misleading phenotype in differentiated cells.

Answers:

Since the first experiments using integrating retroviral vectors, various approaches to deliver the reprogramming genes have been described, most notably newer integration-free and viral-free methods (Stadtfeld and Hochedlinger 2010, Bellin, Marchetto et al. 2012, Robinton and Daley 2012). These new techniques avoid insertional mutagenesis and

transgene reactivation and can minimize variability between reprogrammed cell lines. While non-integrating methods can benefit both disease modelling and the future use in cell transplantation therapies, many excellent studies of disease modelling have used the retroviral system (Bellin, Marchetto et al. 2012). This may, in part, be explained by the limited protocol efficiencies of the new methods. Among the newer techniques, episomal plasmids and Sendai virus are among the more commonly used (Okita, Matsumura et al. 2011, Chumarina, Russ et al. 2019), but reprogramming using synthetic mRNAs is also possible. As the non-integrating reprogramming technology improves, these technologies will be preferred in future studies.

We would also like to add that our co-author, Gareth Sullivan, who provided all the iPSC lines for our study has published several papers using the same retrovirally reprogrammed iPSC control lines. These are presented in the following publications (Siller, Greenhough et al. 2015, Mathapati, Siller et al. 2016, Siller, Naumovska et al. 2016, Siller and Sullivan 2017). Moreover, a recent study in Nature Communications (Araki, Hoki et al. 2020) reported using reprogrammed stem cells including two retroviral iPSC lines.

We would also add that our iPSCs are very well characterized and show the appropriate pluripotent stem cell markers both at protein and mRNA level, as well as the ability to differentiate into all three germ layers. Lastly, as a supplementary experiment in our revised version, we included one iPSC line made with Sendai virus and carried out all the experiments in the disease modelling part to confirm our findings in comparison to patient NSCs.

Therefore, we believe that retroviral reprogramming is still a useful tool and one that is widely accepted in stem cell research field and different high-impact journals. But, more importantly, we agree on that as the technology becomes more refined, non-integrating technologies will be preferred in the future studies. As recommended, we have added a discussion of the weakness of our study in the revised manuscript - "*Secondly, we used retroviral reprogramming to generate most of the cell lines studied. Since the first experiments using integrating retroviral vectors, various approaches to deliver the reprogramming genes have been described, most notably newer integration-free and viral-free methods (Stadtfeld and Hochedlinger 2010, Bellin, Marchetto et al. 2012, Robinton and Daley 2012). These new techniques avoid insertional mutagenesis and transgene reactivation and can minimize variability between reprogrammed cell lines. While non-integrating methods can benefit both disease modelling and the future use in cell transplantation therapies, many excellent studies of disease modelling have used the retroviral system (Bellin, Marchetto et al. 2012). This may, in part, be explained by the limited protocol efficiencies of the new methods. Among the newer techniques, episomal plasmids and Sendai virus are the more commonly used (Okita, Matsumura et al. 2011, Chumarina, Russ et al. 2019), but reprogramming using synthetic mRNAs is also possible. As the non-integrating reprogramming technology improves, these technologies will be preferred in future studies.*"

3) Other points made by referee#2

1. "the authors make clear that they do not intend to resolve them":

Answer: We cannot fully agree with this comment. We have made every attempt to answer the comments made by this referee. We provided more controls that were differentiated into NSCs, and we repeated all the assays to show that the phenotype seen in our patient cells was real.

2. "*Generating better (isogenic) controls is established for several years and takes only a few weeks*"

Answer: This suggestion is based solely on the CRISPR editing step. Following editing the cells must be extensively characterized and thereafter differentiated. Using iPSCs to study neurological diseases such as POLG disease is time consuming research. Our studies, which included generating iPSC, characterizing all the different clones (we have in total 33 clones), differentiating them into NSCs, characterizing NSCs, exploring the diseases phenotypes, investigating the disease mechanisms, and repeating all the experiments, have taken years to complete.

References:

Araki, R., Y. Hoki, T. Suga, C. Obara, M. Sunayama, K. Imadome, M. Fujita, S. Kamimura, M. Nakamura, S. Wakayama, A. Nagy, T. Wakayama and M. Abe (2020). "Genetic aberrations in iPSCs are introduced by a transient G1/S cell cycle checkpoint deficiency." Nat Commun 11(1): 197.

Bellin, M., M. C. Marchetto, F. H. Gage and C. L. Mummery (2012). "Induced pluripotent stem cells: the new patient?" Nat Rev Mol Cell Biol 13(11): 713-726.

Castro-Viñuelas, R., C. Sanjurjo-Rodríguez, M. Piñeiro-Ramil, T. Hermida-Gómez, S. Rodríguez-Fernández, N. Oreiro, J. de Toro, I. Fuentes, F. J. Blanco and S. Díaz-Prado (2020). "Generation and characterization of human induced pluripotent stem cells (iPSCs) from hand osteoarthritis patient-derived fibroblasts." Scientific Reports 10(1): 4272.

Chumarina, M., K. Russ, C. Azevedo, A. Heuer, M. Pihl, A. Collin, E. A. Frostner, E. Elmer, P. Hyttel, G. Cappelletti, M. Zini, S. Goldwurm and L. Roybon (2019). "Cellular alterations identified in pluripotent stem cell-derived midbrain spheroids generated from a female patient with progressive external ophthalmoplegia and parkinsonism who carries a novel variation (p.Q811R) in the POLG1 gene." Acta Neuropathol Commun 7(1): 208.

Chumarina, M., K. Russ, C. Azevedo, A. Heuer, M. Pihl, A. Collin, E. Å. Frostner, E. Elmer, P. Hyttel, G. Cappelletti, M. Zini, S. Goldwurm and L. Roybon (2019). "Cellular alterations identified in pluripotent stem cell-derived midbrain spheroids generated from a female patient with progressive external ophthalmoplegia and parkinsonism who carries a novel variation (p.Q811R) in the POLG1 gene." Acta Neuropathologica Communications 7(1): 208.

Grunwald, L.-M., R. Stock, K. Haag, S. Buckenmaier, M.-C. Eberle, D. Wildgruber, H. Storchak, M. Kriebel, S. Weißgraeber, L. Mathew, Y. Singh, M. Loos, K. W. Li, U. Kraushaar, A. J. Fallgatter and H. Volkmer (2019). "Comparative characterization of human induced pluripotent stem cells (hiPSC) derived from patients with schizophrenia and autism." Translational Psychiatry 9(1): 179.

Mathapati, S., R. Siller, A. A. Impellizzeri, M. Lycke, K. Vegheim, R. Almaas and G. J. Sullivan (2016). "Small-Molecule-Directed Hepatocyte-Like Cell Differentiation of Human Pluripotent Stem Cells." Curr Protoc Stem Cell Biol 38: 1G 6 1-1G 6 18.

Robinton, D. A. and G. Q. Daley (2012). "The promise of induced pluripotent stem cells in research and therapy." Nature 481(7381): 295-305.

Siller, R., S. Greenhough, E. Naumovska and G. J. Sullivan (2015). "Small-molecule-driven hepatocyte differentiation of human pluripotent stem cells." Stem Cell Reports 4(5): 939-952.

Siller, R., E. Naumovska, S. Mathapati, M. Lycke, S. Greenhough and G. J. Sullivan (2016). "Development of a rapid screen for the endodermal differentiation potential of human pluripotent stem cell lines." Sci Rep 6: 37178.

Siller, R. and G. J. Sullivan (2017). "Rapid Screening of the Endodermal Differentiation Potential of Human Pluripotent Stem Cells." Curr Protoc Stem Cell Biol 43: 1G 7 1-1G 7 23.

Stadtfeld, M. and K. Hochedlinger (2010). "Induced pluripotency: history, mechanisms, and applications." Genes Dev 24(20): 2239-2263.

31st Jul 2020

Dear Dr. Liang,

We are happy to inform you that your manuscript is accepted for publication and will be soon sent to our publisher to be included in the next available issue of EMBO Molecular Medicine.

Please read below for additional IMPORTANT information regarding your article, its publication and the production process.

Congratulations on your interesting work! And thank you for choosing EMBO Mol Med for the publication of your article.

Kind regards,
Celine

Celine Carret, PhD
Senior Editor
EMBO Molecular Medicine

Follow us on Twitter @EmboMolMed
Sign up for eTOCs at embopress.org/alertsfeeds

Corresponding Author Name: Kristina Xiao Liang, Laurence Bindoff

Manuscript Number: EMM-2020-12146-V3